# Steroid hormone-induced wingless ligands tune female intestinal size in *Drosophila*

Lisa Zipper [1], Bernat Corominas-Murtra [2] & Tobias Reiff [1] ✉

Female reproduction comes at great expense to energy metabolism compensated by extensive organ adaptations including intestinal size. Upon mating, endocrine signals orchestrate a 30% net increase of absorptive epithelium. Mating increases production of the steroid hormone Ecdysone released by the *Drosophila* ovaries that stimulates intestinal stem cell (ISC) divisions. Here, we uncover the transcription factor *crooked legs (crol)* as an intraepithelial coordinator of Ecdysone-induced ISC mitosis. For the precise investigation of non-autonomous factors on ISC behaviour, we establish Rapport, a spatiotemporally-controlled dual expression and tracing system for the analysis of paracrine genetic manipulation while tracing ISC behaviour. Rapport tracing reveals that Ecdysone-induced Crol controls mitogenic Wnt/Wg-ligand expression from epithelial enterocytes activating ISC mitosis. Paracrine Wg stimulation is counterbalanced by Crol-repression of *string/CDC25* and *CyclinB* autonomously in ISC. Rapport-based ISC tumours confirm paracrine stimulation through the Ecdysone-Crol-Wg axis on mitotic behaviour, whereas the autonomous anti-proliferative role of Crol in ISC is conserved in models of colorectal cancer. Finally, mathematical modelling corroborates increasing enterocyte numbers and Wnt/Wg-degradation to set a stable post-mating intestinal size. Together, our findings provide insights into the complex endocrine growth control mechanisms during mating-induced adaptations and might help untangling pleiotropic hormonal effects observed in gastrointestinal tumorigenesis.

Generation of offspring is an energetically costly process that triggers multiple physiological adaptations of organs such as liver, pancreas and gastrointestinal tract in various species[1,2]. Survival and fitness of progeny relies on tight control of alimentary tract adaptations to metabolic demands in small rodents, in which daily food uptake during lactation can equal the mother´s body weight[1,3]. It is key to understand regulatory mechanisms for hyperplasia and -trophy of the intestine, as it underlies both, physiological tissue functionality and potential malfunctioning in common diseases such as diabetes, obesity and cancer. Physiological adaptations to mating and pregnancy offer a unique opportunity to explore the nature of the underlying interorgan communication.

In *Drosophila melanogaster* females, gut size is increased to match energy consumption when egg production is initiated[4–8]. Endocrine interorgan communication orchestrates this organ size re-set yielding an enlarged intestine with about a third more absorptive enterocytes (EC)[7,8]. This expansion is orchestrated by systemic release of juvenile hormone (JH) from the neuroendocrine corpora allata and the steroid hormone 20-Hydroxy-Ecdysone (20HE) from the ovaries. Both hormones converge on intestinal progenitors increasing intestinal stem cell (ISC) proliferation and enteroblast (EB) differentiation towards EC fate[4,7–10]. 20HE-dependent increases in ISC proliferation depend on presence of

[1]Department of Biology, Institute of Genetics, The Faculty of Mathematics and Natural Sciences, Heinrich Heine University Düsseldorf, Düsseldorf, Germany.
[2]Department of Biology, University of Graz, Graz, Austria. ✉e-mail: reifft@hhu.de

the Ecdysone-receptor (EcR) and early response genes *Broad, Eip75B* and *Hr3*[4,8], but mating-dependent molecular control mechanisms of how active 20HE-signalling affects the cell cycle in ISC, remained unknown.

Here, we report and characterize how *crooked legs* (*crol*) relays endocrine 20HE into local intraepithelial ISC division control. We detected 20HE-dependent *crol* activation in ISC and epithelial EC. Interestingly, functional experiments revealed antagonizing mitogenic effects of *crol* manipulation in the ISC population on one hand and the EC population on the other hand, a function of Crol that is also exerted by its human orthologue ZNF267. This observation prompted us to invent and establish 'Rapport', a bipartite spatiotemporally controlled tracing and expression system, which enabled us to manipulate EC while tracing non-autonomous effects on labelled stem cell progeny. We designed Rapport to offer highest flexibility and compatibility with existing genetic tools and combined it with a driver for epithelial enterocytes in this study. Using Rapport, we discover that Crol relays systemic 20HE signalling in EC into locally acting paracrine Wnt/Wg ligands. This finding is of high interest as EC are the largest cell population in the midgut and Wnt/Wg signalling pathway is central in homeostasis and malignancies of the fly and mammalian gut[11,12]. Manipulation of the 20HE-Crol-Wg axis in microenvironmental EC non-autonomously controls ISC proliferation during mating-dependent intestinal growth and in neoplastic tumours. Quite the contrary, *crol* expression in ISC acts antiproliferative through the CDC25-orthologue *string* and the mitotic cyclin *Cyclin B*, suggestive for a Crol-dependent mitotic balance.

Mathematical modelling supports our hypothesis of opposing autonomous and non-autonomous mitogenic effects of Crol on ISC and that EC numbers are indeed stabilizing mating-adapted organ size depending on 20HE levels. Interestingly, our discovered dynamic pattern is highly robust and can be derived from the fundamental properties of diffusion and degradation of Wnt/Wg ligands. These opposing cell type-dependent consequences of a single hormonal stimulus on stem cell proliferation inside the same epithelium underpin complex hormonal action on epithelial growth during pregnancy-induced hyperplasia and pleiotropic effects observed in cancer of the intestine.

## Results

### Crooked legs responds to mating-dependent 20HE steroid hormone release

The female fly intestine undergoes a variety of post-mating adaptations including a net increase of the absorptive epithelium[5–7]. Mating-dependent enteroplasticity is orchestrated by two hormones, JH and 20HE, which act directly on ISC mitosis increasing the number of absorptive EC[4,7,8]. Aiming to elaborate Ecdysone-responsive genes exerting ISC division control, we followed leads from developmental studies[13–15] and sequencing approaches that suggested expression of *crooked legs* in the adult *Drosophila* midgut[16,17].

Transgenic flies in which Crol is GFP-tagged (Crol::GFP) confirmed *crol*-expression in the adult midgut of female and male flies. We detected GFP-signal in all four major cell types of the intestine: ISC positive for the Notch-ligand Delta (N and Dl, Fig. S1A-A"), EB identified by N-activity (N-reporter GBE+Su(H)-dsRed, Fig. S1B-B"), EE positive for Pros (Prospero, Fig. S1C-C") and EC, positive for the septate junction marker Dlg1 (Discs large 1, Fig. S1C-C"). It was previously shown that mating induces a physiological adaptation of the posterior midgut (PMG) by size and cell number, which is most pronounced in the R5 region of the PMG[7,8]. Mating induces an ovary-to-gut release of 20HE (Fig. 1E)[8], which we also found to significantly increase Crol::GFP levels when comparing adult mated females (MF) with virgin females (VF) (Fig. 1A-B', E, F). As mating induces both, JH and 20HE, we next confirmed *crol* responsiveness to pharmacological EcR activation by feeding the EcR agonist RH5849 to female (Fig. 1E) and male flies[8].

RH5849 significantly increased Crol::GFP fluorescence intensity in control females (Fig. 1C-D', F) whereas no difference was detected in male flies fed with RH5849 (Fig. S1G–I) underlining previously described differences in hormone responsiveness and sexual identity of cells in the female and male midgut[4,8,18,19]. Previous studies showed that manipulation of 20HE levels can be achieved by genetic ovariectomy using the dominant *ovo*[D1] stock, which diminishes later egg stages of vitellogenesis in ovarioles as a sink for 20HE while keeping the main population of ovary cells producing 20HE active[8,20]. In line with a regulation of *crol* by 20HE, we detected a significant increase of Crol::GFP fluorescence in ovariectomized MF (Fig. S1D–D',F). Interestingly, *crol* responded to mating (Fig. 1A-B'), RH5849 (Fig. 1C-D') and 20HE-induction by ovariectomy (Fig. S1D-F) in ISC/EB (Fig. 1A', B', C', D', Fig. S1D', E' encircled Arm[+]/Pros[-]-cells) as well as epithelial EC (Fig. 1A', B', C', D', Fig. S1D', E' arrows) by significant increases in Crol::GFP intensity (Fig. 1F, Fig. S1F).

Recent studies showed that 20HE is taken up through the Ecdysone importer (EcI) in larval development[21] and the adult midgut[8]. Using heat-shock induced Flp-out clones, we investigated whether Crol::GFP levels respond to EcI-overexpression and knockdown (Fig. 1G-H'). Clones positively marked by *UAS-RFP* ( > RFP, '>' abbreviates Gal4/UAS regulation) also overexpress >*EcI*. In line with previous results[8], >*EcI* facilitates 20HE uptake which led to increased Crol::GFP levels in ISC/EB (Arm[+]/Pros[−]) as well as EC (Fig. 1G, G', I) when compared to wild type ISC/EB and EC (Fig. 1G, G', outlined cells). Reciprocally, a clonal reduction of 20HE uptake by *EcI-RNAi* reduces Crol::GFP levels in ISC/EB and EC (Fig. 1H, H', J) compared to non-clonal ISC/EB and EC (Fig. 1H, H'). Having identified *crol* as 20HE target in the adult midgut cells, we investigated its role in intestinal tissue homeostasis by separately addressing Crol in ISC/EB progenitors and epithelial EC (Fig. 1E).

### Crol and its functional human orthologue ZNF267 control proliferation of intestinal stem cells

First, we manipulated *crol* autonomously in ISC and EB using the 'ReDDM' (Repressible Dual Differential Marker, Fig. S2A) tracing method to observe overall impact on tissue renewal with spatiotemporal control of tracing onset and gene manipulation[22]. Briefly, *ReDDM* differentially marks cells having active or inactive *Gal4* expression with fluorophores of different stability. Combined with the enhancer trap *esg-Gal4*, active in progenitors (ISC and EB), *esg*[ReDDM] double marks ISC and EB driving the expression of *UAS-CD8::GFP* ( > CD8::GFP, '>' abbreviates Gal4/UAS regulation and '>>' lexA/AoP regulation in the following) with short half-life and >*H2B::RFP* with long half-life. Upon epithelial replenishment, newly differentiated EC and EE stemming from ISC divisions retain an RFP[+]-nuclear stain due to fluorophore stability[22]. Crosses are grown at 18 °C in which transgene expression is repressed by ubiquitous tubulin-driven temperature sensitive Gal80[ts]. By shifting adult females to 29 °C, Gal80[ts] is destabilized, in turn enabling temporal control of *esg*[ReDDM]-tracing and additional UAS-driven transgenes in progenitors (Fig. S2A).

After seven days of tracing adult female intestines using *esg*[ReDDM], we found that overexpression of *crol* has an antiproliferative effect decreasing the number of progenitor cells (Fig. 2B, B') about 12-fold (Fig. 2E) compared to controls (Fig. 2A). Reciprocally, RNAi-knockdown (Fig. 2C) and guideRNA (gRNA) mediated excision of *crol* (Fig. S2G) using *esg*[ReDDMCas9] tracing significantly increased the number of progenitors (Fig. 2E, Fig. S2H), new epithelial cells (Fig. S2B) and stimulated ISC division (Fig. S2C) compared to controls (Fig. S2D). Similar results were obtained using independent overexpression and loss-of-function transgenics of *crol* (Fig. S2B, E–H). Underlining functionality of both, the reporter and RNAi stock, *crol-RNAi* driven by *esg*> in ISC and EB reduces Crol::GFP fluorescence (Fig. S2I–L).

We next sought to identify human orthologues of *crooked legs* and by mining databases for zinc finger transcription factors with a high

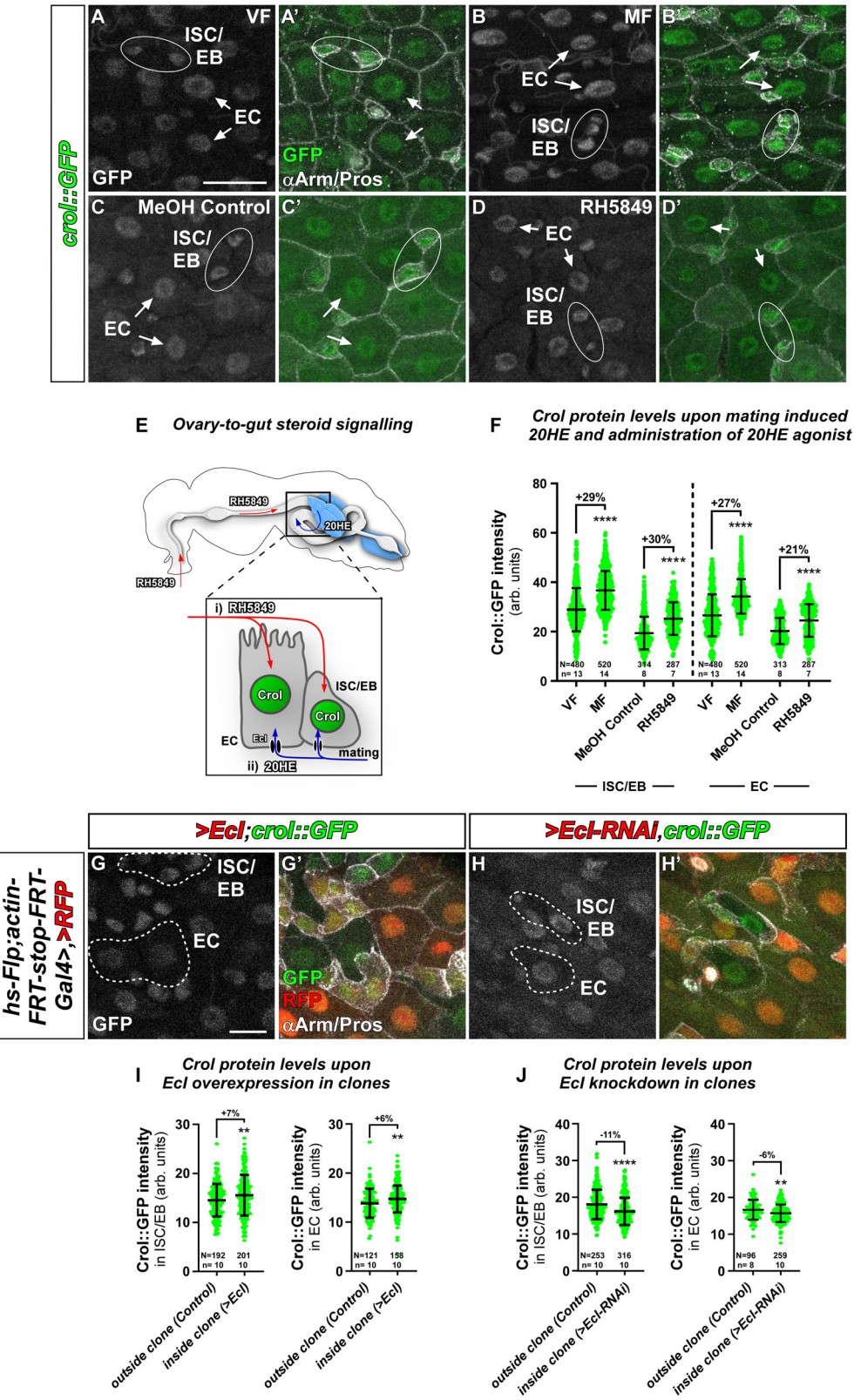

degree of sequence homology to Crol. We isolated the human Krüppel-like zinc finger transcription factor ZNF267 with a high degree of sequence homology (Fig. S2M) using in silico prediction resources (Flybase). To explore the ability of human ZNF267 (hZNF267) to substitute Crol function in intestinal progenitors, we depleted endogenous *crol* by RNAi and expressed hZNF267 at the same time (Fig. 2D) and observed rescue of *crol-RNAi*-induced progenitor accumulations by

hZNF267 (Fig. 2D, E). In line with a role for Crol and hZNF267 in mitosis, hZNF267 regulates cell proliferation and differentiation in liver tissue[23,24]. Crol (Fig. S1) and hZNF267 show wide-ranged expression across human intestinal cell types (GTEx, Proteinatlas) and, like *crol* (Fig. 1I, J), is induced by steroid hormone signalling[25]. This prompted us to investigate how steroid hormones may affect ISC proliferation downstream of Crol and hZNF267.

**Fig. 1 | Crol responds to 20HE steroid hormone release. A–D′** Crol::GFP in R5 of adult midguts of (**A–A′**) virgin female (VF) flies compared to (**B–B′**) mated females (MF), (**C–C′**) MF fed with MeOH (MeOH Control) compared to (**D–D′**) MF fed with RH5849. **A–D** Sole Crol::GFP signal in greyscale and (**A′–D′**) in green colour combined with Arm and Pros antibody staining to identify duplets of ISC/EB (white ellipses) and EC (white arrows). Scale bar is 20 μm. **E** Female fly with midgut and ovaries. Oral RH5849 is (i) absorbed from the midgut lumen into ISC/EB and EC., 20HE from the ovaries travels through the hemolymph (ii) from where it is imported by the Ecdysone importer (EcI). **F** Crol::GFP intensities in ISC/EB and EC upon mating and oral administration of 20HE agonist RH5849. **G–H′** Confocal images showing heat-shock induced Flp-out clones in R5 regions positively marked by *UAS-RFP* and combined with *crol::GFP* and (**G–G′**) *UAS-EcI* or (**H–H′**) *UAS-EcI-*RNAi. (**G–H′**, UAS abbreviated as '>' hereafter in figure panels). Sole Crol::GFP signal is shown in greyscale and (**G′–H′**) in green colour combined with RFP for identification of clones and αArm/Pros for cell type identification. **G–H** Non-clonal areas including ISC/EB and EC are outlined in white dashed lines. Scale bar is 10 μm. **I-J** Quantification of Crol::GFP levels in ISC/EB and EC outside of clones and inside of clones with (**I**) > *EcI* or (**J**) > *EcI-RNAi*. **F, I–J** Scatter dot plots show individual values with indication of means ± SD. *N* and *n* values represent number of cells and number of biological replicas, respectively. Asterisks denote significances from (**F, I–J**) two-sided Mann Whitney *U* tests, and (**I**) unpaired two-sided *t*-tests used for comparison of Crol::GFP intensities in ISC/EB (*p* = 0.0082), (\*\**p* < 0.01;\*\*\*\**p* < 0.0001). Fold changes are shown in percentages. Source data are provided as a Source Data file.

## Crol controls ISC proliferation through String and Cyclin B

During *Drosophila* development, it was shown that the tyrosine protein phosphatase *string* (*stg*, CDC25-orthologue) and the mitotic B-type Cyclin *CycB* are targets of EcR activity[26,27]. Supporting the idea that EcR-signalling controls ISC proliferation through Stg and CycB, EcR agonists not only increase ISC mitosis[4,8], but also *stg* and *CycB* transcript levels (Fig. 2J). In line with previous observations[28,29], up- and down-regulation of *stg* levels in *esg*[ReDDM] traced guts (Fig. 2F, H) reciprocally controlled ISC lineage production encompassing progenitor cells and newly differentiated cells produced by ISC (Fig. 2K). Confirming a function of Stg in mitotic control by Crol downstream of EcR-signalling, ISC proliferation and subsequent increase in lineage production upon >*crol-RNAi* is abolished when >*stg-RNAi* is co-expressed (Fig. 2C, I, K, L). Vice versa, co-expression of >*crol* and >*stg* (Fig. 2G) rescues ISC lineage production (Fig. 2K) and sporadic ISC mitosis as visualized by anti-pH3 staining (Fig. S3F). Comparable results were obtained when we investigated simultaneous depletion of Crol and CycB (Fig. S3A–G). Together, these data suggest an endocrine control of ISC cell cycle exit by Crol (Fig. 2M) acting on *stg* and *CycB* in line with observations during larval development[14,27].

This data suggested an anti-proliferative role for Crol in ISC, whereas we previously observed 20HE signalling in differentiation processes of the intestinal lineage[8]. A function of Crol in ISC proliferation rather than a role in differentiation of EB, is further supported by several lines of evidence: (i) EB express the EB lineage-specifying transcription factor *klumpfuss* (*klu*) and become postmitotic during lineage progression into EC[30–33]. When we manipulated *crol* using *klu*[ReDDM] (Fig. S3H)[34], we revealed no increase in the number of EB nor new EC numbers upon *crol* up- and downregulation (Fig. S3I–M) suggesting no major role in differentiation when compared to strong Ecdysone-induced effectors of differentiation such as Eip75B-A/C[8]. (ii) Klu is described as transcriptional repressor[35], which directly binds *CycB* regulatory regions and is thought to mediate EB cell cycle exit[33]. Interestingly, we detected highest Crol::GFP levels in *klu*-positive EB (Fig. S3Q) pointing to a similar anti-proliferative role for Crol, further supported by our Crol and CycB manipulations (Fig. S3A–G)[27]. (iii) Additionally pointing to a role for Crol and CycB in proliferation control of ISC and EB, *crol-RNAi* (Fig. S3N) and forced *CycB* expression (Fig. S3O) using EB-specific *klu*[ReDDM] result in cell cycle re-entry reflected by occasional mitotic pH3-positive EB (Fig. S3P). Together, our data highlights *crooked legs* as effector of Ecdysone-signalling that autonomously promotes ISC cell cycle exit involving *stg* and *CycB* (Fig. 2M).

## The 20HE-Crol-Wg axis in enterocytes controls non-autonomous Wnt/Wg activity in intestinal stem cells

Intriguingly, *hZNF267*[23,24] as well as *crol*[13,14,27] connect steroid hormone and Wnt/Wg-signalling, which prompted us to investigate whether Crol controls Wnt/Wg-expression downstream of systemic 20HE-signalling during physiological mating-induced intestinal adaptations. Using transgenic flies in which Wg, the primary Wnt-ligand in *Drosophila*, is GFP-tagged (Wg::GFP), we confirmed strong Wg signal at the mid-/hindgut boundary (MHB)[36], the most posterior midgut region known to be patterned by Wnt/Wg signalling[37]. Anterior to the MHB, we detected robust Wg::GFP signal in EC of the mating-responsive R5 region (Fig. 3A–D′)[38,39]. This suggests that Wg acts on ISC in a paracrine manner, which we analysed using the established *frizzled3* (*fz3*) sensor flies for Wnt/Wg signalling pathway activity (Fig. 3K)[40,41]. Examining *fz3-RFP* intensity in posterior midguts, we confirmed Wnt/Wg pathway activity at the MHB[41] and supporting the idea of EC-derived Wg (Fig. 3A–D′) acting on ISC in a paracrine manner, we detected fz3-RFP signal in intestinal progenitors (Arm⁺/Pros⁻, Fig. 3E–H′)[42] in R5 under homeostatic conditions.

Quantification of fluorescence intensities revealed that the EcR agonist RH5849 and *ovo*[D1] increase Wg::GFP fluorescence in EC (Fig. 3A–D′, I, J) and fz3-RFP in ISC/EB (Fig. 3E–J), suggestive for active Wnt/Wg signalling from EC to ISC (Fig. 3K). Mating also increased Wg::GFP (Fig. S4A–C) and fz3-RFP levels (Fig. S4D–F), which is suggestive for a role for paracrine Wnt/Wg signalling during physiological midgut adaptations. Previous work in challenged guts showed autocrine Wnt/Wg signalling between ISC and EB[43]. In contrast, we found that under homeostatic conditions[34] depletion of Wg using >*wg-RNAi* driven in *esg*[ReDDM] flies did not significantly alter intestinal turnover (Fig. S5A–E) nor Wnt/Wg signalling activity using *fz3-RFP* sensor flies (Fig. S5F–I). Together, these findings pointed to EC being the source for Wnt/Wg ligands under homeostatic conditions.

Given Crol::GFP-responsiveness to 20HE (Fig. 1A-B′, F, G–J, Fig. S1D–F) and Wg::GFP (Fig. 3C–D′, Fig. S4A–C) signal in EC, we examined whether *wg* expression is controlled by 20HE and Crol in EC. Therefore, we combined the established EC-driver (*mex* >)[44] with *fz3-RFP*, which enables Wnt/Wg activity assessment in ISC/EB and manipulation of Wnt/Wg ligands from EC employing *tub-Gal80*[ts] for temporal control (Fig. 4A). In line with the idea of a 20HE-Crol-Wg axis, we found that increasing 20HE signalling pathway activity by EC-specific expression of *EcI*[8,21], *crol* as well as *wg* significantly increased *fz3* activity in adjacent ISC/EB (Fig. 4B, E–G′, I, I′). Reciprocally, depletion of *EcI*, *crol* and *wg* in EC, non-autonomously reduced *fz3* activity measured in ISC/EB (Fig. 4B, J–K′, M, M′). Consequently, Wg-depletion downstream of forced *crol* expression reduced paracrine *fz3* activity measured in ISC/EB (Fig. 4C, H, H′), whereas *hZNF267* expression in endogenous *crol*-depleted EC stimulated *wg* expression (Fig. 4D, L, L′). Further supporting Crol acting on *wg* expression control in EC[14,27], we detected a 5.2-fold increase in Wg::GFP intensity upon >*crol* expression (Fig. 4N-O′, Q) and a significant decrease upon >*crol-RNAi* (Fig. 4P–Q).

Combined with previous findings[14,27], our data involves *wg* as a central transcriptional target of Crol in intestinal EC, which in turn non-autonomously stimulates Wnt/Wg signalling pathway activity in ISC/EB. Intrigued by these observations, we investigated whether Wnt/Wg activation in epithelial EC through the 20HE-Crol-Wg axis translates into stem cell driven intestinal homeostasis and size adaptation[4,7,8].

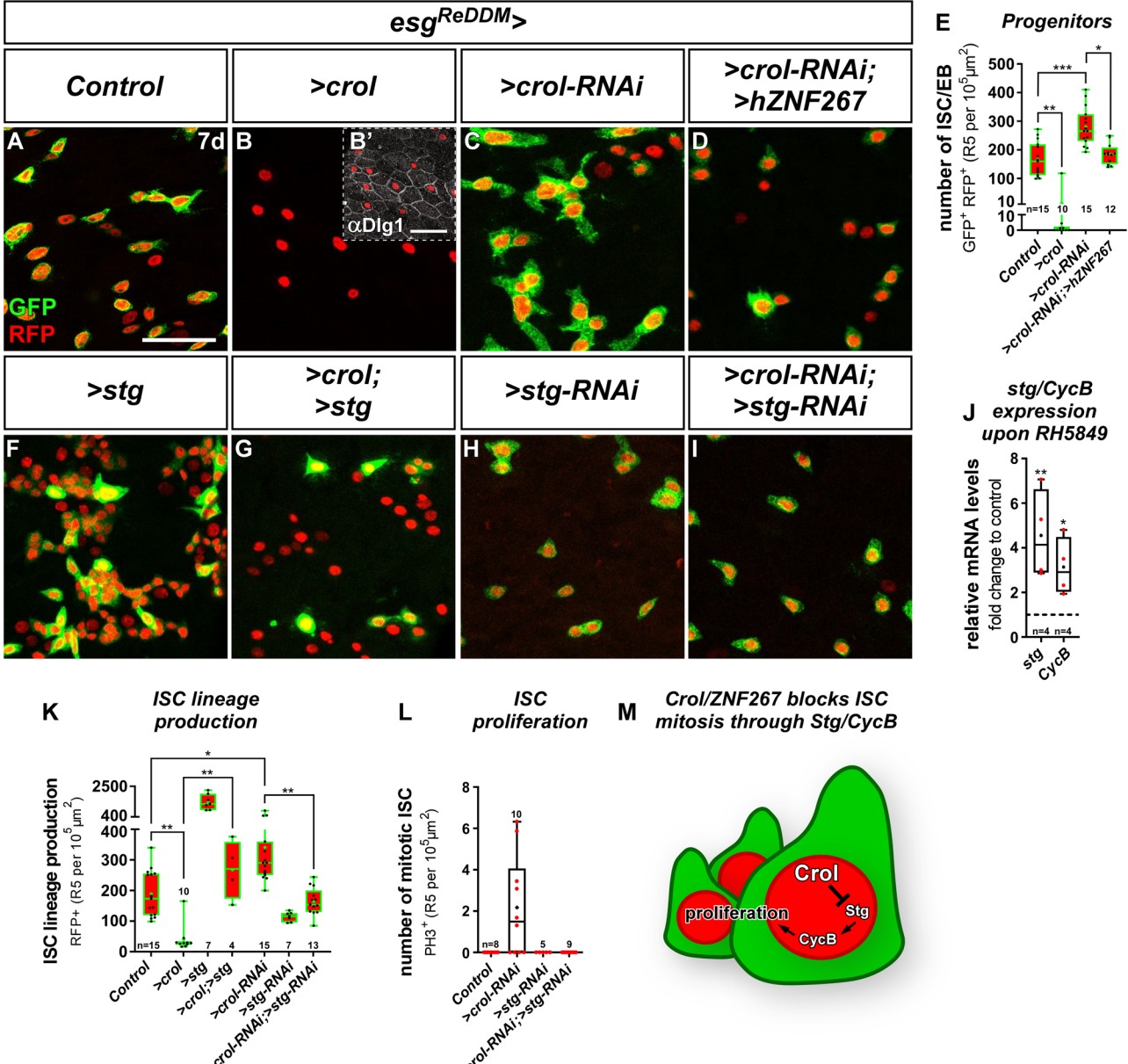

**Fig. 2 | Crol and hZNF267 control ISC proliferation through String/Cyclin B.**
**A–D** Confocal images showing R5 regions of MF midguts after seven days (7 d) of
$esg^{ReDDM}$ tracing showing (**A**) controls with active $esg>$ in ISC/EB driving expression
of >CD8::GFP (membrane GFP) and >H2B::RFP (nuclear RFP), (**B**) ISC/EB specific
overexpression (OE) of *crol* (F003414) with (**B'**) showing anti-Discs large 1 (Dlg1)
staining in septate junctions of EC, (**C**) >crol-RNAi (BL41669), and (**D**) simultaneous
expression of >crol-RNAi (BL41669) and the human Crol orthologue ZNF267
(>hZNF267). Scale bar is 50 μm. **E** Quantification of progenitor numbers upon
$esg^{ReDDM}$ manipulations of *crol* (p = 0.0041; p = 0.0007; p = 0.011). **F–I** Confocal
images of MF midguts after 7 d showing (**F**) ISC/EB specific OE of *string* (*stg*)
(F000926), (**G**) combined with >crol (BL58359), and (**H**) specific KD of *stg* by RNAi
(**I**) combined with >crol-RNAi (BL41669). **J** Quantitative RT-qPCR on midgut cDNA of

MF fed with RH5849 showing relative mRNA levels of *stg* and *Cyclin B* (*CycB*) nor-
malized to MeOH control (p = 0.0052; p = 0.0459). **K**, **L** Quantification of (**K**) ISC
lineage production (ISC/EB and newly differentiated cells, p = 0.002; p = 0.0111;
p = 0.0012; p = 0.0039), and (**L**) number of αPH3⁺ ISC in R5 upon combined
manipulations of *crol* and *stg*. **E**, **J**, **K**, **L** For Box-and-whisker plots: the center is the
median, minima and maxima are 25th and 75th quartile and whiskers indicate full
range of values. All individual values with '*n*' representing numbers of biological
replicas are shown by dots and means are indicated by '+'. Asterisks denote sig-
nificances from multiple comparisons by Kruskal-Wallis test (*p < 0.05; **p < 0.01;
***p < 0.001). Source data are provided as a Source Data file. **M** Schematic showing
inhibition of Stg and CycB by Crol within ISC thereby controlling proliferation.

## 'Rapport' tracing reveals non-autonomous control of intestinal homeostasis through 20HE-Crol-Wg

For the investigation of paracrine effects on stem cell behaviour, we
developed 'Rapport' ('Repressible activity paracrine reporter'), a dual
binary expression system that combines spatiotemporally controlled
transgene expression with ReDDM tracing of stem cell progeny. To
preserve advantage of the existing established Gal4/UAS drivers and
toolbox, we created an entirely new and Gal4-independent lexA/Aop-

based '$esg^{lexReDDM}$' ($esg >> CD8::GFP$, $>>H2B::mCherry::HA$, $tub-Gal80^{ts}$)
tracing system. Importantly, when combined with *mex-Gal4* (*mex >*)
for EC specific expression[44], the lexA-operator driven in $esg^{lexReDDM}$ as
well as Gal4 driven by *mex* are repressed by temperature-sensitive
Gal80[ts], which allows simultaneous temporally controlled onset of
UAS-transgenes as well as $esg^{lexReDDM}$ tracing and Aop-transgenes
(Fig. 5A)[45]. Thanks to the compactness of the $esg^{lexReDDM}$-cassette on
one chromosome, we readily created eight 'flavours' of Rapport

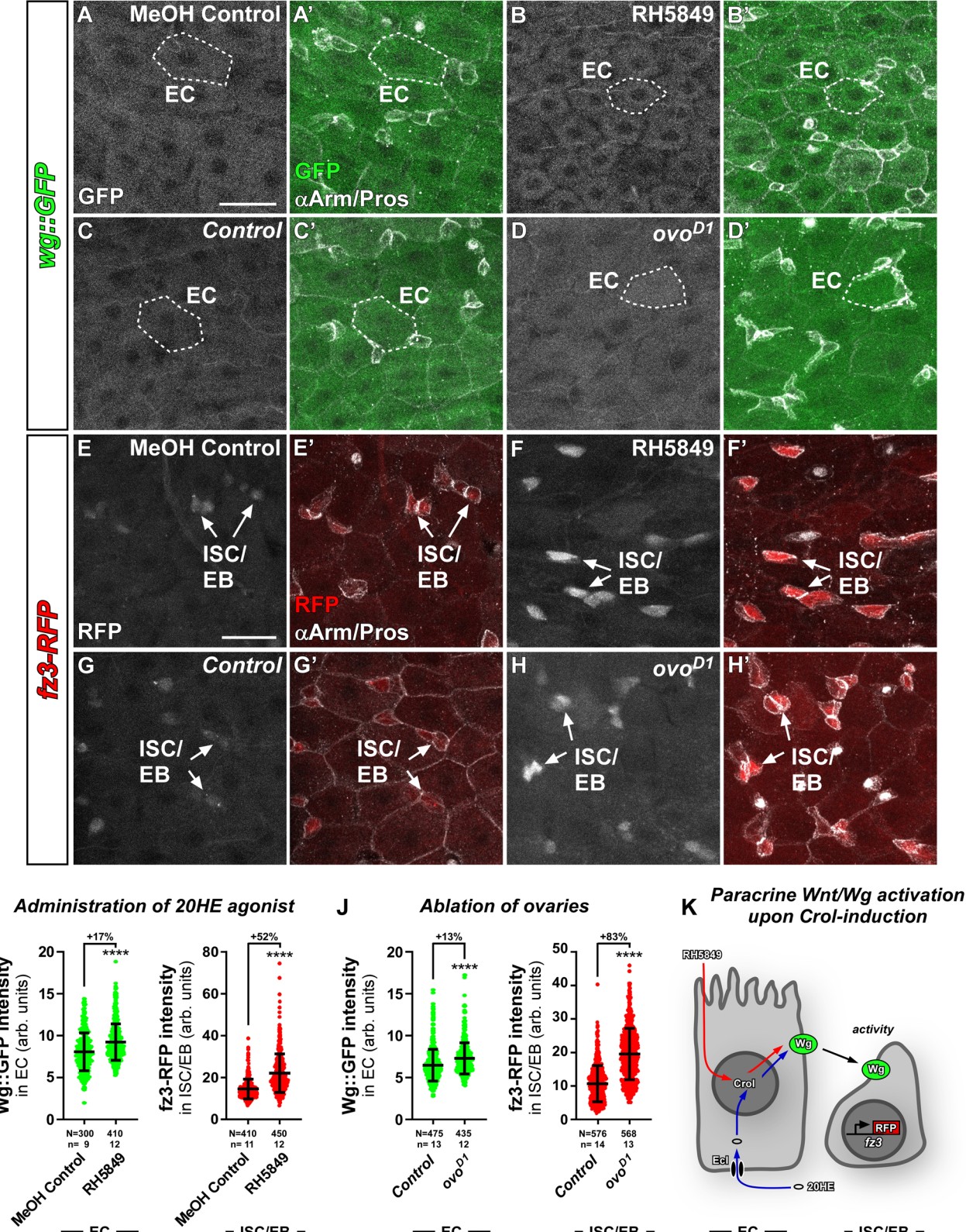

**I** *Administration of 20HE agonist*  **J** *Ablation of ovaries*  **K** *Paracrine Wnt/Wg activation upon Crol-induction*

encompassing all major intestinal cell types and adjacent tissues (Table 1).

We confirmed Rapport tracing functionality by tracing outcrossed controls over three weeks (Fig. S6A–C) and observed an expected linear increase in intestinal renewal (Fig. S6E), while ISC/EB numbers remained constant (Fig. S6D) and comparable to the established Gal4/ UAS-based *ReDDM*-tracing[22]. Previous reports described autocrine

EGFR-stimulation resulting in ISC proliferation and lineage production[28], which we confirmed by crossing Rapport with *mex>* to flies expressing the EGF ligand *Spitz* (>*spi*, TGF alpha homologue)[46] resulting in strongly induced progenitor production (Fig. S6F).

Next, we assessed whether 20HE and Crol controlled Wnt/Wg activity (Fig. 4) stimulates ISC division resulting in EC production and organ size adaptation using Rapport (Fig. 5A). Forced expression of

**Fig. 3 | 20HE induces Wnt/Wg ligand expression in EC and consequently Wnt/Wg signalling activity in ISC/EB. A–D'** Confocal images of GFP-tagged Wingless ligand (*wg::GFP*) in R5 regions of midguts from (**A–A'**) MF fed with MeOH Control compared to (**B–B'**) MF fed with RH5849, and (**C–C'**) *w[1118]* control flies compared to (**D–D'**) dominant *ovo[D1]* MF. **A–D** Sole Wg::GFP signal is shown in greyscale and (**A'–D'**) in green colour combined with αArm and αPros to identify ISC/EB,EE and EC. Exemplary measured EC are outlined by white dashed lines. Scale bar is 20 μm. **E–H'** R5 regions of MF midguts with RFP expressed under control of the *fz3* promotor (*fz3-RFP*) showing Wnt/Wg activity in (**E–E'**) MF fed with MeOH Control compared to (**F–F'**) MF fed with RH5849, and (**G–G'**) *w[1118]* control flies compared to (**H–H'**) dominant *ovo[D1]* MF. Exemplary ISC/EB are marked by white arrows. Scale bar is 20 μm. **I–J** Wg::GFP intensities in EC and fz3-RFP intensities in ISC/EB upon (**I**) RH5849 and (**J**) *ovo[D1]*. **I–J** Scatter dot plots show individual values with indication of means ± SD. *N* and *n* values represent number of cells and number of biological replicas, respectively. Asterisks denote significances from two-sided Mann-Whitney *U* tests (****$p < 0.0001$). Fold changes are shown in percentages. Source data are provided as a Source Data file. **K** Schematic of paracrine Wnt/Wg activation in ISC/EB upon 20HE increment. RH5849 is incorporated into EC from the midgut lumen, whereas 20HE from surrounding haemolymph is imported into EC by EcI. Within EC 20HE hormone and RH5849 activate expression of Crol and Wg. Wg ligand then non-autonomously activates Wnt/Wg signalling in ISC/EB visualized by fz3-RFP activity.

>*EcI*, >*crol*/ >*hZNF267* and >*wg* (Fig. 5C–F) non-autonomously increased progenitor cell number (Fig. 5J) and >*EcI* and >*crol* increase ISC progeny numbers (Fig. 5K) compared to controls (Fig. 5B). Reciprocally, depletion of *EcI, crol* and *wg* reduced progenitor numbers (Fig. 5G–J). Importantly, although >*wg-RNAi* significantly reduces >*crol*-induced progenitor numbers (Fig. 5L–N), this rescue is not fully penetrant (Fig. 5J), which might be related to methodological hurdles such as RNAi-efficiency and additional, unknown mitogens acting non-autonomously on ISC proliferation under the control of Crol.

Our results on the autonomous response to Crol in progenitors suggests that Crol is involved in cell cycle exit in *klu*[+]-positive EB by repressing *CycB* (Fig. S3A–Q)[33]. When expressing >*wg* with Rapport, we detected an upregulation of *CycB* (Fig. 5O) that might point to a de-repression of the CycB promoter by Wg leading to the increase in progenitor production (Fig. 5F, J) and is quite similar in strength to RH5849 induction observed upon EcR-activation (Fig. 2J). Indeed, Wg was previously connected with CycB regulation[27,47] although it is far better known for the regulation of Cyclin D1[48]. Together, our data reveals a direct relay of 20HE activity by Crol into mitogenic paracrine Wnt/Wg signal in enterocytes that is balanced by anti-proliferative Crol autonomously in stem cells. Following this fascinating involvement of endocrine 20HE, Crol and Wg in proliferation control, we sought to investigate their role in intestinal tumour models.

## The 20HE-Crol-Wg mitotic balance is conserved in intestinal tumours

Wnt/Wg signalling is a well-known driver of tumorigenesis with a key role in cancers of the intestine. hZNF267 is upregulated in colorectal cancer (CRC) and regulates cell proliferation and differentiation in epithelial cancer entities[23,24]. We found that hZNF267 expression levels positively and negatively correlate with members of the Wnt/Wg signalling pathway (Fig. S7A). CRC originates from ISC[49], which prompted us to investigate *crol* and hZNF267 in two established intestinal tumour models.

Investigating the autonomous role of Crol/hZNF267 in N loss-of-function (LOF, Fig. S7B) tumours[28,30–32] showed that forced expression of *crol* and *hZNF267* within ISC reduced tumour number (Fig. 6A–C, E) further underlining their anti-proliferative function. Furthermore, >*crol* did not induce new EC, which additionally argues for Crol acting on ISC proliferation rather than factors such as *Eip75B* acting on 20HE-induced differenitation[8]. Reciprocally, we did not observe the expected increase in tumour number upon >*crol-RNAi* (Fig. 6D, E), which is probably attributed to already strong mitotic stimuli such as EGF signalling acting on Stg and Cyclins during tumorigenesis outweighing reduced antiproliferative Crol effects[28,50]. Growing evidence proves that microenvironmental Wnt/Wg ligands are an important contributor to the multifaceted process of colorectal tumorigenesis[51–55]. We thus thought to extend Rapport for the investigation of N-LOF tumours. Therefore, we recombined Aop-driven >>*N-RNAi* with *mex-Gal4* yielding a fly stock with ISC-specific >>*N-LOF* that renders ISC incapable of EC lineage production and instead accumulates ISC- and EEP-like tumoral cells[28,30–32] and allows simultaneous EC-specific manipulations (Fig. S7C, Fig. 6F). Using this model, we investigated

non-autonomous effects of Crol/hZNF267 manipulations in EC and found that >*crol* boosts ISC tumour cell mass by 4-fold leading to confluent tumours along the midgut (Fig. 6G, J) comparable to tumour-induction by microenvironmentally-derived mitogenic EGF ligands[46,56]. Albeit at a lower rate, >*hZNF267* significantly increases tumour number (Fig. 6H, J), whereas lowering of non-autonomous Wg by >*crol-RNAi* (Fig. 6I–J) and direct depletion by >*wg-RNAi* in EC, does not affect N-tumour growth pointing to stronger mitotic stimulus outweighing paracrine Wg in N-tumours[28,50].

Even though Notch-tumours recapitulate important steps of CRC tumorigenesis[28], N is not frequently mutated in CRC patients. Therefore, we also investigated an autonomous function for Crol/hZNF267 in a CRISPR-Cas9 based model of sporadic CRC[57] targeting the most frequently mutated genes (*Apc1,Apc2,p53,Med* and *Pten*) with a multiplex guideRNA array combined with expression of oncogenic >*Ras[G12V]* (Fig. 6O)[57–59]. Initiating these mutations using spatiotemporal induction by *esg[ReDDMCas9]*, CRC from ISC results in severe and pleiotropic cellular phenotypes (Fig. 6K) and early fly demise (Fig. S7D)[57]. Underlining the protective autonomous role of *crol* and hZNF267 (Fig. 6E), their forced expression in CRC avatars significantly improved fly survival (Fig. 6P).

An apparent phenotype of CRC avatars is the loss of epithelial integrity by multilayering of intestinal cells[59–61]. On the cellular level, epithelial deterioration of the tight honeycomb-like intercellular junctional network between EC is visualized by disruption of the septate junction marker *discs large 1* (*dlg1*)[62]. In CRC avatars expressing >*crol* and >*hZNF267*, intact hexagonal Dlg1 junction networks are significantly increased compared to control CRC avatars (Fig. 6K–M, Q). Reciprocally, >*crol-RNAi* significantly increases multilayering (Fig. 6N, Q) strengthening the idea of a tumour suppressive role for Crol and hZNF267. Epithelial deterioration results in hypotrophy of the midgut over time, which strongly reduces midgut length of CRC avatars. Additional >*crol* and >*hZNF267* expression restore midgut hypotrophy to wild-type length (Fig. 6R) and seem to correlate with multilayering and proliferation in predicting survival (Fig. 6P–R).

These functional experiments in intestinal tumour paradigms additionally support the idea of a mitotic balance controlling intestinal growth. Overall, steroidal input on Crol in ISC (Fig. 2) and EC (Fig. 5) provides evidence for an endocrine intestinal size control implicated in mating hyperplasia. Finally, we tested our hypothesis of hormonally controlled intestinal size in a mathematical model.

## Mathematical modelling of endocrine relay by Crol in the control of ISC proliferation underlines complex hormonal actions on intestinal size adaptation

Our model tests whether it is mathematically plausible that these opposing trends between mitotic and anti-proliferative stimuli based on the molecular mechanisms found in this study are able to induce stable population sizes that change consistently with 20HE levels. In the supplementary material we provide all the details for the construction of the model. In addition to our current and previous data[8], we assume constant 20HE levels in VF and a higher constant 20HE level in MF[8]. Then, we test the specific hypothesis whether an 20HE increment is capable to yield a stable, larger organ by temporarily boosting

## Paracrine effects on Wnt/Wg activity in ISC/EB upon EC manipulation

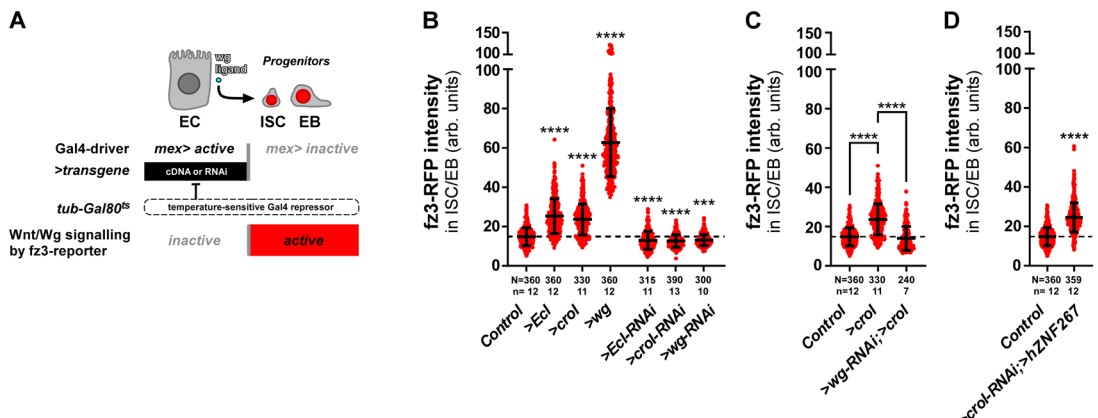

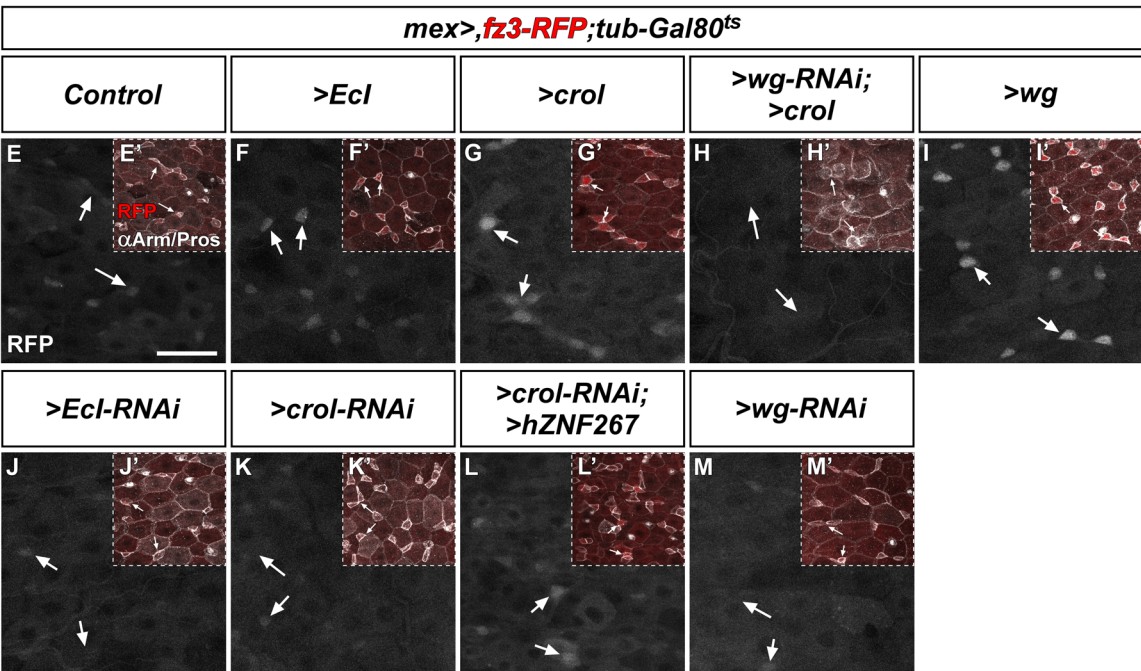

### Wg protein levels upon EC specific Crol manipulations

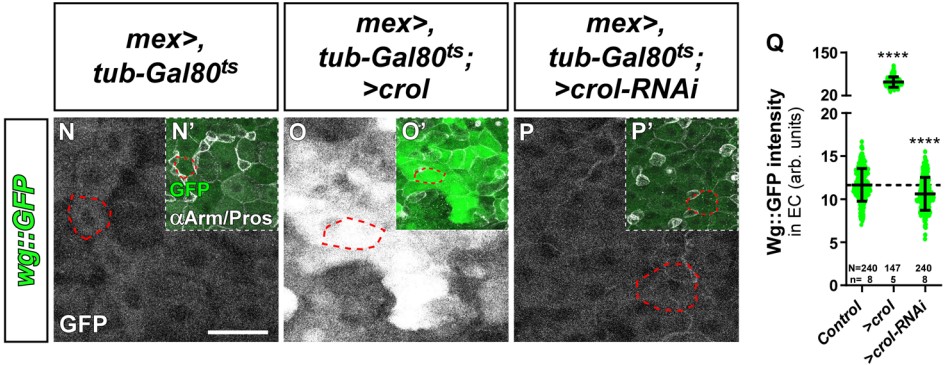

ISC mitosis, which then declines through the increment of EC numbers and the autonomous, anti-proliferative effects in ISC. Our biological findings show that downstream of mating-related 20HE stimulation, Crol autonomously (Fig. 7A, A') and non-autonomously (Fig. 7B, B') controls midgut size and diameter of R5 (Fig. 7A–C). In addition, we found the entire midgut increased in length and consequently cell

numbers (Fig. 7C, Fig. S7F) in concordance with previous reports using cell numbers to address intestinal size[18,19,63,64].

For our model, we postulated constant hormonal input (Fig. 7D) produces equal amounts of mitogenic Wnt/Wg independent of EC numbers. In these conditions the amount of Wnt/Wg produced in a neighbourhood of an ISC decays with the increase of EC, since the

**Fig. 4 | The 20HE-Crol-Wg axis in EC controls Wnt/Wg signalling activity in ISC/EB. A** Schematic of the system enabling EC manipulation and simultaneous visualization of Wnt/Wg activity in ISC/EB. EC specificity of *mex-Gal4* allows manipulation of UAS-driven transgenes that are timely controlled by the ubiquitously expressed temperature sensitive Gal4 repressor (*tub-Gal80ts*) and combined with the fz3-RFP reporter reflecting Wnt/Wg signalling activity in ISC/EB independent of Gal4. **B–D** fz3-RFP intensities in ISC/EB with (**B**) OE and KD of *EcI*, *crol* (F003414, BL41669) and *wg* (>*wg-RNAi*: *p* = 0.0005), (**C**) EC specific OE of *crol* (F003414) combined with >*wg-RNAi*, and (**D**) ectopic expression of >*hZNF267* in *crol* depleted EC (BL41669). **B–D** Experiments have been performed in parallel. **E–M'** Confocal images of *fz3-RFP* combined with *mex>* and *tub-Gal80ts* from R5 regions of MF midguts after 24 h in (**E–E'**) controls, EC specific OE of (**F–F'**) *EcI*, (**G–G'**) *crol* (**H–H'**) with simultaneous KD of *wg*, and (**I–I'**) *wg*, and KD of (**J–J'**) *EcI*, (**K–K'**) *crol* (**L–L'**) with simultaneous expression of >*hZNF267*, and (**M–M'**) KD of *wg* in EC. **E–M** Sole

fz3-RFP signal is shown in greyscale and (**E'–M'**) in red colour combined with αArm and αPros staining for identification of ISC/EB. Exemplary measured ISC/EB are marked by white arrows. Scale bar is 20 μm. **N–P'** MF midguts after 3 d with *wg::GFP* crossed to (**N–N'**) *mex>,tub-Gal80ts* serving as control, (**O–O'**) combined with EC specific OE of *crol* and (**P–P'**) >*crol-RNAi*. **N–P** Sole Wg::GFP signal is shown in greyscale and (**N'–P'**) in green colour combined with αArm and αPros staining for identification of ISC/EB, EE and EC. Exemplary measured EC are outlined by red dashed lines. Scale bar is 20 μm. **Q** Wg::GFP levels in EC upon *mex>* driven manipulations of *crol* (F003414, BL41669). **B–D**, **Q** Scatter dot plots show individual values with indication of means ± SD. *N* and *n* values represent number of cells and number of biological replicas, respectively. Asterisks denote significances from multiple comparisons by Kruskal-Wallis tests (***$p < 0.001$; ****$p < 0.0001$). Source data are provided as a Source Data file.

same amount of hormone per EC must be shared by a larger number of cells[65]. We call $\gamma$ the net amount of hormone, $E$ the net amount of EC and, therefore, $\gamma/E$ the concentration of the hormone sensed by an EC. Generically, the concentration $c(\ell, \gamma/E)$ of Wnt/Wg-ligand decays exponentially with the distance to the source $l$ [66,67]:

$$c(\ell, \gamma/E) = \frac{\gamma}{E} e^{-\ell\sqrt{\frac{k}{D}}} \qquad (1)$$

where $k$ is the degradation rate and $D$ the diffusion constant of Wnt/Wg-ligand—we assume constant mapping between its concentration and 20HE concentration, with proportionality constant 1, for the sake of simplicity. Therefore, considering a section of the intestine as a 1D ring, the average amount of mitogenic Wnt/Wg-ligand reaching equidistantly scattered ISC located at position $x_I$ will be approximately described by:

$$\approx \frac{\gamma}{E} \int_{\Omega} \left\{ e^{-|\ell - x_I|_{\rightarrow}\sqrt{\frac{k}{D}}} + e^{-|\ell - x_I|_{\leftarrow}\sqrt{\frac{k}{D}}} \right\} d\ell \qquad (2)$$

where the integration runs along the whole ring of cells $\Omega$, and the two exponential terms describe the contribution of the diffusion of Wnt/Wg-ligand either clockwise or counterclockwise from the source. The inside of the integral will decline sharply when intestinal size increases, leaving only the close neighbourhood of the ISC to effectively contribute to the Wnt/Wg levels playing an active role in their proliferation, independently of organ size. Indeed, our biological measurements show that ISC numbers remain constant after mating, thus increasing average distances between ISC upon size adaptation (Fig. S7E). Consequently, in the equations of evolution for the number of EC we have a mitotic term that is proportional to the concentration of 20HE ($\approx C\gamma/E$) and an anti-proliferative term that can be assumed to be constant or, in a more general setting, declining slower than the mitotic term as a function of the concentration (Fig. 7E). We assume the net increase of 20HE production after mating to follow a sigmoid shape (Fig. 7D). In addition, the differentiation of ISC into EC occurs in a finite time span $\tau$. This implies that, aside from ISC and EC, we must consider a population of cells differentiating from ISC to EC. We can consider two scenarios for the inhibiting effect of the hormone: either the presence of 20HE 1/blocks the differentiation process or 2/blocks ISC proliferation. Interestingly, the predicted behaviour of the EC population is the same in these two scenarios. Overall, the equations of evolution for the number of transient cells ($u$) and EC ($E$) in case 1/ (Fig. 7F) read:

$$\frac{du}{dt} = C\frac{\gamma}{E}I - \theta u \qquad (3)$$

$$\frac{dE}{dt} = \theta u - \alpha I \qquad (4)$$

where $I$ the number of ISC, $\alpha$ the anti-proliferative rate and $\theta = 1/\tau$ is the rate of EC production out of the population of cells coming from the proliferation of ISC transiting to EC (Fig. 7E). The key result is that incorporating the antagonistic effects of the 20HE hormone the system has a stable, fixed point for the amount of EC that grows and declines consistently with the net amount of 20HE hormone (Fig. 7F). Specifically, the stable point is found at:

$$E^{\star} = C\frac{\gamma}{\alpha}. \qquad (5)$$

Interestingly, similar dynamical equations have been proposed to model the dynamics of cortisol concentration in blood for humans[68]. In the supplementary information file we provide detailed information about the construction and mathematical properties of the model.

As an example, we hypothetically explored how constant EC numbers as generated in N-LOF tumours intestines would affect ISC division dynamics. With constant EC numbers and hormone level (Fig. 7D, Fig. S7G), ISC counts in our model increase in a square-root-like manner (Fig. S7H, I) as the ISC population would provide a growing sink for mitogens such as Wg. In our experiments, block of EC generation and organ size is recapitulated in N-LOF tumours using Rapport (Fig. 6F, Fig. S7C), where upregulation of Crol/hZNF267 significantly increases ISC numbers (Fig. 6G, H, J, Fig. S7I). Our discovered interdependencies between cell population sizes, Wnt/Wg-degradation and their endocrine mitotic balance shed light on the complex endocrine involvement when tumour growth mechanisms are investigated. Together, our model is capable of capturing the emergence of a stable organ size from the antagonism of mitogenic and anti-proliferative hormonal input on ISC thresholding organ size as suggested by our functional data (Fig. 7G, G').

## Discussion

Here we identify the transcription factor *crooked legs* as coordinator of endocrine input into intestinal organ size. The discovered molecular mechanisms underline the complexity of heterologous cellular interactions: a hormonal stimulus bifurcates on stem cells and microenvironment, where it is relayed differentially into an antiproliferative and a mitogenic stimulus. This interdependent opposing crosstalk of forces balances stem cell divisions and ultimately stabilizes organ size, which is sustained by both, empirical observations and mathematical modelling.

Our novel Rapport system contributes to disentangle the underlying endocrine and local signalling organ size control mechanisms, by allowing precise genetic intervention in cell types surrounding the ISC. The independent tracing of the whole stem cell population and easy fluorophore identification of different fate choices ensures robust progeny counts in fluctuating demand situations compared to previous systems[28,46,56]. Endocrine actions on ISC are complex and involve

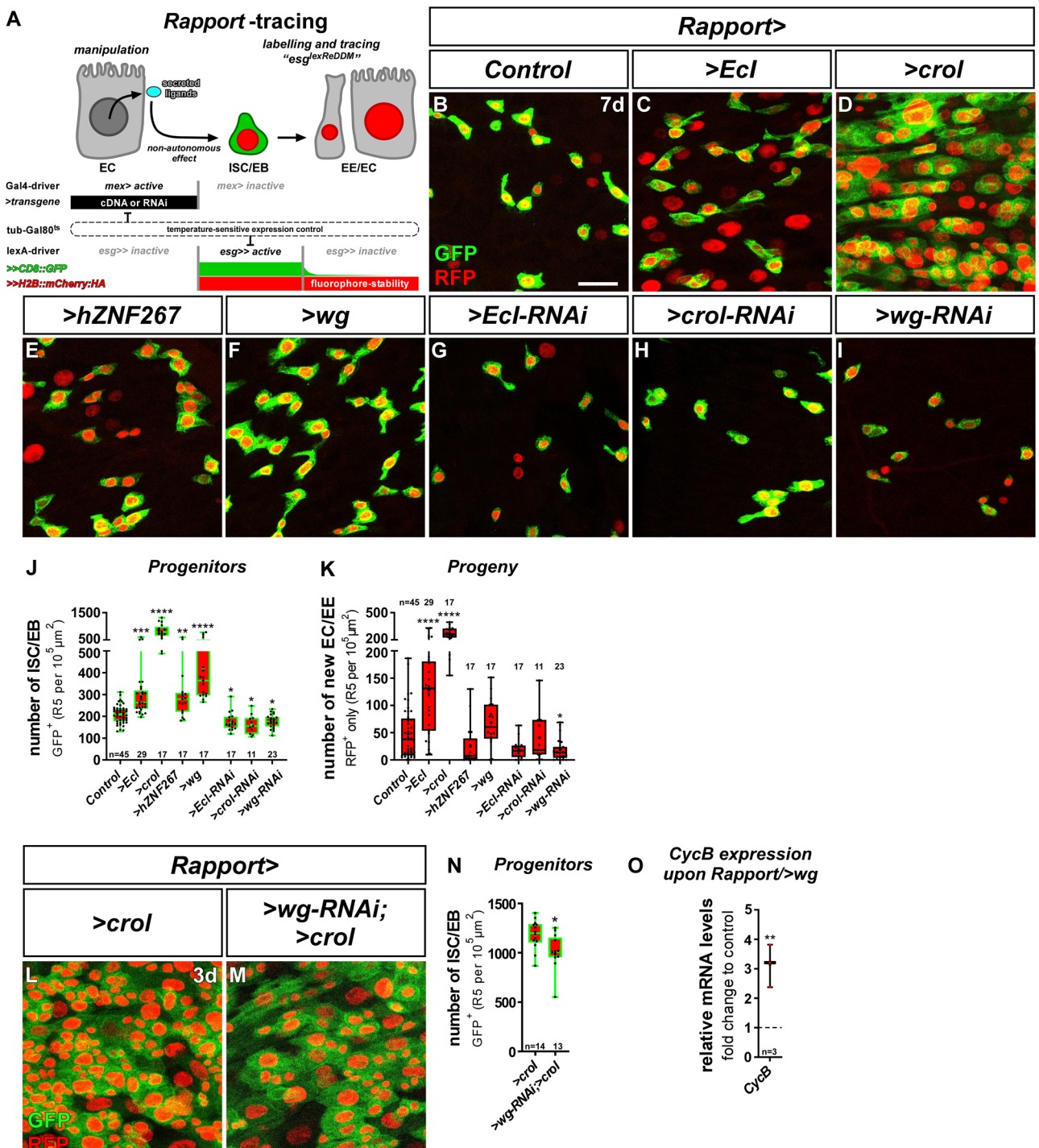

**Fig. 5 | The 20HE-Crol-Wg axis in EC controls intestinal homeostasis. A** Schematic of 'Repressible activity paracrine reporter' (Rapport) tracing system. Rapport consists of the EC *mex-Gal4* driver for UAS-driven transgenes like cDNAs or RNAi. Additionally, an *esg^{lexReDDM}* system based on lexA/Aop allows Gal4/UAS independent labelling and tracing of ISC/EB by *esg-lexA* driven expression of *Aop-CD8::GFP* and *Aop-H2B::mCherry::HA* (lexA/Aop abbreviated as '>>' hereafter). Both expression systems are timely controlled by a tub-Gal80^ts repressor. **B–I** Confocal images showing R5 regions of MF midguts after 7 d of Rapport tracing in (**B**) controls, with OE of (**C**) *EcI*, (**D**) *crol* (BL58359), (**E**) *hZNF267* and (**F**) *wg*, and KD of (**G**) *EcI*, (**H**) *crol* (BL41669) and (**I**) *wg* showing GFP+ and mCherry+ ISC/EB and their differentiated progeny labelled by mCherry only. Scale bar is 20 μm. **J, K** Quantification of (**J**) progenitor cell numbers ( >*EcI*:p = 0.0003; >*hZNF267*:p = 0.0053; >*EcI-RNAi*:p = 0.0467; >*crol-RNAi*:p = 0.0375; >*wg-RNAi*:p = 0.0388) and (**K**) their progeny ( >*wg-*

*RNAi:p* = 0.0285) upon EC specific manipulations using Rapport. **L, M** Confocal images showing R5 regions of MF midguts after 3 d of Rapport tracing with (**L**) sole OE of *crol* (F003414) and (**M**) combined with >*wg-RNAi*. (**N**) Progenitors upon EC specific OE of *crol* alone and combined with >*wg-RNAi* (p = 0.0109). Scale bar is 20 μm. **O** Quantitative RT-qPCR on midgut cDNA of MF with Rapport specific expression of >*wg* showing relative mRNA levels of *CycB*. **J, K, N, O** For Box-and-whisker plots: the center is the median, minima and maxima are 25th and 75th quartile and whiskers indicate full range of values. All individual values with 'n' representing numbers of biological replicas are shown by dots and means are indicated by '+'. Asterisks denote significances from (**J, K**) multiple comparison by Kruskal-Wallis tests and (**N**) unpaired two-sided *t*-test (*p < 0.05; **p < 0.01; ***p < 0.001; ****p < 0.0001). Source data are provided as a Source Data file.

**Table 1 | List of Rapport-tracing variants established in our lab with indication of Gal4 driver used and the cell type/tissue with Gal4-activity**

| Rapport-tracing variant | Gal4-driver | Cell type/tissue with Gal4-activity |
|---|---|---|
| 'EC-Rapport' | *mex-Gal4* | Intestinal EC |
| 'EB-Rapport' | *klu-Gal4* | Intestinal EB |
| 'EE-Rapport' | *Rab3-Gal4* | Intestinal EE |
| 'ISC-Rapport' | *Dl-Gal4* | Intestinal ISC |
| 'VM-Rapport' | *how-Gal4* | Visceral muscle (VM) |
| 'HG-Rapport' | *byn-Gal4* | Hindgut (HG) |
| 'CA-Rapport' | *Aug21-Gal4* | *Corpus allatum* (CA) |
| 'hemocyte-Rapport' | *HmlΔ-Gal4* | Hemocytes |

hormonal dosage and mating status[4,7,8], sex differences[19] and feeding[63]. Hypertrophy upon pregnancy is described in the mammalian gut[1,3] and is well-studied in reproductive organs such as the mammary epithelium where steroid hormones induce extensive remodelling and cancer susceptibility[69–71].

In gastrointestinal tumours such as CRC, epidemiological evidence about the role of steroid hormones remains controversial and ranges from favourable to detrimental[72–78]. Functional studies of both mammalian oestrogen receptors (ER) in rodents underline the complexity of oestrogen signalling in gut tumorgenicity[79,80] and reveal further complexities as pharmacological (E2/P4) and endogenous oestradiol levels differentially affect patient outcome[72,75,81]. Our functional biological and mathematical modelling data provides an initial logic to disentangle complex observations and involves the heterogeneity of tumour cell composition and its capacity to contribute to mitotic signals.

A targeted therapeutic intervention of steroid hormone signalling is supported by: (i) an overall protective tendency of ER signalling in CRC[78]. (ii) Like Crol, hZNF267 is stimulated by ER[25] and is involved in Wnt/Wg signalling (Fig. S7A)[23,24] suggesting conservation of the 20HE-Crol-Wg axis. (iii) Wnt/Wg signalling hyperactivation is central to CRC malignancy and Wnt/Wg-ligands remain indispensable for CRC growth albeit absence of APC[82]. (iv) Effectors of steroid signalling like PPARγ/Eip75B play protective roles in fly pathophysiology[4,8] and human disease[83,84].

Our findings of antagonizing autonomous and paracrine effects of Crol and hZNF267 on tumour growth (Figs. 6, 7, S7) emphasize that targeted genetic investigation is of key importance to understand how mutational heterogeneity and cell type composition differentially affect the proliferative response to hormonal input. Precise intervention and tracing methods such as Rapport open the door for untangling heterogenous findings of epidemiological and functional studies.

## Methods
### Genetics and fly husbandry/fly strains
The following transgenic fly stocks were used: *Gbe + Su(H)-dsRed* (T. Klein), *esg^ReDDM*[22], *esg^ReDDMCas9*[85], *klu^ReDDM*[34], *UAS-wg*[86], *UAS-EcI*[21], *UAS-EcI-RNAi*[21], *mex-Gal4* on II. Chromosome[44], *fz3-RFP*[40], *wg::GFP*[36], *13x LexAop2-H2B::mCherry::HA*[87], *UAS-N^DN* (J. Treisman), *UAS-Ras^G12V*, *>Apc1, Apc2, p53, Med, Pten^sgRNAs*[85], *hs-Flp;;actin-FRT-stop-FRT-Gal4,UAS-RFP* (A. Wodarz).

From Bloomington *Drosophila* Stock Center (BDSC): *ovo^D1* (BL1309), *UAS-crol* (BL56762 and BL58359), *UAS-crol-RNAi* (BL41669 and BL44643), *UAS-stg-RNAi* (BL34831), *UAS-CycB-RNAi* (BL40916), *UAS-CycB^gRNA* (BL80319), *UAS-hZNF267* (BL65797), *esg-lexA* (BL66632), *13xLexAop2-mCD8::GFP* (BL32205), *mex-Gal4* on X Chromosome (BL91367), *UAS-N-RNAi* (BL33616), *nos-phiC31;;attP86Fb* (BL24749), *UAS-spi* (BL63134), *UAS-nls.GFP* (BL4775).

From FlyORF, Switzerland: *UAS-crol* (F003414), *UAS-stg* (F000926).

From Vienna *Drosophila* Resource Center: *crol::GFP* (V318880), *UAS-wg-RNAi* (V104579), *UAS-y^gRNA* (V341666), *UAS-se^gRNA* (V341664).

Throughout the manuscript we use a uniform notation to distinguish between enhancer- and protein trapping. For enhancer traps we separate enhancer and reporter/transgene by '-' as in *Gbe + Su(H)-dsRed* or *UAS-wg*, whereas for protein traps we separate protein and reporter by '::' as in *wg::GFP*. Additionally, Gal4-UAS regulation is indicated by '>', and lexA-Aop by '>>'.

### Food composition and fly keeping
Fly food contained 1424 g corn meal, 900 g malt extract, 800 g sugar beet syrup, 336 g dried yeast, 190 g soy flour, 100 g agarose in 20 l H$_2$O. The ingredients were mixed and cooked for about one hour to reduce bioburden. After cooling down the food 90 ml propionic acid and 30 g NIPAGIN (antimycotic agent) were added, and the food was filled in small plastic vials plugged with foam. Flies were kept at 25 °C. Crosses containing a temperature sensitive Gal80^ts repressor were kept at 18 °C to repress Gal4 activity during development and shifted to 29 °C to start transgene expression in adult flies. Experiments distinguishing between virgin female flies and mated female flies were run on food with twice the amount of NIPAGIN to prevent the induction of tissue renewal caused by pathogenic stress upon mucous formation in the absence of larvae.

### Cloning of *crol^gRNA* construct
Two individual guide RNAs (gRNA) targeting the coding region of genomic *crol* DNA with a distance of 603 bp were chosen using the CRISPR Optimal Target Finder[88]. Primers for amplification of these gRNAs were designed and used as described previously[85]: crol_gRNA_for (5′-ATAAGAAGACCTTGCAGGCCACTGCGTCGTCGCAAGCGGGTTTCAGAGCTATGCTGGAAAC-3′) and crol_gRNA_rev (5′-ATAAGAAGACCCAAACCCCGGTGTTAACTGGACCGCACCTGCACCAGCCGGGAATCGAACC-3′).

Cloning of the amplified crol^gRNA construct into pCFD6_noSapI was performed as described previously[85]. After amplification and verification of the construct, the DNA was injected into embryos of *nos-phiC31;;attP86Fb* flies for integration on the third chromosome.

### Generation of Rapport-tracing
Rapport-tracing consists of '*esg^lexReDDM*' which is a quadruple recombinant of *esg-lexA,13xLexAop2-CD8::GFP,13xLexAop2-H2B::mCherry::HA,tub-Gal80^ts* on the second chromosome. Rapport can easily be combined with any Gal4-driver on the first or third chromosome enabling manipulation of other cell types or even organs and allow manipulation independent of *esg^lexReDDM* tracing. For our investigation of Crol we combined Rapport-tracing with an EC specific *mex-Gal4* on the third chromosome that was generated by mobilization of the *mex-Gal4* P-element in BL91367, and a *klu-Gal4* on the third chromosome for unequivocal identification of ISC. Additional Rapport-tracing variants from our lab can be found in the author response letter and Table 1.

### Cloning of *lexAop-N-RNAi* construct
The *lexAop-N-RNAi* construct was generated as described previously[89]. Genomic DNA of the VALIUM20 based N-RNAi line from the Transgenic RNAi Project (Trip-3, BL33616) was isolated and added to a PCR reaction using the Q5 High-Fidelity DNA Polymerase (NEB) and the following primers: shRNA_GA_F primer (5′-GAGAACTCTGAATAGATCTGTTCTAGAAAACATCCCATAAAACATCCCATATTCA-3′) and shRNA_GA_R1 primer (5′-CTCTAGTCCTAGGTGCATATGTCCACTCTAGTA-3′)[89].

The pWALEXA20 vector[89] was digested with XbaI and NdeI prior to assembling with the amplified N-shRNA by Gibson-assembly. After verification by colony PCR and sanger sequencing constructed plasmids were amplified and injected into embryos of *nos-phiC31;;attP86Fb* flies for integration on the third chromosome.

**Autonomous function of Crol/ZNF267 on tumours**

**Paracrine function of Crol/ZNF267 on tumours**

**Autonomous function of Crol/ZNF267 in CRC avatars**

## Hormone analogue treatments

RH5849 feeding experiments have been performed as described previously[90], except for adjusted feeding durations[90]. The ecdysone agonist RH5849 (DRE-C16813000, DrEhrenstorfer) was diluted in MeOH to create a stock solution of 20 μg/μl. Then, 20 μl of this stock solution was mixed with 4 ml of reheated, liquid fly food in a fresh vial (340 μM final concentration). As a control, an equivalent volume of MeOH was added to the fly food. Flies were starved for four hours prior to RH5849 treatment to make them eat the prepared food immediately. In initial experiments we additionally added the blue dye (Erioglaucine Disodium Salt (E133) (BLD Pharmatech Ltd., Shanghai, China)) known from the so called 'Smurf assay' to the food as described previously[62,91]. The blue dye

**Fig. 6 | The mitotic balance of Crol/hZNF267 is preserved in intestinal tumour models. A–D** Confocal images showing R5 regions of MF midguts with $esg^{ReDDM}$ tracing of dominant negative Notch ($> N^{DN}$) for 3 d in (**A**) controls, with (**B**) >*crol* (F003414), (**C**) > *hZNF267*, and (**D**) >*crol-RNAi* (BL44643). Scale bar is 50 µm. **E** Quantification of ISC tumours encompassing clusters of five or more ISC ($p = 0.0003$; $p = 0.0006$). **F–I** Confocal images showing R5 regions of MF midguts with ISC/EB specific >>*N-RNAi* using *esg-lexA* of the Rapport-tracing system for 7 d in (**F**) controls, combined with EC specific (**G**) >*crol* (F003414), (**H**) > *hZNF267*, and (**I**) >*crol-RNAi* (BL44643). Scale bar is 50 µm. **J** Quantification of ISC tumours Rapport manipulations of Crol/hZNF267 ($p = 0.0045$). **K–N** Confocal images showing R5 of MF midguts after 7 d of $esg^{ReDDMCas9}$ tracing and induction of 'CRC avatars' by multiplex guide RNA array. $esg^{ReDDMCas9}$ allows labelling and tracing of tumour cells in (**K**) controls and with tumour specific expression of (**L**) >*crol* (F003414), (**M**) > *hZNF267* and (**N**) >*crol-RNAi* (BL44643). Scale bar is 50 µm. **O** Schematic of 'CRC avatars' combined with $esg^{ReDDMCas9}$ tracing with oncogenic >*Ras^{G12V}* and CRISPR/ Cas9 induced knockout of *Apc1,Apc2,p53,Med* and *Pten*. **P** Kaplan-Meier estimation of survival in flies with $esg^{ReDDMCas9}$ induced knockout of *yellow* (*y*) serving as 'mock'-gRNA control and CRC avatars combined with manipulations of Crol/hZNF267. Number of analyzed flies is indicated by 'n' and asterisks denote significances from Kaplan-Meier estimation (****$p < 0.0001$). (**Q, R**) Quantification of (**Q**) multilayered tissue encompassing percentage of area with disrupted αDlg1 staining ($p = 0.0006$; $p = 0.0205$) and (**R**) midgut length as indicator of epithelial deterioration (CRC avatars; >*crol*;$p = 0.0004$). **E, J, Q, R** For Box-and-whisker plots: the center is the median, minima and maxima are 25th and 75th quartile and whiskers indicate full range of values. All individual values with 'n' representing numbers of biological replicas are shown by dots and means are indicated by '+'. Asterisks denote significances from multiple comparisons by (**E–J**) Kruskal-Wallis tests and (**Q, R**) One-way ANOVA (*$p < 0.05$; **$p < 0.01$; ***$p < 0.001$; ****$p < 0.0001$). Source data are provided as a Source Data file.

allowed us to determine the food's passage time through the gut and to establish a feeding duration of 48 h for our final experiments. RH5849 treatments were performed at a temperature of 25 °C.

## Flp-out clones
Flp-out clones were induced in midguts by flippase under control of a *heat-shock* promotor. Flippase expression was activated for 45 min in a 37 °C-water bath to induce positively marked clones and expression of >*EcI* and >*EcI-RNAi* within the marked clones. Guts were dissected 3 days after clone induction.

## Immunohistochemistry
Guts of adult female flies were dissected in 1 × PBS and transferred into glass wells containing 4% PFA immediately after dissection. After 45 min of fixation the guts were washed once by replacing the PFA with 1 × PBS. Primary antibodies were diluted in 1 × PBS with 0.5% Triton-X and 5% normal goat serum. The incubation with primary antibodies (1:250 anti-Arm [mouse; Developmental studies Hybridoma Bank (DSHB)]; 1:200 anti-Dl [mouse; Developmental studies Hybridoma Bank (DSHB)]; 1:250 anti-Dlg1 [mouse; Developmental studies Hybridoma Bank (DSHB)]; 1:5000 anti-PH3 (Ser10), Mitosis Marker [rabbit; Sigma-Aldrich];1:250 anti-Pros [mouse; Developmental studies Hybridoma Bank (DSHB)]) was performed on an orbital shaker at 4 °C overnight. The guts were washed with 1 × PBS prior to incubation with secondary antibodies (1:500 Goat anti-Mouse Alexa647 [Invitrogen]; 1:500 Goat anti-Mouse Alexa561 [Invitrogen]; 1:500 Goat anti-Rabbit Alexa647 [Invitrogen]) and DAPI (1:1000; 100 µg/ml stock solution in 0.18 M Tris pH 7.4; DAPI No. 18860, Serva, Heidelberg) for at least 1½ h at RT. After washing with 1 × PBS for a last time, the stained guts were mounted in a drop of Fluoromount-G Mounting Medium (Electron Microscopy Sciences) mixed with a drop of 1 × PBS on a microscope slide and covered with a coverslip. For squeezing of the mounted tissues, a defined weight of 28.5 g was put on top of the coverslip.

## Image acquisition
After immunostaining posterior midguts were imaged using a LSM 710 confocal microscope (Carl Zeiss Microscopy GmbH, Germany) with an 40× objective. Image resolution was set to at least 3440 × 3440 pixels. About four to five confocal planes with 1 µm interval were scanned and combined into a Z-stack to image one cell layer with all different cell types.

For comparison in midgut length, diameter and total cell number of VF to MF, the entire midguts from MHB to proventriculus were imaged using a 10× objective and the tile-scan mode.

For length measurements of midguts from CRC avatars flies were imaged with an Axioplan2 microscope equipped with a PixelFly camera. Images were taken using a 5× objective and the AxioVision Rel. 4.7 software.

## Quantification of cell numbers, midgut lengths, and diameters
Fiji (ImageJ 1.51 n, Wayne Rasband, National Institutes of Health, USA) was used to calculate maximum intensity images out of Z-stacks from posterior midguts. Progenitor cell numbers and numbers of progeny from $esg^{ReDDM}$[22] experiments were counted manually using Fiji. Numbers of DAPI positive cells in R5 were analysed semi-automatically by self-written macro for Fiji (macro available from the authors), whereas total cell numbers of entire midguts were counted manually. Midgut lengths and diameters were analysed manually in Fiji using the freehand line tool.

## Quantification of fluorescence intensities
For intensity measurements of fluorophores, posterior midguts were scanned with fixed laser/exposure time settings. Fluorescence intensities were analysed in Fiji by determining the mean intensity per area of manually selected ROI.

For quantification of Crol::GFP levels midguts were stained with antibodies targeting the ISC/EB marker Arm[31] and the EE marker Pros[30] to unequivocally distinguish the different cell types. After imaging the nuclei of the different cell types were outlined manually and mean intensities per area were determined using Fiji. 30–56 cells per cell type and gut (indicated as 'N' in figure panels) in three to four different guts (indicated as 'n' in figure panels) were measured. We also measured the total area of R5 in each group and calculated a fold change in R5 area of the tested conditions compared to the corresponding control. We multiplied the Crol::GFP intensities by the calculated fold change in R5 area thereby normalizing the intensities to take into account mating induced area changes, which lead to a dilution of actual hormone dosage reaching a single cell in a mating adapted intestine. Crol::GFP levels for crol-RNAi validation (Fig. S2I–L) are not normalized. We then compared the Crol::GFP levels between different groups by statistical analyses.

The procedures we used to measure fz3-RFP intensities were the same as described for Crol::GFP intensities except for the normalization. Fz3-RFP intensities in nulcei of aArm⁺ ISC/EB were normalized to ISC/EB numbers. Therefore, we compared ISC/EB numbers of the tested groups and calculated the fold changes to the corresponding control. We multiplied measured fz3-RFP intensities with the calculated fold changes to compensate for dilution and degradation of Wg ligands reaching individual cells with increasing ISC/EB numbers as sink for Wg ligand.

For quantification of Wg::GFP levels midguts we also used immunostainings to unequivocally distinguish between the different cell types. After imaging we outlined 25–30 single EC per gut in R5 to define a ROI in which we measured mean intensities per area. Due to a low signal-to-noise ratio in the Wg::GFP signal we subtracted background signal from the measured Wg::GFP intensities. Therefore, we outlined ten areas per midgut within nuclei of EC where we would not

# Endocrine control of ISC proliferation balances gut size

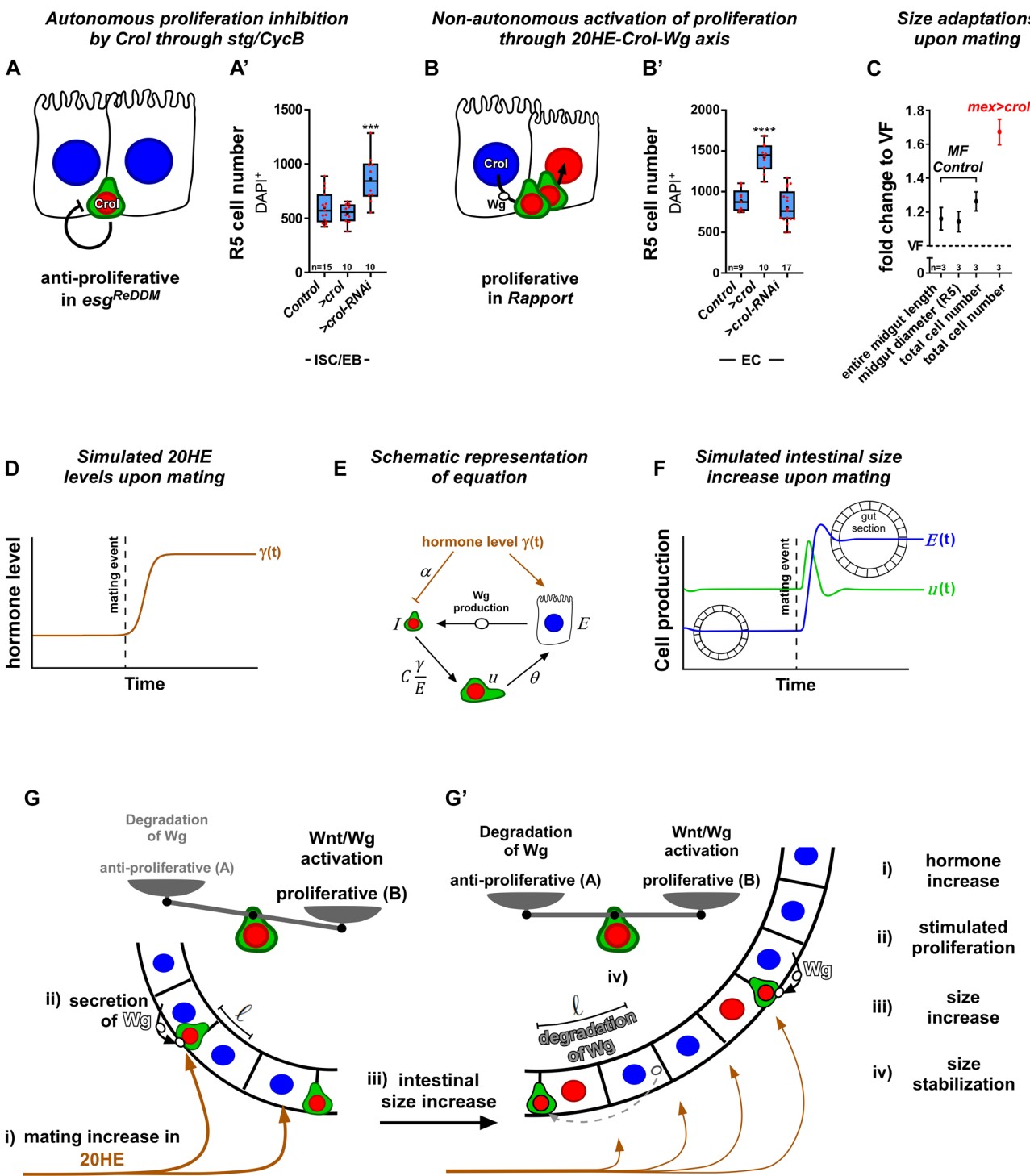

expect Wg ligands to localize. We then measured mean intensities per area within these defined ROI and calculated the average background signal for each gut. The calculated average background signal was subtracted from the individual Wg::GFP intensities we measured in EC. Resulting Wg::GFP levels of different groups were then compared by statistical analyses.

## Survival assays

CRC avatar flies were grown at 18 °C and shifted to 29 °C at latest 24 h after eclosion with at least ten flies per vial. Female and male flies were assayed with 34 to 118 flies for each genotype. Dead animals were counted once a day and statistical significance of survival differences determined using Kaplan-Meier log-rank survival test.

## Analysis of correlation of hZNF267 and Wnt/Wg signalling components in colon adenocarcinoma

Correlating expression of hZNF267 and Wnt/Wg signalling components in colon adenocarcinoma was analyzed using the R2 Genomics analysis and visualization platform. In a single dataset analysis using the Mixed Colon Adenocarcinoma (2022-v32) from tcga we searched

**Fig. 7 | Mathematical modelling of hormone level controlled intestinal size. A, B** Schematics of Crol inhibiting proliferation in ISC through stg/CycB and non-autonomously activating proliferation by Wg expression in EC. **A′–B′** Quantification of cell numbers in R5 regions upon (**A′**) ISC/EB specific ($p = 0.0003$) and (**B′**) EC specific manipulations of *crol* (F003414, BL41669). **A′–B′** For Box-and-whisker plots: the center is the median, minima and maxima are 25th and 75th quartile and whiskers indicate full range of values. All individual values with 'n' representing numbers of biological replicas are shown by dots and means are indicated by '+'. Asterisks denote significances from multiple comparisons by one-way ANOVA (***$p < 0.001$; ****$p < 0.0001$). **C** MF size adaptations, shown are means ± SD of fold changes in entire midgut length, R5 diameter and total cell number of MF and EC specific >*crol* (F003414) compared to VF after 3 d of Rapport tracing. Biological replicas are indicated by 'n' values. Source data are provided as a Source Data file. **D** 20HE levels upon mating as a function of time ($y(t)$). **E** Equation visualizing the divergent effect of 20HE hormone on ISC ($I$) inhibiting ISC proliferation and on EC ($E$) stimulating Wg production and thereby production of EB ($u$) with $C\frac{y}{E}$. EB ($u$) are an intermediate state between ISC and EC and differentiate into EC with a rate θ. **F** A graph visualizing intestinal size increase upon mating showing functions for production of intermediate EB ($u(t)$) and EC ($E(t)$) per time with a growing gut section following the mating event. **G** Gut section with indicated mating increase in 20HE (i) reaching EC thus inducing expression of Wg ligands (ii). Wg release stimulates ISC proliferation within a distance l and thereby intestinal size increase (iii). In a growing midgut the non-autonomous Wnt/Wg activation outweighs the degradation of Wg and autonomous proliferation control. **G′** Upon an increase in EC numbers the average distance l of EC (Wg source) and the ligand receiving ISC increases. With a longer distance l Wg ligands degrade leading to a decrease in Wg concentration sensed by ISC and thereby to a decline in ISC proliferation (iv) stabilizing gut size.

**Table 2 | List of primers used in real-time qPCR to investigate expression levels of *CycB* and *stg* upon oral RH5849 administration**

| Primer | Forward (5′–3′) | Reverse (5′–3′) |
|---|---|---|
| *rp49* | TGGTTTCCGGCAAGCTTCAA | TGTTGTCGATACCCTTGGGC |
| *CycB* | TTTGCAGAATCGCGGCATAAG | GTCTGTGAGCTTGAGATCCTTG |
| *stg* | GAAAACAACTGCAGCATGGATTGCA | CGACAGCTCCTCCTGGTC |

for KEGG pathways that correlate with hZNF267 expression. Within the collection of Wnt/Wg signalling pathway members single genes with significant positive or negative correlation were selected for visualization in a scheme.

### Alignment of Crol and hZNF267
The protein alignment of Crol and hZNF267 (Fig. S2M) has been performed with the protein sequence of Crol-PA deposited at Flybase, and the protein sequence of hZNF267 isoform 1 deposited at NCBI using the CLC Main Workbench software from QIAGEN.

### RNA isolation and cDNA synthesis
Freshly hatched female $w^{1118}$ flies were collected and kept on food containing 340 μM RH5849 together with male flies. After two days at 25 °C midguts of female flies were dissected for RNA-isolation and cDNA synthesis (Fig. 2J). Adult female flies with Rapport specific overexpression of *wg* were grown at 18 °C and shifted to 29 °C for three days prior to dissection and subsequent RNA isolation and cDNA synthesis (Fig. 5O), which were performed as described previously[85].

### Quantitative real-time PCR
Expression levels of *CycB* and *stg* in whole midguts of mated female flies were determined upon oral administration of RH5849 agonist (Fig. 2J) or expression of >*wg* using Rapport (Fig. 5O). Real-time qPCR was performed as described previously[85]. Relative expression levels were normalized to the house-keeping gene rp49 and calculated by ΔΔCt-Method. The primers used in real-time qPCRs are listed in Table 2.

### Statistical analyses
Statistical analyses were run in GraphPad Prism 9.0. Dot plots show all individual data points with no exclusions and indication of mean plus and minus the standard deviation. Box plots show the median and the first and third quartile with mean indicated by '+' and full range of values indicated by whiskers, additionally all individual values are shown by dots. No data were excluded. For statistical comparisons we first tested all data for normal distribution using the Shapiro-Wilk test[92] and then used either unpaired two-sided *t*-test[93] for comparison of two groups with normal distribution or One-way ANOVA[94] and the two-stage step-up method of Benjamini, Krieger and Yekutieli[95] for multiple comparisons of normally distributed data[96]. For data that did not show normal distribution we used non-parametric two-sided Mann Whitney *U* test[97] for comparison of two groups and Kruskal-Wallis test[98] following two-stage step-up method of Benjamini, Krieger and Yekutieli[95] for multiple comparisons[96]. Survival curves were analyzed using Kaplan-Meier log-rank tests[99,100]. Significant differences are displayed as * for $p \leq 0.05$, ** for $p \leq 0.01$, *** for $p \leq 0.001$ and **** for $p \leq 0.0001$. Results of all statistical tests can be found in the Source Data file.

### Reporting summary
Further information on research design is available in the Nature Portfolio Reporting Summary linked to this article.

## Data availability
All data generated in this study are provided in the Source Data file. The expression data of Wnt/Wg signalling components in mixed colon adenocarcinoma (2022-v32, tcga) used in this study (Fig. S7A) are available in the R2 Genomics analysis and visualization platform (https://hgserver1.amc.nl/cgi-bin/r2/main.cgi) with the R2 internal identifier: ps_avgpres_tcgacoadv32a512_gencode36. Source data are provided with this paper.

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

## Acknowledgements

The authors thank the Vienna Drosophila RNAi Center (VDRC) and the Bloomington Drosophila Stock Center (NIHP400OD018537) as well as Thomas Klein and Andreas Wodarz for reagents. We also thank the Center for Advanced Imaging (CAi) at Heinrich Heine University (DFG INST 208/539-1 FUGG) for imaging training and facilities. In addition, we are grateful to Reiff laboratory members for comments and feedback on experiments and the manuscript. TR and LZ are supported by the Deutsche Forschungsgemeinschaft (DFG-Sachbeihilfe RE 3453/2–1), the Wilhelm Sander-Stiftung (2018.145.1) and the Deutsche Krebshilfe (70115333). BCM wants to acknowledge the support of the field of excellence "Complexity of Life, Basic Research and Innovation" of the University of Graz.

## Author contributions

LZ, BCM, and TR designed the research. LZ conducted all biological experiments and BCM performed the mathematical modelling. LZ and TR analysed biological data, TR supervised the project, and all authors contributed to writing the manuscript.

## Funding

## Competing interests

The authors declare no competing interests.
