## [Transparent Peer Review file · Nature Communications]

Steroid hormone-induced wingless ligands tune female intestinal size in *Drosophila*

Corresponding Author: Dr Tobias Reiff

Version 0:

Reviewer comments:

Reviewer #1

(Remarks to the Author)

The intestine is a highly adaptive organ, constantly remodelled in response to environmental cues and stress. One such instance when the intestinal epithelium undergoes extensive transformation is during pregnancy to accommodate the increasing need for nutrient uptake. This requires an interorgan signalling cascade that emerge from reproductive units which then signal to the intestinal epithelium to remodel. Past research in *Drosophila* has shown that the steroid hormone Ecdysone, released upon mating from the ovaries, acts on intestinal stem cells (ISCs) to increase proliferation and consequently increase gut size. However, how these steroid signals are transduced to produce a proliferative response in the intestinal epithelium remains still a mystery. In this article, Zipper et al. explore the idea of Crooked legs (Crol) (a conserved transcription factor belonging to the zinc-finger family) transducing the Ecdysone-Ecdysone receptor (20HE-EcR) signalling pathway in both cell-autonomous and non-autonomous manner during mating-dependent regenerative response. Interestingly, the authors have uncovered that the Crol signalling pathway constitutes a feedback loop determining the final organ size in which activation of Crol in ISCs is anti-proliferative whereas Crol activity in enterocytes (ECs), through secretion of Wingless, induces proliferative response in neighbouring ISCs. The authors have combined the experimental data with theoretical modelling to support this idea. In my opinion, the article can be considered for publication in Nature Communications, after addressing some key questions left unaddressed in this current version.

Major comments:

1) In Figure 1, the micrographs shown for Crol localization in nuclei do not correspond with the quantifications of the fluorescence intensity. For example, according to the quantification, there is more than 50% increase in Crol fluorescence intensity after mating compared to that midgut cells from virgin flies. But on the confocal micrographs this change is not obvious.

According to the authors, the activation of Crol expression is driven by the 20HE-EcR signaling. It would be interesting to have mosaic guts where heat-shock flipped clones are induced either with a knockdown of EcR or gain of function of EcR and quantify if the change in fluorescence intensity for Crol still occurs between the 2 populations of cells.

The authors could also use the reporter for EcR activity and perform a correlation analysis between the reporter activity and the fluorescence intensity of Crol localized in the nuclei. A positive correlation indicates that Crol activation depends on activation of 20HE-EcR pathway.

I would have expected ovoD1 mutants to have less 20HE since they are sterile and have smaller ovaries. Could the authors validate this tool by measuring ecdysone levels using ELISA kit (Bertin Bioreagent).

2) It is not clear to me how 20HE-Crol signalling works in ISCs as antiproliferative signal. According to the model that the authors suggest, Crol inhibits Stg. But this inhibition is not at the level of transcription as the mRNAs of both stg and cycB go up during RH5849 feeding. Also, knocking down a cell cycle complex together with crol will indeed block the proliferative response that happens after crol knockdown alone. This cannot be used to support the idea that it is an epistatic interaction.

3) From literature, it is known that the entire posterior region of the female midgut grows in response to mating. It is not clear why the authors chose only R5 region in this study? Is Crol response to 20HE signalling and function restricted to R5 region? Does modulating Crol levels in ISCs/ECs also affect the mating-dependent proliferation and differentiation in R4 region of the midgut? Could the authors include the length of guts and diameter in this measurement?

4) The authors find that Wg is produced from midgut-hindgut boundary (MHB) and predicts that it acts like a morphogen. If so, can it account for the remodelling of entire posterior midgut in response to mating in female flies? If one looks at the proliferation pattern, is there a crescendo similar to the fz3-reporter observed also for the pH3 signal, peak at the MHB boundary and waning as the distance increases from MHB?

5) Figure 5 micrographs are a bit puzzling. If the Rapport constructs work as the authors claim, there shouldn't be any GFP+ cells that are not RFP+. But in these micrographs, there are many GFP+ cells that are negative for RFP, especially in panes C and F. This could be misleading as the population of ISCs/EBs is measured as total number of GFP+RFP+ cells.

Minor comments:

- In general, the authors have used student T-tests for statistical analysis. I would suggest them to use one-way ANOVA wherever there is comparison between more than two groups.
 - The authors claim that Crol has a cell autonomous function in ISCs as an antiproliferative agent by inhibiting Stg. The dual knockdown of stg and crol is in line with this, however the dual overexpression of Stg and Crol sheds light also on to a cell autonomous role of Crol in regulating cell differentiation. Overexpressing Stg in progenitors expands the ISC/EB pool together with increase in EC/EE number whereas Crol overexpression had an opposite effect. Overexpressing both Crol and Stg brought the ISC/EB number even lower than the control whereas a larger number of newly produced mature cells are visible, indicating an accelerated maturation of progenitors. mRNA expression of both stg and cycB increases upon EcR signalling, suggesting that stg/cycB is not inhibited transcriptionally by Crol. Dual expression of Crol and Stg also supports this idea because Stg OE under UAS/Gal4 bypasses endogenous regulation, hence proving that Crol induces faster maturation of progenitors to EC fate. For me, it is not clear if the antimitotic effect of Crol in ISCs is due to faster maturation to ECs or blockade of cell cycle exit through Stg/CycB. A discussion regarding how Crol cell autonomously regulates Stg/CycB in ISCs would be appreciated.
 - In Figure 3, the authors could label some landmarks and point to some cells to make it clear for the readers to understand what they are looking at especially the micrographs for Wg::GFP. Also, there is a disparity between Wg signal and fz3 reporter activity after RH5849 feeding. The quantification for Wg::GFP after feeding RH5849 shows only a 3% increase whereas fz3 reporter shows a 58% percent increase. How is it possible? Is it something to do with the way the authors have measured the intensities? Also, I am a bit lost at such a high number of sample values for Wg measurements? Do these high n values correspond to pixel values? If so, a better way would be to measure the mean from selected ROI as values from individual pixels can be very noisy.
 - In Figure 1, the fold-change after feeding RH5849 (>10%) is much lower than mating-dependent response (>50%)? Is it because the titre of RH5849 that the cells experience at the used concentration is much lesser than endogenous 20HE produced after mating? Have you tried feeding the flies 20-HE (Sigma Aldrich H5142) to see if this can bring in a relative change similar to that of during mating?
 - Is Crol expressed in midgut epithelium of male flies and is it sensitive to 20HE feeding?
 - Crol seems to be expressed ubiquitously. The authors have tried to explore the role of Crol in different cell types except enteroendocrine (EECs) lineage. Since Crol plays a crucial role in cell differentiation and loss of Crol in ECs non-cell autonomously affects ISC proliferation, could the authors try to knockdown Crol in EECs and report the non-cell autonomous effects, if any.
 - The model based on cell autonomous inhibition of proliferation by Crol in ISCs and mitogenic effect of Crol in ECs is interesting, but it is too qualitative and not sure how relevant it is in gut physiology is not clear. It assumes that ISCs are distributed uniformly and this distance to ECs increases during organ growth. This might be true in case of homeostasis, but regenerative responses might induce a bout of cell divisions that alter this pattern of ISC distribution. Even though the total number of ISCs stays before and after mating-induced proliferative response, authors should also quantify ISC distribution before and after mating. This model holds true only if the distribution shows that the ISCs are uniformly distributed even after mating.
- One important component missing in this current model is time. The model is constructed as if there is one to one relation between ISC mitosis and an EC production. Authors did not take into consideration how different is the time for a proliferative cycle of an ISC versus the time taken for EBs to transform to mature ECs. If ISCs can finish 3 to 4 cycles of division before a new EC is produced, I am not sure how this model can hold true. Are these dynamics known during homeostasis and mating-dependent regeneration. I would bring a temporal information into the model.
- Another event where time matters is the signalling itself. 20HE has a direct effect on Crol in ISCs as an antiproliferative signal whereas mitogenic effect of 20HE from ECs is indirect. 20HE increases Crol in ECs which in turn increases Wg and its secretion, which then act in ISCs as the mitogenic signal. Since both ISCs and ECs on the same epithelium are exposed to the same levels of circulating 20HE. We can speculate that the action of antiproliferative signal should be faster than the mitogenic signal. How will such a system result in an increased organ size unless there is a thresholding or a cell-type specific regulation of 20HE-Crol activity? For the current model to be exact, ECs should respond to 20HE faster than ISC indicating that is ECs should be more sensitive to circulating 20HE than ISCs.
- In this model, the authors consider that as the tissue proliferates, the distance from the Wg source to the ISC increases. But what is not clear to me is that even when the tissue is in a proliferative state, the new mature cells that are produced are still ECs and they are still responsive to circulating 20HE, which means there is no change in effective distance between Wg source and ISCs. How this parameter is included in the model.
- In figure 2L, one of the bars corresponding to a p-value is shifted.
 - In Figure 3, S4, and S5, if the LUT for fz3-RFP is grey, it would be easier to recognise the signal.
 - Could the authors include more a picture of tumours than a scheme in the Fig 6K?
 - Have the authors tried to knock down Wg only in the MHB other than using a pan-EC driver mex?
 - In line 222, regarding fly survival, the legend is mislabelled as Fig S6L but it should be Fig 6L.

- In line 223, regarding gut length, the legend is mislabelled as Fig 6M but it should be Fig S6D.
- In line 225, regarding multilayering, the legend is mislabelled as Fig S7D but it should be Fig 6M.

Reviewer #2

(Remarks to the Author)

The study by Zipper and colleagues highlights a zinc finger transcription factor Crooked legs (Crol) as a key coordinator of cell behavior and organ size adaptation in the adult female *Drosophila* intestine in response to changes in the steroid hormone ecdysone induced by mating. The authors developed a Rapport ('Repressible activity paracrine reporter') system that allows precise spatiotemporal control of transgene expression while performing stem cell lineage tracing. Using the Rapport, they demonstrated the differential effect of *crol* and its putative human ortholog ZNF267 on intestinal stem cells (ISCs) and enterocytes (ECs). Their results show that an excess of Crol in ISCs inhibits their proliferation by suppressing String (Stg)/CDC25 and Cyclin-B (CycB) through an unknown mechanism. Conversely, upregulation of Crol in enterocytes (ECs) stimulated ISC division in a paracrine manner by increasing the production of Wingless (Wg). Importantly, they underscored the importance of finely regulating Crol levels in different intestinal cells, extending beyond physiological functions, by manipulating Crol in cancer stem cells and the tumor microenvironment using two different intestinal fly tumor models. This is an exciting study, that uses state-of-the-art *Drosophila* genetics to provide novel mechanistic insights into organ size regulation in response to endocrine inputs by exploiting distinct cell type-specific functions of a single transcription factor. The results and developed tools would be of broad interest to the readership and scientific community. Unfortunately, several conclusions are not well supported by the data presented. The tumor data appear sketchy and incomplete.

Major Comments

The upregulation of Crol in response to changes in ecdysone levels post-mating in female fly intestines is the primary discovery of the manuscript. I find it difficult to reconcile the representative images in Figure 1A-C with the quantifications in Figure 1E. It is unclear whether the quantifications of Crol::GFP fluorescence were performed on guts stained with antibodies that distinguish ISC/EBs from EE and ECs. The nuclei marked as ISC/EB in Figure 1A-C seem larger and the distinction isn't as pronounced compared to Figure S1B-C.

The quantifications throughout the manuscript look impressive with remarkably high power. Unfortunately, the choice of a T-test is often inappropriate. It is recommended to utilize ANOVA with multiple comparisons for most data. Moreover, considering the negative binomial distribution of numerical/count data, employing non-parametric tests like Kruskal-Wallis and Dunn's test is advisable.

Reusing of data in Figure 4B, C, D and the claim of significance using a t-test are incorrect.

The authors' statement that the Wg::GFP signal is increased in enterocytes at the midgut-hindgut boundary lacks sufficient support in the provided data. To validate this claim, it would be beneficial to include images combining Wg::GFP with staining highlighting various midgut cells. Additionally, clarification on the specific region of the midgut displayed in the high magnification image (Figure S4A) is needed. Did author use any normalization when quantifying the Wg::GFP signal? Can they comment on a clear difference in Wg::GFP signal between control and MeOH treated intestine?

How do the authors reconcile the spatially restricted induction of Wg::GFP at the midgut-hindgut boundary with the increased levels of 20HE in the hemolymph circulating throughout the body? Do the authors have evidence for differential uptake of the 20HE agonist RH5849 along the adult intestine?

The authors state that "we detected fz3-RFP signal in intestinal precursors along the midgut (Fig. 3E)" (line 152).

Unfortunately, I find no support for this statement in Figure 3E. From the images provided, it is impossible to determine where the entire midgut might be. The authors need to provide confocal micrographs of fz3-RFP midguts where individual cell types can be clearly distinguished based on nuclear staining combined with immunostaining with antibodies against e.g. Arm, Dlg1, Pros. Please note that the current micrographs are of insufficient quality. Providing grayscale variants is highly recommended to highlight potential differences between genotypes. As presented, I find it difficult to reconcile micrographs showing fz3-RFP signal with the quantifications and significance presented in the corresponding graphs.

Clarification is needed regarding the evidence supporting human ZNF267 as the functional orthologue of fly *crol*. Reviewing Flybase, Diopt, etc., other ZNFs such as ZNF606, ZNF850 are listed. The authors need to provide additional evidence and analysis to justify their focus on human ZNF267. The authors show that hZNF267 OE attenuates the mitogenic effect of *esgReDDM>crol-RNAi*. Can the authors rule out that the "rescue" effect of hZNF267OE is a general response to ZNF excess? Would ZNF606 or ZNF850 have the same effect? Can they rule out a saturation effect of the system with the two UAS constructs? Does +hZNF267OE in wild-type background recapitulate the effect of *crol*OE?

Although potentially very interesting, the analysis of *crol*/ZNF267 in the two different intestinal tumor models seems incomplete. The results suggest a differential effect of the paracrine *crol*/ZNF267 on ISCs/EBs. Furthermore, the authors fail to describe and discuss the *crol*-RNAi data. How do the authors explain the limited to no effect of *crol*-RNAi in Notch-RNAi tumors? It appears that EE accumulation is prevented when *crol*-RNAi is expressed in Notch-deficient ISCs. In contrast, *crol*-RNAi does not exert anti-tumor effect when expressed non-autonomously in ECs? How relevant is the paracrine *crol*-Wg axis in the two different tumor models?

Quantification of data: There's ambiguity regarding what "n" represents in certain figures. For example, in Figure 3I or Figure 4B, the authors seem to be referring to the number of cells counted. If this is the case, they need to specify the number of intestines/animals they analyzed and include in the methods how many independent experiments the samples are from.

Minor comments

In the Materials and Methods, line 26, the authors likely mean genomic DNA rather than cDNA as the target of gRNAs.

The authors should discuss how increased Wg-GFP in MHB affects fz3-RFP levels along the entire midgut. One might expect a gradient of fz3-RFP induction. Can the authors provide any evidence?

Would crol-GFP also be induced in male intestines fed with the 20E agonist RH5849? Is the 20HE-Crol-Wg axis sex specific?

The authors may want to provide a more detailed explanation of the CRC model to allow a broader audience to appreciate their findings.

Consistency in writing genotypes in tables and over confocal micrographs. Authors may consider listing all transgenes for individual panels to be self-explanatory and consistent throughout the manuscript. For example, writing +hZNF267 (Figure 2E) or +crol-RNAi (Figure 2J) may not be properly understood.

Why do the control tumors in Figure 6A and 6F look so different?

Figure 6L - The viability of the control line used for CRC tumor induction should be shown, ideally with pCFD6_noSapI transgene expressing mock gRNAs.

Figure 5C - Can the authors comment on why most cells are RFP negative after EclOE?

General comments

Considering that colorblind individuals may have difficulty distinguishing between red and green colors, using a colorblind friendly palette is recommended.

Reviewer #3

(Remarks to the Author)

The manuscript by Zipper et al. tackles the question how intestinal size is regulated upon mating in *Drosophila*. They identify a transcription factor Crol which is regulated by the ecdysone hormone followed by mating. They find an interesting mode of regulation where Crol regulate ISC proliferation in an opposing manner depending in which cell type it is expressed. In ISCs, Crol inhibit ISC divisions, whereas in ECs, Crol induce ISC proliferation in a non-cell autonomous manner. In addition, Crol manipulation in intestinal tumor model has a profound effect on tumor size and fly survival. The authors suggest mechanisms for the cell autonomous and non-cell autonomous function of Crol. In ISCs, it is suggested that Crol negatively regulate String and block ISC cell cycle progression. In ECs, it is suggested that Crol regulate Wnt secretion and thus the proliferation of the nearby ISCs. Based on their data, the authors conclude that Crol regulate physiological intestine size regulation upon mating downstream of ecdysone and upstream of Wnt signaling to increase intestinal stem cell proliferation. Finally, the authors provide a mathematical model to support their conclusions.

While the effect of Crol in regulating ISC proliferation in wild type and tumor model is clear and interesting, the authors do not provide sufficient data to support the conclusion that Crol regulate intestinal size as a response to mating through Wnt. While these are the main conclusions of the manuscript, additional evidence is needed, or the conclusions and manuscript need to be significantly readjusted.

Key conclusions which are not supported/not shown by the provided data:

1. Crol regulates the physiological intestine size adaptation upon mating.
2. Crol regulates Wnt secretion from ECs.

In addition, there are serious problems with data analysis, particularly with the statistical tests used. This hinders my evaluation of the data. The authors incorrectly use t-test for multiple group testing, do not provide information what type of t-test is used, or use it with data which is not normally distributed or with non-parametric data. These issues should be corrected.

Please find more specific comments/questions below.

Fig1

How are the ISC/EB cells vs ECs identified? If only nuclei size is used as a criterion, more appropriate wording is small nuclei cells/large nuclei cells.

E: please provide the N guts and N cells in the figure panel instead of the legend (refers to all figure panels). The data in VF samples seems not to be normally distributed, hence the use of t-test might not be appropriate here. Please test for normality and use a proper statistical test (applies to all statistical tests).

Please provide more information about the intensity measurements (applies to all figures):

In the materials and methods it is stated as following: "Fluorescence intensities were analysed in Fiji by determining the

mean intensity per area of manually selected nuclei. This way, *cro1::GFP* and *fz3-RFP* expression levels were measured and normalized to number of ISC/EB or area of R5 respectively." What does it mean "normalized to number of ISC/EB or area of R5 respectively"?

Are the intensity measurements background subtracted or not?

Fig2

Based on epistatic analysis, the authors make the conclusion that *Crol* directly inhibits *stg* to control ISC proliferation (Fig2 M). To support this conclusion more data is needed (to show *Crol* regulates *stg* transcription), or the conclusion needs to be rephrased to allow indirect regulation.

E, K, L: Student's t-test is not appropriate to test multiple samples. Please provide ANOVA to compare the means between groups (if normally distributed data). This refers to all figures in the manuscript where t-test has been used to test multiple groups. Please show the individual data points in the plots and provide N in the figure panels (refers to all box plots in the manuscript).

F: What statistical method was used?

G-J: Please provide the control image for these experiments. Please provide the genotype in the Figure panels (instead of + or >*cro1* etc., it is difficult for the reader to follow what manipulation/genotype is used) (this applies to all figure panels).

Please show the sequence homology between *Crol* and ZNF267 (can be in the supplement).

Fig3

A-D: The authors conclude *WG::GFP* signal is seen in the EC cells (row 147-148), however it is not possible to make this conclusion from the images provided. Please provide DAPI channel as well as A-P direction to the images and indicate the mid/hindgut boundary. Is *WG::GFP* also increased in other region boundaries or is it specific to mid/hindgut boundary?

E-H: The authors state they have quantified *fz3-RFP* signal from ISC/EB cells, but from the images it is impossible to say what cells are shown as RFP positive. Proper markers for ISC/EB should be provided. From which region are the images and quantifications from?

I-J: How is the quantification of *WG::GFP* intensity made? Are there more *WG::GFP* positive cells in RH5849 fed flies or *ovoD1* flies? The +3% increase in the mean *Wg::GFP* intensity signal is very modest in I. Do the authors consider this to be biologically significant? What type of t-test is used to calculate the p value in I?

K The authors make the conclusion that *Crol* regulate *Wg* secretion from the ECs but no data is provided to support this conclusion.

Fig4

This is the only image providing data about *Crol* regulating *wnt* pathway in ECs. More evidence is needed to support this conclusion.

B-D Please provide representative confocal microscope images showing the quantified effects (now in the supplement but should be in the main figure). I am puzzled by this quantification since when looking at the representative images provided in Fig5 J-R I cannot see the measured differences. For example, in J vs K I could not see an increase in the *fz3-RFP* signal. In the current images, the signal-to-noise ratio of the *fz3-RFP* is very low and does not allow the reader to make conclusions about differences. Please provide more information how the quantification was performed? From which region was it done and how were the cells selected for measurement? Is the background signal considered in the measurements? Also, there is no ISC marker provided? How were those cells identified as ISCs?

Fig5

A The authors have developed a genetic tool, *rapport-tracing*, enabling transgene expression in the EC and simultaneous tracing of progenitors through *ReDDM*. This is a nice tool and is of interest to many in the field.

B-K By using their *Rapport-tracing* method the authors make the conclusion that *Crol* controls *Wnt/wg* secretion from the ECs which non-autonomously activate ISC proliferation. While their results show that manipulating *Crol* levels in the ECs do regulate ISC proliferation, there is no evidence to support *Crol* works through *Wnt*. The very least, a genetic epistasis experiment with *Wnt* pathway and *Crol* is required to make this conclusion. In addition, some evidence is needed to show that *Crol* regulates genes involved in *Wnt* secretion in the ECs.

Fig7

A-B If the authors wish to make conclusions about *Crol* regulating intestinal size, then intestinal size should be measured. Intestinal tissue size regulation is a complex process involving cell proliferation, differentiation, cell death and cell size regulation. These processes are regulated in a region-specific manner. Thus, counting cell numbers from R5 region does

not equal to intestinal size. Alternatively, the conclusions about intestinal size needs to be rephrased.

What additional value does the mathematical modelling bring to the field? How can it be used to predict intestinal adaptation? Could these topics be considered in the discussion?

Supplementary figures

FigS1

Row 485: “dsRed positive EC” – should be “dsRed positive EB” as EB marker is used here.
Please use the more common abbreviation “ α ” for antibody instead of “a” (refers to all figures).

FigS2

I: If control value is 0, it does not make sense to apply statistics on that.
J: Please provide quantification for this important data about RNAi validation.

FigS3

A: again, testing against zero is not suggested.

FigS4

B-E I do not see an increase in wg::GFP or fz3::RFP in the images of MF sample compared to the VF sample.
Is the intensity of wg::GFP really increased or are there just more cells expressing wg? How are the ISC/EB cells identified in the quantification of fz3-RFP?

Reviewer #4

(Remarks to the Author)

Version 1:

Reviewer comments:

Reviewer #1

(Remarks to the Author)

The authors have addressed my concerns and the manuscript seems ready for publication

Reviewer #2

(Remarks to the Author)

The authors have provided additional data and revised their statistical analyses to strengthen their working model. In their rebuttal letter, they provide a thorough and detailed response to the reviewers' individual comments.

However, my main concern with this study remains and relates to the reproducibility of the data and the biological significance of the findings. The authors base their conclusions and model on a limited number of biological replicates. Although they quantify signal intensities in a large number of cells, these cells are often derived from only three to four adult *Drosophila* guts. Given the inherent and well-documented variability among adult flies, coupled with the fact that the effect of RH5849 requires feeding (NOTE: administration of RH5849 has been shown to reduce food intake by approximately 75%, as reported by Landis et al., 2022, Fly, <https://doi.org/10.1080/19336934.2022.2149209>), I find the number of biological replicates insufficient to support the conclusions drawn in this study.

This concern is particularly relevant for the quantification of signal intensities, such as Crol::GFP, wg::GFP, and fz3-RFP in the cells of the R5 region, as well as comparisons between virgin/mated female flies, control/ovoD1, and MeOH/RH5849 treatments. One of the major and appreciated advantages of using the *Drosophila* model is the ability to generate robust, reproducible data from a large number of biological replicates in independent, temporally-separated experiments. Given that the differences observed are modest, less than 20%, I would expect that the experiment should have been blinded to avoid bias in the interpretation.

Additionally, in the authors' response to reviewer comment 24 and in the revised Materials and Methods section, they state that fluorescence signal intensities were normalized to mating-induced area changes. However, they do not provide any evidence to support the claims of “dilution of actual hormone dosage” or “dilution and degradation of Wg ligands.” For this normalization approach to be valid, the authors need to present unequivocal data demonstrating these dilution effects.

The functional genetic and epistasis experiments are the stronger part of the study. At the same time, I find it interesting to see the same controls in response 20 and Figure 2K. How many times have these experiments been repeated. Are the data all coming from one experimental trial?

Reviewer #3

(Remarks to the Author)

The authors have adequately addressed all of my concerns, and I find this manuscript to fulfill the high standards required for publication in Nature Communications.

Congratulations for the authors for their nice study.

Reviewer #4

(Remarks to the Author)

Version 2:

Reviewer comments:

Reviewer #2

(Remarks to the Author)

The authors provided satisfactory answers to my questions. The repetition of the experiments and the addition of the data to the manuscript strengthen their conclusions. It is very encouraging to see the reproducibility of the phenotypes in the authors' hands. As such, the study provides a solid and exciting foundation for future follow-up studies.

I would like to invite the authors to review the attached PDF, which outlines the benefits of using a color-blind-friendly palette.

General response to all reviewers:

Dear reviewers,

we thank all reviewers for their constructive suggestions. To address the points raised by the reviewers, we have conducted a series of supporting experiments strengthening and expanding our data from the initial first submission. The new data consolidates our biological findings and mathematical model

A common aspect from the initial review concerned the statistical analyses of our datasets. Combining and following the suggestions from the reviews, we now performed and added requested normality tests on our raw data and performed the suggested statistical analyses. For transparency, we now included all analyses in our raw data sheets. Importantly, the requested additional statistical tests support our previous conclusions and new data consolidates and strengthens the original biological data and mathematical model. Thanks to the less restrictive space limitations compared to NCB, we added explanatory text, supporting literature and expanded our discussion throughout the manuscript based on the reviewer's suggestions.

Together, our revised manuscript confirms, extends and details on our findings presented in the original submission. Please find below a point-by-point response to each concern raised by the reviewers. We took the liberty to consecutively number the comments and refer to prior comments for similar concerns.

REVIEWER COMMENTS

Reviewer #1 (Remarks to the Author):

The intestine is a highly adaptive organ, constantly remodelled in response to environmental cues and stress. One such instance when the intestinal epithelium undergoes extensive transformation is during pregnancy to accommodate the increasing need for nutrient uptake. This requires an interorgan signalling cascade that emerge from reproductive units which then signal to the intestinal epithelium to remodel. Past research in *Drosophila* has shown that the steroid hormone Ecdysone, released upon mating from the ovaries, acts on intestinal stem cells (ISCs) to increase proliferation and consequently increase gut size. However, how these steroid signals are transduced to produce a proliferative response in the intestinal epithelium remains still a mystery. In this article, Zipper et al. explore the idea of Crooked legs (Crol) (a conserved transcription factor belonging to the zinc-finger family) transducing the Ecdysone-Ecdysone receptor (20HE-EcR) signalling pathway in both cell-autonomous and non-autonomous manner during mating-dependent regenerative response. Interestingly, the authors have uncovered that the Crol signalling pathway constitutes a feedback loop determining the final organ size in which activation of Crol in ISCs is anti-proliferative whereas Crol activity in enterocytes (ECs), through secretion of Wingless, induces proliferative response in neighbouring ISCs. The authors have combined the experimental data with theoretical modelling to support this idea. In my opinion, the article can be considered for publication in *Nature Communications*, after addressing some key questions left unaddressed in this current version.

We thank the reviewer for the positive evaluation of our work and appreciate the constructive comments and suggestions that helped improving our manuscript.

Major comments:

1) In Figure 1, the micrographs shown for Crol localization in nuclei do not correspond with the quantifications of the fluorescence intensity. For example, according to the quantification, there is more than 50% increase in Crol fluorescence intensity after mating compared to that midgut cells from virgin flies. But on the confocal micrographs this change is not obvious.

We agree with the reviewer and now include new representative images (also in grey scale for better visibility) of actual fluorescence intensity levels as requested (Fig.1A-D). Please keep in mind that a substantial number of individual cells (n=120) was measured in four biological replicas. Biological and technical replicas are now indicated in quantifications of all figures (see comment #6 for details). Our new Figure 1 also includes the requested merged micrographs showing the aArm and aPros staining that were used for unequivocal identification of different cell types (Fig.1A'-D').

According to the authors, the activation of Crol expression is driven by the 20HE-EcR signaling. It would be interesting to have mosaic guts where heat-shock flipped clones are induced either with a knockdown of EcR or gain of function of Ecl and quantify if the change in fluorescence intensity for Crol still occurs between the 2 populations of cells.

We thank the reviewer for suggesting this experiment, which shows that Crol::GFP levels indeed depend on 20HE uptake by Ecl in mosaic guts. As suggested, we performed Flp-out clones induced by heat-shock driven flippase which are then positively labelled by UAS-RFP and combined with Crol::GFP and UAS-Ecl or UAS-Ecl-RNAi. Measurement of Crol::GFP levels in ISC/EB and EC outside and inside clonal areas showed that Crol levels reciprocally depend on 20HE uptake by Ecl. This important data is now included as Fig.1 (G-J) and described in the revised manuscript (lines 74-81).

The authors could also use the reporter for EcR activity and perform a correlation analysis between the reporter activity and the fluorescence intensity of Crol localized in the nuclei. A positive correlation indicates that Crol activation depends on activation of 20HE-EcR pathway.

Previous studies of fly development investigated a role for Crol in wing imaginal discs and could already show that crol levels depend on 20HE¹. Additionally, previous papers and work from our lab already established that mating as well as the EcR agonist RH5849 induces mRNA levels of ecdysone responsible elements²⁻⁴ as well as the early EcR target gene Eip75B-A/C⁴⁻⁶.

Our present data extends Crol as effector of EcR signalling showing Crol dependency with mating, genetic and pharmacological manipulation of Ecdysone signalling (Fig.1, Fig.S1D-F). These findings are now additionally supported by the flp-out clone experiment (Fig.1G-J) with genetically increased levels of 20HE activating the EcR by Ecl uptake⁷ leading to Crol::GFP increases (lines 74-81).

I would have expected ovoD1 mutants to have less 20HE since they are sterile and have smaller ovaries. Could the authors validate this tool by measuring ecdysone levels using ELISA kit (Bertin Bioreagent).

We previously measured 20HE levels in haemolymph as well as ovaries of ovo^{D1} mutant female flies compared to controls using abovementioned ELISA kit⁴. Briefly and already discussed in Zipper et al., 2020 and its peer review, we detected a mating induced increase of 20HE levels in ovaries as well as haemolymph of ovo^{D1} mutant flies⁴. In conclusion, in ovo^{D1} mutant flies, later egg stages of vitellogenesis are 'abolished as a sink for 20HE', which was the phrase we

included in the original manuscript. We now include and extend on this finding more in the main text (lines 66-69).

For a detailed discussion of this issue please see the publication and peer review process of Zipper et al., 2020 (<https://elifesciences.org/articles/55795/peer-reviews#content>).

2) It is not clear to me how 20HE-Crol signalling works in ISCs as antiproliferative signal. According to the model that the authors suggest, Crol inhibits Stg. But this inhibition is not at the level of transcription as the mRNAs of both stg and cycB go up during RH5849 feeding. Also, knocking down a cell cycle complex together with crol will indeed block the proliferative response that happens after crol knockdown alone. This cannot be used to support the idea that it is an epistatic interaction.

The reviewer raises an important point here that we now address and discuss more exhaustively in the main text. Concerning the autonomous proliferative response upon crol knockdown in ISC/EB, additional depletion of stg abolishes ISC proliferation (Fig.2L) and ISC lineage production (Fig.2K). A dependency of Crol LOF induced proliferation on Stg is additionally supported by studies of wing disc development, where proliferation phenotypes caused by heightened Crol levels are sensitive to stg and CycB heterozygous mutant backgrounds and stg transcription is regulated by Crol shown by stg-LacZ levels⁸.

In the course of this reviewing process, we also tried to address stg and CycB protein levels in the adult female midgut to complement qPCR data (Fig.2J) and enable analysis on a single cell level and post-transcriptional effects. Therefore, we checked two yet uncharacterized reporter stocks for CycB::GFP (A-B', BL84952) as well as stg::GFP (C-D', BL50879) from the Bloomington Stock Center. Unfortunately, we were not able to detect a reliable signal in the midgut (A-D), even not upon feeding of RH5849 or by crossing with the ovo^{D1} mutant allele (B-B' and D-D').

In concordance with our model, we propose that the induction of stg and CycB mRNA levels (Fig.2J) in ISC is caused by non-autonomous induction through Wg from EC by RH5849 (Fig.3A-B',I) initially outweighing the anti-proliferative role of Crol in ISC (that also depends on stg and CycB, Fig.2F-I,K-L, Fig.S3A-G). Indeed, such induction of Stg by Wg is further supported by previous findings⁸ and thanks to this suggestion, we can now show that CycB mRNA levels are increased upon EC specific expression of wg (E, Fig.5O). Notably this induction is equally strong as RH5849 stimulation of CycB (Fig.2J) where the exact cause of CycB induction upon mating is unclear and probably also stems from the 20HE-Crol-Wg axis. We present this new data in the text now (lines 219-220).

3) From literature, it is known that the entire posterior region of the female midgut grows in response to mating. It is not clear why the authors chose only R5 region in this study? Is Crol response to 20HE signalling and function restricted to R5 region? Does modulating Crol levels in ISCs/ECs also affect the mating-dependent proliferation and differentiation in R4 region of the midgut? Could the authors include the length of guts and diameter in this measurement?

Indeed, we focussed our analyses on the R5 region of female flies that our work identified in previous studies to robustly react to mating^{4,9}. An additional advantage is that R5 is unequivocally identifiable using the midgut-hindgut boundary (MHB) as landmark and thus ensures the analysis of the exact same region in all conditions. Furthermore, it was already shown that mating-dependent activation of sterol regulatory element-binding proteins (SREBPs) and transcripts involved in fatty acid synthesis and activation was highest in EC of the R5 region⁹.

*Following the reviewer's comment, we investigated effects of Crol in ISC and EC in R4. ISC/EB autonomous overexpression of Crol using the *esg^{ReDDM}* system is blocking ISC lineage production (B), whereas knockdown of Crol induces turnover also in R4 of adult female midguts (C). R4 is known for higher turnover rates compared to R5 and is completely renewed within four days¹⁰. The observed phenocopy of block of ISC proliferation in R4 (B) further underlines Crol's strong antiproliferative role and suggests Crol function might be conserved all along the midgut. In addition, phenotypes upon EC-specific, non-autonomous manipulation of Crol using *Rapport* are consistent in R4 as well. EC-specific manipulation of Crol using *Rapport* induces ISC/EB production in R4 (E), whereas *crol*-RNAi non-autonomously reduces lineage production (F) compared to controls (D).*

*Concerning the length and diameter of midguts, we now include the requested gut length and diameter measurements in Fig.7C and Fig.S7F to further support initial cell number analysis (Fig.7A',B'). Comparing VF and MF we also include total midgut cell number of three individuals (Fig.S7F) and present it as fold changes of MF midgut length, R5 diameters and total cell numbers compared to VF (Fig.7C). Importantly, by analysing the same parameters using EC specific expression of Crol (*Mex>crol*), we now show that increases in total cell number are not*

restricted to R5 but also detectable along the whole midgut (Fig.7C, >1.6 fold). This new data is now described in the main text (lines 288-290).

For detailed information and discussion regarding the correlation between midgut length and cell number please check comment #39 to reviewer 3.

4) The authors find that Wg is produced from midgut-hindgut boundary (MHB) and predicts that it acts like a morphogen. If so, can it account for the remodelling of entire posterior midgut in response to mating in female flies? If one looks at the proliferation pattern, is there a crescendo similar to the fz3-reporter observed also for the pH3 signal, peak at the MHB boundary and waning as the distance increases from MHB?

Indeed, the gradient of Wg activity as well as the fz3-gradient from MHB was described previously¹¹⁻¹³ cited in our initial manuscript. In the initial submission, we chose the MHB region because its established in the field as an area of active Wg/fz3 signalling^{11,12} and our observed Wg::GFP levels at the MHB reacted strongly to 20HE manipulations. Following reviewers suggestions, we now show increased Wg::GFP levels in EC of R5 anterior to the MHB/Malpighian tubules instead as described in the Materials and Methods. As shown at the MHB, Wg::GFP level in EC of R5 react to 20HE induction by mating, feeding RH5849 and genetic ablation of ovaries (Fig.3A-D',I-J, Fig.S4A-C). In line with this, we now show an increase in Wg::GFP levels upon EC specific overexpression of crol in R5, whereas Wg::GFP levels are lower upon crol-RNAi. Data are added to the revised text (lines 182-184) and added as Fig.4N-Q and further support a role for Crol in the control of wg expression in EC.

Concerning proliferation of the Wg activity area, we initially shared the intriguing idea the reviewer is implying, such that the Wg-activity crescendo might control growth and length extension of R5 (and thus also other Wg-controlled intestinal boundaries) into anterior direction. Unfortunately, we only collected evidence arguing against this tempting hypothesis: we did not detect pH3 positive cells at the MHB nor of the 'length' of the crescendo between VF and MF (not shown), although Wg::GFP is significantly increased between VF and MF at the MHB. Additionally, Wg and fz3 positive cells at the MHB seem to be a population of not mitotically responding ISC as they are negative for esg-Gal4 and immunostaining for the ISC marker Delta (not shown), in accordance with previous observations¹⁴. In conclusion, these findings prompted us to remove the data of Wg-activity at the MHB and now focus our presented data on the characterized R5 region of PMG anterior to the MHB and Malpighian tubules (Fig.3A-D',I-J, Fig.S4A-C).

5) Figure 5 micrographs are a bit puzzling. If the Rapport constructs work as the authors claim, there shouldn't be any GFP+ cells that are not RFP+. But in these micrographs, there are many GFP+ cells that are negative for RFP, especially in panes C and F. This could be misleading as the population of ISCs/EBs is measured as total number of GFP+RFP+ cells.

This is a very important point that we took very serious during the design, setup and functional testing of Rapport. As the reviewer mentions there are several GFP⁺ cells in micrographs of Rapport manipulations that show a weak mCherry signal (which is hard to see in such small images with merged channels without oversaturating the red channel). We noticed that weak mCherry⁺-cells occur with a higher frequency when tissue renewal is stimulated (e.g. in the initial Fig.5C and F showing >Ecl and >wg manipulations), which might be attributed to a slower fluorophore maturation time and/or expression strength compared to CD8::GFP.

The figure attached to this comment provides micrographs showing the mCherry signal of micrographs (shown in Fig.5C-D,F of the initial manuscript) in grey scale next to the original merged micrographs. Indeed, in conditions (such as e.g. >Ecl in A-A', >crol in B-B' and >wg in C-C') in which turnover is stimulated, more cytoplasmic mCherry signal is observed (outlined by yellow dashed lines). Important to point out, this signal does not stem from channel bleeding from the GFP channel, as the nuclear cavity seen weakly on the green channel, is strong in the mCherry channel. Additionally, arguing for reliable (but delayed) folding of mCherry is that Rapport delivers ReDDM tracing-like increases on progeny numbers (Fig.S6D-E)^{15,16}.

Taken together, we now provide a stronger boost of RFP signal in Fig.5 to more clearly show that all ISC/EB are double labelled.

Minor comments:

6) - In general, the authors have used student T-tests for statistical analysis. I would suggest them to use one-way ANOVA wherever there is comparison between more than two groups.

We thank all reviewers for commenting on the use of statistical tests in our comparisons between several groups. Following these suggestions, we now revised all statistical analyses by performing normality tests of our data, and then used unpaired t-tests for comparison of two groups with normal distribution and one-way ANOVA for comparison of more than two normally distributed groups. For data that did not pass all normality tests, we used Mann-Whitney U tests when comparing two groups and Kruskal-Wallis test for comparing more than two groups as recommended by others¹⁷ and now described more precisely in the material and methods section (lines 548-561). The results of performed statistical tests are now added to the raw data sheets submitted with the revised manuscript and we indicate in the figure legends which test was used for the presented data. In addition, we indicate technical (N) and biological replica (n) in all figures with quantifications.

7) - The authors claim that Crol has a cell autonomous function in in ISCs as an antiproliferative agent by inhibiting Stg. The dual knockdown of stg and crol is in line with this, however the dual overexpression of Stg and Crol sheds light also on to a cell autonomous role of Crol in regulating cell differentiation. Overexpressing Stg in progenitors expands the ISC/EB pool together with increase in EC/EE number whereas Crol overexpression had an opposite effect. Overexpressing both Crol and Stg brought the ISC/EB number even lower than the control whereas a larger number of newly produced mature cells are visible, indicating an accelerated maturation of progenitors. mRNA expression of both stg and cycB increases upon EcR signalling, suggesting that stg/cycB is not inhibited transcriptionally by Crol. Dual expression of Crol and Stg also supports this idea because Stg OE under UAS/Gal4 bypasses endogenous regulation, hence proving that Crol induces faster maturation of progenitors to EC fate. For me, it is not clear if the antimitotic effect of Crol in ISCs is due to faster maturation to ECs or blockade of cell cycle exit through Stg/CycB. A discussion regarding how Crol cell autonomously regulates Stg/CycB in ISCs would be appreciated.

The reviewer is raising an important point here concerning the role of Crol and whether Crol acts anti-proliferative only or whether Crol also includes differentiation, which we are now more extensively discussing in the revised manuscript (lines 134-148). Briefly, our data provides several lines of evidence that Crol acts on ISC proliferation and has no major effect on EB differentiation.

- i) *Crol does not induce EB to EC differentiation when driven in klu^{ReDDM} (Fig.S3J,M), which Eip75B-A/C (another 20HE induced factor involved in differentiation) strongly exerts⁴*
- ii) *Crol does not induce EC differentiation in N-LOF tumours (Fig.6B), whereas Eip75B-A/C expression expands the whole ISC pool into EC⁴.*
- iii) *Autonomous stg-RNAi as well as CycB-RNAi normalize ISC/EB numbers and ISC lineage production when combined with crol-RNAi (Fig.2I,K, Fig.S3C,E,G)*
- iv) *A main argument for a role of Crol in ISC and their proliferation control is our finding that Crol levels seem to be highest in EB (Fig.S3Q). EB are described as postmitotic population¹⁸⁻²⁰ and in physiology only very strong proliferation stimuli such as overexpression of oncogenic Ras^{G12V} or expression of constitutive active EGFR are capable of inducing mitosis within EB²¹. Upon knockdown of crol using klu^{ReDDM} , we observed mitotically active EB ($PH3^+$, Fig.S3N) which seem to re-enter cell cycle as shown for Ras^{G12V} and $EGFR^{Act}$ ²¹ suggesting that Crol is necessary to suppress proliferation in EB rather than a role in differentiation.*
- v) *The observed upregulation of CycB/stg upon 20HE activation, reflects resulting CycB/stg levels in ISC that are also affected by Wg from EC (Fig.5O), thus underlining our hypothesis of a mitotic balance (please see also comment #2).*

8) In Figure 3, the authors could label some landmarks and point to some cells to make it clear for the readers to understand what they are looking at especially the micrographs for Wg::GFP. Also, there is a disparity between Wg signal and fz3 reporter activity after RH5849 feeding. The quantification for Wg::GFP after feeding RH5849 shows only a 3% increase whereas fz3 reporter shows a 58% percent increase. How is it possible? Is it something to do with the way the authors have measured the intensities? Also, I am a bit lost at such a high number of sample values for Wg measurements? Do these high n values correspond to pixel values? If so, a better way would be to measure the mean from selected ROI as values from individual pixels can be very noisy.

As indicated in comment #4, we now provide new data where we measured Wg::GFP signal within EC of the R5 region instead of measuring at the MHB. New micrographs with clear indication of cell type and measurement area were added, and data shown in Fig.3 and Fig.S4 are as described in our answer to comment #4 and material and methods. Concerning pixel measurement, we measure ROI in EC of R5 and measure the mean intensity and given values for N is cell number whereas n reflects the biological replica.

9) - In Figure 1, the fold-change after feeding RH5849 (>10%) is much lower than mating-dependent response (>50%)? Is it because the titre of RH5849 that the cells experience at the used concentration is much lesser than endogenous 20HE produced after mating? Have you tried feeding the flies 20-HE (Sigma Aldrich H5142) to see if this can bring in a relative change similar to that of during mating?

We and others²² previously successfully used RH5849 instead of 20HE and prefer RH5849 due to its robust EcR activation and its design for high stability for use as an insecticide^{23,24}. 20HE levels necessary to elicit a similar response are highly unphysiological because of a very high concentration of 5mM 20HE to induce ISC mitosis in adult female midguts²⁵. We were able to reproduce these datapoint in our project investigating Eip75B but stopped using this high concentration of 20HE (2.400.000pg/ml) which is far from physiological levels of around 90 pg/ml that have been measured in the haemolymph of mated female flies⁴.

There is also a difference in the way RH5849 and 20HE are taken up: physiological 20HE induced by mating is transported by the haemolymph and imported into midgut cells by the ecdysone importer (Ecl)⁷, whereas stable RH5849 is passing the midgut lumen together with the food (Fig.1E/schematic) and is taken up by midgut cells.

*In addition, mating does not only induce EcR signalling, but also juvenile hormone (JH) signalling that was also already shown to reach and affect cells of the adult posterior midgut and contribute to organ size and cell number adaptation⁹. We briefly mention this in the manuscript now. By analysing 5kb upstream of the *crol* transcriptional start sequence, we found consensus sequences for basic helix-loop-helix transcription factors such as the Krüppel Homologue 1 transcription factor, the main effector of JH response, showing the probability that *Crol* levels could also be affected by JH signalling synergistically upon mating, which is also known for other hormonally stimulated factors such as *Taiman*²⁶. In addition, the literature points to an EcR activity regulation of *Crol*^{1,8,27}.*

10) - Is *Crol* expressed in midgut epithelium of male flies and is it sensitive to 20HE feeding?

*Indeed, *Crol* is expressed in the midgut epithelium of male flies. We detected *Crol*::GFP within all the different cell types of the R5 region as in females. However, we did not detect a change in *Crol*::GFP intensities upon feeding RH5849 and male flies are known to produce less 20HE²⁸ and are limited in cellular uptake and conversion of ecdysone by lower expression levels of *Ecl* and the converting enzyme *shade*²⁵. We added this data now to Fig.S1G-I and in the main text (lines 63-66).*

11) - *Crol* seems to be expressed ubiquitously. The authors have tried to explore the role of *Crol* in different cell types except enteroendocrine (EECs) lineage. Since *Crol* plays a crucial role in cell differentiation and loss of *Crol* in ECs non-cell autonomously affects ISC proliferation, could the authors try to knockdown *Crol* in EECs and report the non-cell autonomous effects, if any.

We thank the reviewer for this experimental suggestion. When designing Rapport, we not only envisioned and established a Rapport system for EC (*mex>*), but also established several additional Rapport tracing systems for further intestinal cell types (Table 1). One of them is 'EE-Rapport' (A), which enables manipulations of EE and simultaneous labelling and tracing of progenitor cells by complementing the system with an EE specific *Rab3-Gal4*²⁹. For the midgut aficionados, we want to mention here that *pros-Gal4* is not suitable as it is also expressed in *esg*-positive/EEP like cells. In accordance with Li and colleagues, we found *Rab3-Gal4* exclusively EE specific.

When we used the EE-Rapport System for manipulations of *Crol* and ectopic expression of *hZNF267*, we observed no paracrine effects on progenitor cell numbers (B,D-G) and cell differentiation (C-G). We also checked EE specific overexpression of ligands such as *wg* and *spi* (data not shown), but observed no paracrine effects on ISC proliferation or differentiation. This suggests that EE do not possess the cellular machinery necessary to secrete Wnt/Wg and EGF ligands, whereas other ligands can be released from EE³⁰ and Zipper&Reiff (unpublished).

12) - The model based on cell autonomous inhibition of proliferation by *Crol* in ISCs and mitogenic effect of *Crol* in ECs is interesting, but it is too qualitative and not sure how relevant it is in gut physiology is not clear. It assumes that ISCs are distributed uniformly and this distance to ECs increases during organ growth. This might be true in case of homeostasis, but regenerative responses might induce a bout of cell divisions that alter this pattern of ISC distribution. Even though the total number of ISCs stays before and after mating-induced proliferative response, authors should also quantify ISC distribution before and after mating. This model holds true only if the distribution shows that the ISCs are uniformly distributed even after mating.

The nature of our model was intended to be qualitative. With modelling organ growth, we wanted to test whether its mathematically plausible that organ size and cell numbers are adjusted by opposing roles of *Crol* in response to a single hormonal stimulus. Thanks to the less

strict space restrictions of NComms and the constructive review process, we now describe the model and its applicability much more exhaustive (lines 279-286).

In addition to our initial submission, where we showed that ISC numbers remain constant upon mating, we now added data showing that normalized to the total area, mated females (MF) have less ISC per 100000 μm^2 compared to virgin females (VF, Fig.S7E). Here, we now include results that show that distances between ISC are longer in MF compared to VF (micrographs A-B' and quantification C). ISC are unequivocally identified by being GFP⁺ only (*esg*⁺-only) whereas EB are double marked by GFP and RFP (*esg*⁺/*klu*⁺). Distances between two ISC are shown with white lines.

It is true that there is a marked difference between homeostatic and regenerative responses. In our study we concentrated on homeostatic conditions, whereas other very recent studies are looking at ISC (re-)distribution through migration upon insults to the intestine³¹. Under homeostatic conditions, we did not specifically investigate ISC migration but the increase in distance between ISC in C is around 30%, which perfectly reflects the usual mating dependent size adaptation caused by epithelial growth⁹.

13) One important component missing in this current model is time. The model is constructed as if there is one to one relation between ISC mitosis and an EC production. Authors did not take into consideration how different is the time for a proliferative cycle of an ISC versus the time taken for EBs to transform to mature ECs. If ISCs can finish 3 to 4 cycles of division before a new EC is produced, I am not sure how this model can hold true. Are these dynamics known during homeostasis and mating-dependent regeneration. I would bring a temporal information into the model. Another event where time matters is the signalling itself. 20HE has a direct effect on Crol in ISCs as an antiproliferative signal whereas mitogenic effect of 20HE from ECs is indirect. 20HE increases Crol in ECs which in turn increases Wg and its secretion, which then act in ISCs as the mitogenic signal. Since both ISCs and ECs on the same epithelium are exposed to the same levels of circulating 20HE. We can speculate that the action of antiproliferative signal should be faster than the mitogenic signal. How will such a system result in an increased organ size unless there is a thresholding or a cell-type specific

regulation of 20HE-Crol activity? For the current model to be exact, ECs should respond to 20HE faster than ISC indicating that is ECs should be more sensitive to circulating 20HE than ISCs. In this model, the authors consider that as the tissue proliferates, the distance from the Wg source to the ISC increases. But what is not clear to me is that even when the tissue is in a proliferative state, the new mature cells that are produced are still ECs and to they are still responsive circulating 20HE, which means there is no change in effective distance between Wg source and ISCs. How this parameter is included in the model.

Thanks for so many useful comments and we really appreciate the suggestions for improvements of our model. As mentioned in comment #12, we vastly extended our model now based on the review including a new temporal component. In particular, we consider, as the reviewer pointed out, that the transit between ISC to EC is not immediate. This implies, in abstract terms, that a population of cells in an "intermediate" state transiting in finite time from one to another has to be taken into account. We consider that, on average, these cells differentiate after a certain characteristic time span. The inverse of such characteristic time span is the rate at which cells differentiate into EC and, as such, can enter into the dynamical equations. Now we have a two dimensional system, with one equation accounting for the evolution of EC and another for the evolution of the cells in the intermediate state transiting from ISC to EC. Interestingly, the steady state for the EC population in terms of the hormone concentration remains the same, along, as expected, with a strong non-linearity on the behaviour of "intermediate" cells after the increase of the hormone. All in all reinforce and enrich the main question behind the model: Whether the opposing trends of 20HE can lead to 1/ a stable EC population and 2/ that such EC populations evolve consistently with the concentration of 20HE. We considered other extensions of the model, such as introducing a time delay in the equations, or introducing a degree of stochasticity. However, these extensions would imply an additional layer of mathematical complexity that would, in our opinion, blurry the message of the model, mentioned above. In the current manuscript, we are confident that we propose a model that is simple enough but not too simplistic. We also expanded the section related to the construction of the model, to show that our assumptions are grounded by plausible dynamical considerations regarding raw Wnt/wg-ligand diffusion and degradation over a certain geometry.

As commented in #12 as well, we now provide biological data that indeed shows that ISC distances grow upon mating. Concerning the last part about the parameter: Growing tissue reflected by EC numbers in our mathematical model will lead indeed to the same amount of Wnt/Wg by the same amount of 20HE but then distributed over the increased EC population. Thus, some EC will be more distant to an ISC, in turn aggravating the proliferative effect of Wg degradation. This declining proliferative stimulation on ISC is balanced in ISC by 20HE-Crol acting anti-proliferative. Initially the Wnt/Wg stimulus outweighs degradation of Wg (Fig.7G) and the autonomous anti-proliferative effect (Fig.7A,G), which then changes with mating size growth so both effects come into balance (Fig.7G'). As mentioned above, we substantially added to the paragraph describing the model (lines 291-321) and the according Figures (Fig.7, Fig.S7).

We are deeply grateful to this reviewer, as her/his comments really pushed us to improve the modelization part.

- In figure 2L, one of the bars corresponding to a p-value is shifted.
- In Figure 3, S4, and S5, if the LUT for fz3-RFP is grey, it would be easier to recognise the signal.
- Could the authors include more a picture of tumours than a scheme in the Fig 6K?

- Have the authors tried to knock down Wg only in the MHB other than using a pan-EC driver mex?
- In line 222, regarding fly survival, the legend is mislabelled as Fig S6L but it should be Fig 6L.
- In line 223, regarding gut length, the legend is mislabelled as Fig 6M but it should be Fig S6D.
- In line 225, regarding multilayering, the legend is mislabelled as Fig S7D but it should be Fig 6M.

We thank the reviewer for pointing out these seven issues that we now corrected in the revised manuscript.

Concerning the fourth comment: Unfortunately, there is no driver known to us being specific in the MHB region.

Reviewer #2 (Remarks to the Author):

The study by Zipper and colleagues highlights a zinc finger transcription factor Crooked legs (Crol) as a key coordinator of cell behavior and organ size adaptation in the adult female *Drosophila* intestine in response to changes in the steroid hormone ecdysone induced by mating. The authors developed a Rapport ('Repressible activity paracrine reporter') system that allows precise spatiotemporal control of transgene expression while performing stem cell lineage tracing. Using the Rapport, they demonstrated the differential effect of *crol* and its putative human ortholog ZNF267 on intestinal stem cells (ISCs) and enterocytes (ECs). Their results show that an excess of Crol in ISCs inhibits their proliferation by suppressing String (Stg)/CDC25 and Cyclin-B (CycB) through an unknown mechanism. Conversely, upregulation of Crol in enterocytes (ECs) stimulated ISC division in a paracrine manner by increasing the production of Wingless (Wg). Importantly, they underscored the importance of finely regulating Crol levels in different intestinal cells, extending beyond physiological functions, by manipulating Crol in cancer stem cells and the tumor microenvironment using two different intestinal fly tumor models. This is an exciting study, that uses state-of-the-art *Drosophila* genetics to provide novel mechanistic insights into organ size regulation in response to endocrine inputs by exploiting distinct cell type-specific functions of a single transcription factor. The results and developed tools would be of broad interest to the readership and scientific community. Unfortunately, several conclusions are not well supported by the data presented. The tumor data appear sketchy and incomplete.

We thank the reviewer for the positive evaluation and acknowledgement of the possibilities of Rapport for the community.

Major Comments

14) The upregulation of Crol in response to changes in ecdysone levels post-mating in female fly intestines is the primary discovery of the manuscript. I find it difficult to reconcile the representative images in Figure 1A-C with the quantifications in Figure 1E. It is unclear whether the quantifications of Crol::GFP fluorescence were performed on guts stained with antibodies that distinguish ISC/EBs from EE and ECs. The nuclei marked as ISC/EB in Figure 1A-C seem larger and the distinction isn't as pronounced compared to Figure S1B-C.

We agree with the reviewer that Crol responding to 20HE in the intestine is a central finding of our paper. Guts were stained with antibodies targeting Armadillo and Prospero^{18,32} to unequivocally distinguish ISC/EB, EC and EE for quantifications of Crol::GFP levels in the different cell types (Fig.1, Fig.S1D-H'). We adapted the micrographs in Fig.1 to this valid comment and now show images of Crol::GFP in greyscale and Crol::GFP combined with aArm and aPros staining (Fig.1, Fig.S1D-H'). Importantly, in all fluorescence measurements, we did not use antibodies to amplify signals like GFP or RFP as it affects fluorescence intensity range and we now indicate biological and technical replica.

15) The quantifications throughout the manuscript look impressive with remarkably high power. Unfortunately, the choice of a T-test is often inappropriate. It is recommended to utilize ANOVA with multiple comparisons for most data. Moreover, considering the negative binomial distribution of numerical/count data, employing non-parametric tests like Kruskal-Wallis and Dunn's test is advisable.

Following this comment by the reviewers, we now reworked statistics as indicated in our introduction to this letter, the revised material and methods section (lines 548-561) and described in point #6 responding to Reviewer 1.

16) Reusing of data in Figure 4B, C, D and the claim of significance using a t-test are incorrect.

Data leading to panels 4B-D are from experiments that have been performed in parallel. For simplicity and logic, data was split up into separate panels. We revised the figure legends and the new statistical analysis using Graphpad (described in the material and methods section in lines 548-561), which can now be found in the raw data (comment#6).

17) The authors' statement that the Wg::GFP signal is increased in enterocytes at the midgut-hindgut boundary lacks sufficient support in the provided data. To validate this claim, it would be beneficial to include images combining Wg::GFP with staining highlighting various midgut cells. Additionally, clarification on the specific region of the midgut displayed in the high magnification image (Figure S4A) is needed. Did author use any normalization when quantifying the Wg::GFP signal? Can they comment on a clear difference in Wg::GFP signal between control and MeOH treated intestine?

Instead of measuring the Wg::GFP signal in cells of the MHB, we now analyzed Wg::GFP signal within EC of the R5 region which is the region that we actually investigated with ReDDM and Rapport (also see comment #4). In this revised manuscript, we added new micrographs to Fig.3, Fig.4 and Fig.S4 showing Wg::GFP in EC of R5 combined with antibody staining targeting Arm and Pros for identification of different cell types and for better recognition of differences we provide greyscale images of the GFP (Wg) / RFP (fz3) channel. We also added an outlining to an exemplary measured EC to facilitate understanding of how Wg::GFP was measured. We did not use normalization of the signal. For additional details, please see also comment #8/reviewer 1 and the revised Material and Methods (lines 486-515).

Fortunately, our new measurements of EC in R5 show much less variation when controls are compared (Fig.3I,J, Fig.S4C) and we also have to state that Wg::GFP levels between different experimental sets (VF and MF (Fig.S3A-C), MeOH and RH5849 comparison (Fig.3A-B',I), and w^{1118} and ovo^{D1} comparison (Fig.3C-D',J)) shown in the original manuscript are not directly comparable. As example VF vs. MF sets (Fig.S3A-C) have been performed at different time points than MeOH/RH5849 sets (Fig.3A-B',I) and so on. In addition, guts were imaged at different time points and even though settings were identical, confocal lasers intensity depends on various factors such as temperature, humidity and age as intensity declines during months of duty³³. Our experts at the Centre for Advanced Imaging do regular intensity checks to minimize these factors.

18) How do the authors reconcile the spatially restricted induction of Wg::GFP at the midgut-hindgut boundary with the increased levels of 20HE in the hemolymph circulating throughout the body? Do the authors have evidence for differential uptake of the 20HE agonist RH5849 along the adult intestine?

The new data added to this revised manuscript in Fig.3, Fig.4 and Fig.S4 shows that the Wg::GFP induction is not restricted to the MHB but also detectable in EC of the R5 region. In addition, the observed difference of wg::GFP induced by 20HE-signalling is even more pronounced than at the MHB from the initial submission (Fig.3I,J, Fig.S4C). In general, we did not observe different regional responses to RH5849, whereas mating clearly affects R5 the strongest probably due to the synergistic effects with JH (see comments #3 and #9), which is why we focussed on R5. What we do know is that the 20HE uptake response in R4 is also taking place addressed by Rapport (see comment #5).

19) The authors state that "we detected fz3-RFP signal in intestinal precursors along the midgut (Fig. 3E)" (line 152). Unfortunately, I find no support for this statement in Figure 3E. From the images provided, it is impossible to determine where the entire midgut might be. The authors need to provide confocal micrographs of fz3-RFP midguts where individual cell types can be clearly distinguished based on nuclear staining combined with immunostaining with antibodies against e.g. Arm, Dlg1, Pros. Please note that the current micrographs are of insufficient quality. Providing grayscale variants is highly recommended to highlight potential differences between genotypes. As presented, I find it difficult to reconcile micrographs showing fz3-RFP signal with the quantifications and significance presented in the corresponding graphs.

Following all reviewers' suggestions, we reworked the describing text and Fig.3 (and Fig.S4) in such a way that we provide high quality grayscale images for fz3-RFP in Fig.3E-H and Fig.S4D-E allowing to see visible fluorescence intensity changes upon manipulations. Facilitating identification, we added images showing fz3-RFP combined with antibody staining against Arm and Pros to identify intestinal cell types and point to the cells measured with arrows (Fig.3E'-H', Fig.S4D'-E', as described for Crol::GFP in previous comments #1 and #14).

Detectable fz3-RFP signal in precursor cells all along the midgut was already shown by other groups^{12,34}. We now describe the identification and location of the measured fz3-RFP signal in aArm positive progenitor cells in R5 more clearly (Fig.3E'-H') in the revised manuscript and legends (lines 159-162).

20) Clarification is needed regarding the evidence supporting human ZNF267 as the functional orthologue of fly crol. Reviewing Flybase, Diopt, etc., other ZNFs such as ZNF606, ZNF850 are listed. The authors need to provide additional evidence and analysis to justify their focus on human ZNF267. The authors show that hZNF267 OE attenuates the mitogenic effect of esgReDDM>crol-RNAi. Can the authors rule out that the "rescue" effect of hZNF267OE is a general response to ZNF excess? Would ZNF606 or ZNF850 have the same effect? Can they rule out a saturation effect of the system with the two UAS constructs? Does +hZNF267OE in wild-type background recapitulate the effect of crolOE?

We took the liberty to split this important comment into two questions, one concerning ZNF267 and Crol and the other on Crol/ZNF267 in tumour models found on the next page:

Concerning Crol and ZNF267 homology, the reviewer is entirely right asking about more details regarding the identification of ZNF267. When we began working on Crol, ZNF267, ZNF366 and ZNF841 were the top hits among the orthologues in Flybase. ZNF606 was not mentioned, which points to constant updated of Flybase and its ortho predictions. (When we last checked Flybase (27.07.2024) ZNF850 mentioned by the reviewer is listed at 29th position of human Crol orthologues suggesting a high dynamic of this list). When we started functional investigations of ZNF267, ZNF366 and ZNF841, stocks for expression of hZNF267 and hZNF366 were commercially available and we decided and took the effort to make transgenics for expression of hZNF841 in our lab based on that initial list. Both, hZNF267 and hZNF366 were interesting candidates as they were already connected with cancer initiation³⁵, associated with Wnt/Wg signalling (Fig.S7A)³⁶ and Estrogen Receptor signalling^{37,38}.

Expression of hZNF267, hZNF366 and hZNF841 did not recapitulate the effect of Crol OE in esg^{ReDDM}, whereas hZNF267 and hZNF366 attenuated the mitogenic effect co-expressed with crol-RNAi (A, Fig.2D-E). When we investigated paracrine Wnt/Wg signalling activation through hZNF366, we detected opposing effects compared to Crol (B), and the tumour reducing effect of hZNF366 and hZNF841 was comparably weak in contrast to hZNF267 (C-G) suggesting an anti-proliferative role like Crol (Fig.6B,E). Thus, our functional analysis of these three ZNFs

pointed to hZNF267 as the functional orthologue of Crol, which is why we focussed on hZNF267. We now added a brief description of the identification of ZNF267 as functional Crol orthologue to the revised manuscript (lines 106-114).

Concerning the question about a saturation effect of the system with two UAS constructs: Our *esg^{ReDDM}* system plus *N^{DN}* and both UAS-driven fluorophores does not visibly affect their expression strength. As exemplary: as shown in Fig.6A-D, we can add additional UAS-driven transgenes to the system like UAS-*N^{DN}* to the UAS-driven *Crol* manipulations and we can still see all fluorophores and other transgenes being expressed as they result in tumour formation and additional effects.

Although potentially very interesting, the analysis of *crol*/*ZNF267* in the two different intestinal tumor models seems incomplete. The results suggest a differential effect of the paracrine *crol*/*ZNF267* on ISCs/EBs. Furthermore, the authors fail to describe and discuss the *crol*-RNAi data. How do the authors explain the limited to no effect of *crol*-RNAi in Notch-RNAi tumors? It appears that EE accumulation is prevented when *crol*-RNAi is expressed in Notch-deficient ISCs. In contrast, *crol*-RNAi does not exert anti-tumor effect when expressed non-autonomously in ECs? How relevant is the paracrine *crol*-Wg axis in the two different tumor models?

We thank the reviewer for pointing out the potential of our two tumour models and now provide more details in the manuscript and a more exhaustive discussion here:

In summary, in so called Notch-tumours EC differentiation is blocked by the LOF of Notch, which results in an accumulation of proliferative ISC-like and endocrine progenitor cells and the absence of EB³⁹. In this elegant work, Patel and colleagues show that strong mitotic stimuli (EGF and then JNK) take over during tumorigenesis by the simple LOF of *N*. In our paradigms, we make similar observations in the sense of that more and more factors contribute to tumour growth. Concerning autonomous *crol*-RNAi, which under physiological conditions stimulates

proliferation (Fig.2C,E,L), we interpret the lack of further mitotic induction in N tumours as a result of the induction of further mitogens outweighing crol's antiproliferative role. Indeed, Parthive Patel found that EGF signalling³⁹ that, like crol-RNAi, acts on Stg and Cyclins⁴⁰ is upregulated relative early during tumorigenesis of N tumours.

In addition, Patel and colleagues showed that expression levels of wg are induced within the niche of Notch tumours³⁹, but the source for Wg was not further specified. It is tempting to speculate that hormonally induced Wg by the 20HE-Crol-Wg axis is involved from the tumour surrounding based on our observations: In Fig.5F,J, we show that Wg acts on progenitor proliferation and Fig.6G shows that Crol-Wg from EC strongly boosts proliferation acting on CycB as well (Fig.5O), an effect which is only partially reduced when Wg is depleted by RNAi. This points to either a not fully penetrant RNAi depletion and/or secondary mitogens induced by Crol. In addition, it is likely that tumorigenesis itself induces stronger proliferation stimulating mechanisms that take over mitotic control³⁹, that might indicate less relevant paracrine Crol-Wg axis in Notch-tumours underlined by the fact that crol-RNAi does not induce N-tumour growth (Fig.6I).

Altogether, our two N-LOF paradigms investigating the Crol-Wg and Crol-Stg axes reflect a simplification of the heterogenic cell type composition found in patients and show that in less aberrant cells, the Crol-Stg axis acts anti-proliferative. Our multigenic CRC avatars underline this anti-proliferative role in the tumour stem cell like cells further even strongly improving survival. A model which will allow investigation of non-autonomous effects on 'true sporadic' CRC growth in avatars is a mid-term goal in the lab. We now discuss this issue more exhaustive in the revised manuscript (lines 239-242, 250-253).

21) Quantification of data: There's ambiguity regarding what "n" represents in certain figures. For example, in Figure 3I or Figure 4B, the authors seem to be referring to the number of cells counted. If this is the case, they need to specify the number of intestines/animals they analyzed and include in the methods how many independent experiments the samples are from.

Thank you for pointing out this lack of uniformity in data presentation, we added n (biological replica) and N values (e.g. cell number) to the graphs in all figure panels.

Minor comments

In the Materials and Methods, line 26, the authors likely mean genomic DNA rather than cDNA as the target of gRNAs.

Thank you, we corrected this.

The authors should discuss how increased Wg-GFP in MHB affects fz3-RFP levels along the entire midgut. One might expect a gradient of fz3-RFP induction. Can the authors provide any evidence?

Please see our comment to reviewer 1, point #4.

Would crol-GFP also be induced in male intestines fed with the 20E agonist RH5849? Is the 20HE-Crol-Wg axis sex specific?

Please see our comment to reviewer 1, point #10, with new data that is now included in Fig.S1G-I of the manuscript.

The authors may want to provide a more detailed explanation of the CRC model to allow a broader audience to appreciate their findings.

Thank you for appreciating our model created in Zipper et al., 2022. Thanks to less strict word limitations, we now added a more detailed description of our CRC model (lines 254-260). In addition to the scheme (Fig.6O) we now include micrographs showing tumours induced by our CRC model in Fig.6K-N in the revised manuscript.

Consistency in writing genotypes in tables and over confocal micrographs. Authors may consider listing all transgenes for individual panels to be self-explanatory and consistent throughout the manuscript. For example, writing +hZNF267 (Figure 2E) or +crol-RNAi (Figure 2J) may not be properly understood.

Thank you for this suggestion, we understand that the incomplete indication of genotypes written in the figures saves space but might not be self-explanatory. We changed them to complete genotypes throughout the revised manuscript (Fig.1, Fig.4, Fig.5, Fig.6, Fig.S3).

Why do the control tumors in Figure 6A and 6F look so different?

Control tumours shown in Fig.6A are induced by esg-Gal4 driving expression of UAS-driven dominant-negative Notch, whereas control tumours shown in Fig.6F are induced with esg-lexA driving expression of Aop-driven N-RNAi. We usually observe that the use of dominant-negative N to induce tumours is more efficient (Fig.6A-E, compare with Zipper et al., 2020) as the RNAi-mediated generation of N-LOF tumours. A delay in N-LOF tumour generation by RNAi depletion might involve mechanisms like N-receptor half-life, endocytosis and RNAi efficiency itself. In addition, genetic background and enhancer activity (esg-gal4 vs. -lexA) might cause additional differences in observed tumour sizes. The original micrograph shown in Fig.6F did not contain EE tumours (clusters of small, only RFP⁺ cells), which is why we now replaced the micrograph shown in Fig.6F in the revised figures.

Figure 6L - The viability of the control line used for CRC tumor induction should be shown, ideally with pCFD6_noSapI transgene expressing mock gRNAs.

We agree that performing controls based on gRNAs is important as well in addition to the genetic background control CRC avatars that we show in Fig.6P.

In Filip Port's publications that introduce gRNA flies based on pCFD, the authors suggest the use of γ -(yellow) and se-(sepia)-gRNAs as controls as both gRNAs target nonessential pigmentation genes⁴¹. Concerning the no SapI plasmid, the pCFD6_no SapI plasmid⁴² contains a non-coding single nucleotide exchange to remove the SapI restriction site compared to the original pCFD6⁴³ and thus no difference in the guideRNA coding regions.

We previously performed survival experiments with ISC/EB specific CRISPR/Cas9 induced knockout of γ and se in pCFD6 using esg^{ReDDMCas9}. Survival upon γ - and se KO is slightly (but significantly) reducing overall survival probability compared to esg^{ReDDMCas9} w¹¹¹⁸ background controls (green). Overall, the median survival time of 28 days for the γ -gRNA and 26 days for se-gRNA are quite close to our shown white control (28 days) and still significantly (35%) higher compared to 9 days median survival in CRC avatars.

Importantly, survival of w¹¹¹⁸ control CRC avatars significantly differs to γ (and se gRNAs) as well. We agree with the reviewer that the broad readership of Nature communications is not

used to control survival times of *Drosophila* and now include the γ -guideRNAs and statistics showing differences to 'standard' CRC avatars in the revised Fig.6P. For simplicity, we would like to refrain from showing w and se controls in the main figure and add the graph below to Fig.S7D and the main text (lines 258-260).

Figure 5C - Can the authors comment on why most cells are RFP negative after EcIOE?

Please see reviewer 1 point #5 for this issue.

General comments

Considering that colorblind individuals may have difficulty distinguishing between red and green colors, using a colorblind friendly palette is recommended.

We discussed this issue with the editor and now provide additional greyscale micrographs in Figures 1, 3, 4, S1, S2, S4 and S5.

Reviewer #3 (Remarks to the Author):

The manuscript by Zipper et al. tackles the question how intestinal size is regulated upon mating in *Drosophila*. They identify a transcription factor Crol which is regulated by the ecdysone hormone followed by mating. They find an interesting mode of regulation where Crol regulate ISC proliferation in an opposing manner depending in which cell type it is expressed. In ISCs, Crol inhibit ISC divisions, whereas in ECs, Crol induce ISC proliferation in a non-cell autonomous manner. In addition, Crol manipulation in intestinal tumor model has a profound effect on tumor size and fly survival. The authors suggest mechanisms for the cell autonomous and non-cell autonomous function of Crol. In ISCs, it is suggested that Crol negatively regulate String and block ISC cell cycle progression. In ECs, it is suggested that Crol regulate Wnt secretion and thus the proliferation of the nearby ISCs. Based on their data, the authors conclude that Crol regulate physiological intestine size regulation upon mating downstream of ecdysone and upstream of Wnt signaling to increase intestinal stem cell proliferation. Finally, the authors provide a mathematical model to support their conclusions.

While the effect of Crol in regulating ISC proliferation in wild type and tumor model is clear and interesting, the authors do not provide sufficient data to support the conclusion that Crol regulate intestinal size as a response to mating through Wnt. While these are the main conclusions of the manuscript, additional evidence is needed, or the conclusions and manuscript need to be significantly readjusted.

Key conclusions which are not supported/not shown by the provided data:

1. Crol regulates the physiological intestine size adaptation upon mating.
2. Crol regulates Wnt secretion from ECs.

In addition, there are serious problems with data analysis, particularly with the statistical tests used. This hinders my evaluation of the data. The authors incorrectly use t-test for multiple group testing, do not provide information what type of t-test is used, or use it with data which is not normally distributed or with non-parametric data. These issues should be corrected.

Thank you for the critical evaluation of our data and drawn conclusions, which we will address and discuss in the following comments.

Concerning the statistical analysis, we revised all statistical analyses as stated in the general response to all reviewers, further detailed in comment #6 of reviewer 1 and the revised material and methods section (lines 548-561). We reworked and extended the revised manuscript, data and figures accordingly.

Please find more specific comments/questions below.

Fig1

22) How are the ISC/EB cells vs ECs identified? If only nuclei size is used as a criterion, more appropriate wording is small nuclei cells/large nuclei cells.

Please see reviewer 1 point #11 for how ISC/EB and EC are unequivocally identified. We reworked the figures and text accordingly.

23) E: please provide the N guts and N cells in the figure panel instead of the legend (refers to all figure panels). The data in VF samples seems not to be normally distributed, hence the use of t-test might not be appropriate here. Please test for normality and use a proper statistical test (applies to all statistical tests).

Please provide more information about the intensity measurements (applies to all figures):

We added n (biological replica) and N values (cell numbers or technical replica) to all figure panels (please see comment #21). Concerning the statistical analysis, please allow us to refer to the general comment in the beginning of this letter and detailed on in comment #6. Details about how the intensity measurements of the different fluorophores were performed can now be found in detail in the material and methods section (lines 486-515) and we also updated the figure with arrows and borders pointing to measured cells in order to be more comprehensive (Fig.1A-D', Fig.S1D-E'). Please find more details in the following comments as well.

24) In the materials and methods it is stated as following: "Fluorescence intensities were analysed in Fiji by determining the mean intensity per area of manually selected nuclei. This way, *cro1::GFP* and *fz3-RFP* expression levels were measured and normalized to number of ISC/EB or area of R5 respectively." What does it mean "normalized to number of ISC/EB or area of R5 respectively"?

*We now more precisely describe all fluorescence measurements in the material and methods section and comments #14, #17 and #19. Concerning normalization, *Cro1::GFP* levels were normalized to area of R5 by multiplying the measured intensities with a calculated fold-change in R5 area (in comparison to the corresponding control). Considering mating induced area changes, the dilution of actual hormone dosage reaching a single cell in a more numerous mating adapted intestine is taken into account. Similarly, *fz3-RFP* intensities have been normalized to ISC/EB numbers by multiplying the measured intensities with a calculated fold-change in ISC/EB numbers compared to controls. In this way, *fz3*-activity is compensated for dilution and degradation of *Wg* ligands reaching individual cells with increasing ISC/EB numbers as sink for *Wg* ligand, which is also included in our mathematical modelling. We now more precisely describe this in the manuscript (lines 291-294 and Supplemental material for mathematical model) in addition to material and methods (lines 486-515).*

25) Are the intensity measurements background subtracted or not?

*For *Cro1::GFP* and *fz3-RFP* intensities, we did not subtract background as these nuclear signals are clear and robust when outlined. For *Wg::GFP* intensity measurements, we outlined EC for measuring mean intensities and subtracted the background signal. Supporting this mode of action, *Wg::GFP* signal is also seen in the nucleus of EC where *Wg* should not localize (compare Fig.3A-D', Fig.S4A-B'). *Wg::GFP* signal is generally more noisy and shows a low signal-to-noise ratio. A detailed description of how intensities were measured is now part of the revised material and methods (lines 486-515) and is more intuitively described in the micrographs.*

Fig2

26) Based on epistatic analysis, the authors make the conclusion that *Cro1* directly inhibits *stg* to control ISC proliferation (Fig2 M). To support this conclusion more data is needed (to show *Cro1* regulates *stg* transcription), or the conclusion needs to be rephrased to allow indirect regulation.

Please see point #2 for a detailed clarification of this issue including data from the literature, new data and revisions of the manuscript text (lines 132-133, 147-148).

27) E, K, L: Student's t-test is not appropriate to test multiple samples. Please provide ANOVA to compare the means between groups (if normally distributed data). This refers to all figures in the manuscript where t-test has been used to test multiple groups. Please show the individual data points in the plots and provide N in the figure panels (refers to all box plots in the manuscript).

Please see point #6 and the revised materials and methods section (lines 548-561) for details on this issue. We revised all graphs to show individual points (Fig.2E,J-L, Fig.5J-K,N-O, Fig.6E,J,Q-R, Fig.7A',B', Fig.S2B-C,H, Fig.S3G,L-M,P, Fig.S5D-E, Fig.S6D-E, Fig.S7E-) and added N and n numbers in the figure panels. We thank the reviewer for this constructive improvement of our data presentation that we will apply in future works.

28) F: What statistical method was used?

We thank the reviewer for pointing this out. We used unpaired student's test for the analysis of our qPCR data. In addition, we include a Kruskal-Wallis test in our raw data section that also detected a significant change and added this information to the figure legend. Please see point #6 and the materials and methods section (lines 548-561) for details.

29) G-J: Please provide the control image for these experiments. Please provide the genotype in the Figure panels (instead of + or >crol etc., it is difficult for the reader to follow what manipulation/genotype is used) (this applies to all figure panels).

*We thank the reviewer for pointing out these difficulties in comprehensibility of the exact genotypes in our figure panels. The according *esg*^{ReDDM} control for Fig.2B-I is shown in Fig.2A. For better understanding and uniformity, we now adjusted the order of panels and added full genotypes throughout the revised manuscript and figures (Fig.1, Fig.4, Fig.5, Fig.6, Fig.S3).*

30) Please show the sequence homology between Crol and ZNF267 (can be in the supplement).

We added an alignment showing the sequence homology between Crol and hZNF267 in Fig.S2M and describe it in the revised manuscript (lines 107-109). A more detailed discussion about the isolation of ZNF267 as functional Crol orthologue can be found in our answer to comment #20.

Fig3

31) A-D: The authors conclude WG::GFP signal is seen in the EC cells (row 147-148), however it is not possible to make this conclusion from the images provided. Please provide DAPI channel as well as A-P direction to the images and indicate the mid/hindgut boundary. Is WG::GFP also increased in other region boundaries or is it specific to mid/hindgut boundary?

*In the revised figures (Fig.3, Fig.4 and Fig.S4), we now added micrographs showing *Wg*::GFP in EC of the R5 region instead of MHB. Please see our answer to comment #17 for a detailed description.*

32) E-H: The authors state they have quantified fz3-RFP signal from ISC/EB cells, but from the images it is impossible to say what cells are shown as RFP positive. Proper markers for ISC/EB should be provided. From which region are the images and quantifications from?

Thanks for pointing this out. We added micrographs showing antibody staining targeting the ISC/EB marker Armadillo³² and the EE marker Prospero¹⁸ which we used for identification of different cell types (pointed out by arrows now). The images and quantifications are from R5 regions. We added this information to the figure legend and main text (lines 159-162).

33) I-J: How is the quantification of Wg::GFP intensity made? Are there more Wg::GFP positive cells in RH5849 fed flies or ovoD1 flies? The +3% increase in the mean Wg::GFP intensity signal is very modest in I. Do the authors consider this to be biologically significant? What type of t-test is used to calculate the p value in I?

Please see our answer to comments #4 + #17. Following the constructive evaluation of our initial statistics, we now performed new tests for the analysis Wg::GFP in Fig.3I-J (20% upon administration of RH5849 and 15% upon genetic ablation of ovaries). We used now unpaired-t-test for the normally distributed values (Fig.3I Wg::GFP) and Mann-Whitney U test for not normally distributed values (Fig.3I fz3-RFP and Fig.3J). A detailed description of quantifications of fluorescence intensities and statistical analyses can be found in the revised materials and methods section (lines 486-515, 548-561).

34) K The authors make the conclusion that Crol regulate Wg secretion from the ECs but no data is provided to support this conclusion.

The reviewer is correct about the fact that we did not formally address Wg secretion in our experiments. Our only statement concerning Wg secretion stated 'Wg release' in the abstract, which we now corrected to 'Wg expression'. The direct binding of Crol to the Wg promoter was previously shown¹ and this reference is now included in the revised manuscript (lines 185-187).

Our initial and new data strongly suggests that Crol induced Wg from EC in turn non-autonomously acts on ISC/EB with several lines of evidence. In Fig.4 we show changes in Wnt/Wg signalling activity in ISC/EB upon EC specific Crol manipulations. Additionally, we now show that Wg protein levels in EC of R5 increase and decrease upon EC specific overexpression and knockdown of Crol (Fig.4N-Q) and that Crol acts through Wg in non-autonomously controlling ISC/EB behaviour in Rapport (Fig.5D, I, L-N). In the course of this revision, we also tested whether wntless (wls), a protein required for secretion of Wnt ligands^{44,45} and localized inter alia in cytoplasmic vesicles, is expressed in the adult midgut. Using a wls::GFP stock, we found that the expression of wg (A-B') and wls (C-D') overlap at the MHB (A-A',C-C') and that both proteins are expressed exclusively in the EC population in R5 under homeostatic conditions (B-B',D-D'). Whereas the fz3-RFP reporter for Wnt/Wg signalling activity reflects Wnt/Wg signalling induction in the MHB (E-E') and solely the ISC/EB population of R5 (F-F').

Together, these data indicate that *Crol* is inducing expression of *Wg* ligands in EC in the adult midgut, which in turn implies *Wg* release by EC to non-autonomously activate *Wnt/Wg* signalling (*fz3-RFP*) in ISC/EB. We now mention the expression of *wls* in the revised manuscript (lines 156-158).

Fig4

35) This is the only image providing data about *Crol* regulating *wnt* pathway in ECs. More evidence is needed to support this conclusion.

Our initial and new data provides several lines of evidence that Crol controls Wg ligand expression in EC that in turn non-autonomously activates Wnt/Wg signalling pathway in ISC/EB (comment #34). We did not detect any Wnt/Wg signalling cascade activity in EC (by fz3-RFP) suggesting Wg ligand expression is the only component of the Wnt/Wg pathway in EC. Including the new Wg::GFP data in Fig.4N-Q, results in Fig.4 and Fig.5, showing the non-autonomous effect of EC-derived Wg, and the direct binding of Crol to the Wg promoter from the literature strongly support the hypothesis of Crol acting on Wg expression.

36) B-D Please provide representative confocal microscope images showing the quantified effects (now in the supplement but should be in the main figure). I am puzzled by this quantification since when looking at the representative images provided in Fig5 J-R I cannot see the measured differences. For example, in J vs K I could not see an increase in the *fz3-RFP* signal. In the current images, the signal-to-noise ratio of the *fz3-RFP* is very low and does not allow the reader to make conclusions about differences. Please provide more information how the quantification was performed? From which region was it done and how were the cells selected for measurement? Is the background signal considered in the measurements? Also, there is no ISC marker provided? How were those cells identified as ISCs?

We thank all reviewers for these suggestions. By adding representative images of fz3-RFP signals with higher quality to the revised main figure (now Fig.4), we now made it easier for readers to see the differences in fz3-RFP intensities by showing RFP signal in greyscale (Fig.4E-M). We also added micrographs showing antibody staining targeting Arm and Pros (ISC/EB and EE specific), which we used to unequivocally identify progenitor cells (marked by arrows in the revised Fig.4E'-M', see comment #32). In fz3-RFP intensity measurements, no background subtraction was applied as the signal-to-noise ratio is high as is now visible in the new micrographs. Detailed information about the region (R5), cell type specific markers, as well as how measurements were performed can now be found in the new figure legends as well as the revised material and methods (lines 486-515).

Fig5

37) A The authors have developed a genetic tool, rapport-tracing, enabling transgene expression in the EC and simultaneous tracing of progenitors through ReDDM. This is a nice tool and is of interest to many in the field.

Thank you, we are convinced that Rapport is a robust tool for detection of non-autonomous effects (comment #11) and will hopefully receive similar acceptance as ReDDM tracing adding to the intestinal lineage tracing toolkit. We now include a table showing different flavours of Rapport that we already created for current and future studies.

38) B-K By using their Rapport-tracing method the authors make the conclusion that Crol controls Wnt/wg secretion from the ECs which non-autonomously activate ISC proliferation. While their results show that manipulating Crol levels in the ECs do regulate ISC proliferation, there is no evidence to support Crol works through Wnt. The very least, a genetic epistasis experiment with Wnt pathway and Crol is required to make this conclusion. In addition, some evidence is needed to show that Crol regulates genes involved in Wnt secretion in the ECs.

We kindly want to refer reviewer 3 here to comments#32-36 in which we show that Wnt/Wg signalling activity in ISC/EB is responding to EC specific manipulations of Crol and Wg.

Concerning Crol acting through Wg, we followed the reviewer's suggestion and now provide evidence that depleting Wg by RNAi downstream of UAS-crol activation in our Rapport system significantly reduces progenitor numbers (Fig.5L-N) and fz3 activity (Fig.4B,D,G-H), which supports our hypothesis in addition to data discussed in comment #34. Of note, the increase in progenitor numbers upon sole wg overexpression (Fig.5F,J,K) is weaker compared to UAS-crol (Fig.5D,J,K), and proliferation induced by UAS-crol cannot completely be rescued by wg-RNAi (Fig.5L-N). These results indicate that in addition to the here identified mitogenic Wg, additional unknown factors likely contribute to the non-autonomous stimulation of proliferation downstream of Crol. We included this new data in the manuscript in results (lines 214-217) accordingly and updated the new Fig.5.

Fig7

39) A-B If the authors wish to make conclusions about Crol regulating intestinal size, then intestinal size should be measured. Intestinal tissue size regulation is a complex process involving cell proliferation, differentiation, cell death and cell size regulation. These processes are regulated in a region-specific manner. Thus, counting cell numbers from R5 region does not equal to intestinal size. Alternatively, the conclusions about intestinal size needs to be rephrased.

We agree with the reviewer that intestinal size regulation is a complex multimodal process and now more carefully describe the according region investigated in the revised manuscript. Following this important suggestion, we now include new measurements of whole gut length, diameter (R5) and cell number of the entire midgut in VF compared to MF and added the results in Fig.7C and Fig.S7F and the main text (lines 288-290). Together, these data indicate that cell numbers of a whole midgut indeed can be used for intestinal growth under homeostatic conditions. In addition, cell numbers have also previously been used to complement and approximate intestinal size measurements⁴⁶⁻⁴⁸. Logically following these encouraging findings, we also compared total cell numbers of entire midguts with EC specific overexpression of Crol to total cell numbers of MF controls (Fig.7C, Fig.S7F). Underlining our data for R5, we observed an increase in cell numbers in the whole midgut as well (Fig.7A-C,, S7F).

In addition to the parameters mentioned by the reviewer, we chose to refine our measurements and select cell numbers as the best reference for organ size as cell numbers are known to be less affected by general stress responses of the body that may limit intestinal organ growth. An example for this is that tissue stress induced by infection results in contractions of the visceral muscle which is surrounding the midgut and strongly affect midgut size^{49,50}. Another example is that strong increases in proliferation have already been shown to result not only in increased diameters but also in tissue growth towards the gut lumen^{39,51} conflicting diameter measurements. We now more carefully discuss this in the revised manuscript's results and discussion (lines 286-290).

40) What additional value does the mathematical modelling bring to the field? How can it be used to predict intestinal adaptation? Could these topics be considered in the discussion?

The intention of our mathematical model was to test whether its mathematically plausible that organ size and cell numbers respond to a single hormonal stimulus. We then challenged this model with findings from our functional analyses of Crol and evaluated whether opposing cell type specific roles for Crol indeed are capable of controlling organ size. Following this approach, we think that the consideration of e.g. degradation of a ligand such as Wg is an important contribution for the interpretation of future data elaborated with new methods such as Rapport.

Concerning a prediction of intestinal adaptations, we want to point out here that the examples shown in Fig.S7G-I indeed allow a prediction of how our model can be applied for ISC production (Fig.S7I) in situations with constant EC numbers (Fig.S7G). By changing the parameter of EC numbers to constant ones regardless of increasing hormone levels, the model gives a logarithmic increase in ISC proliferation (Fig.S7I). Following the reviewers suggesting, we now discuss applicability of our model more extensively in the revised manuscript (lines 333-343).

Supplementary figures

FigS1

Row 485: "dsRed positive EC" – should be "dsRed positive EB" as EB marker is used here. Please use the more common abbreviation "α" for antibody instead of "a" (refers to all figures).

Thank you for pointing out this mistake, we corrected it.

FigS2

I: If control value is 0, it does not make sense to apply statistics on that.

Thank you, we removed all statistics applied to values of 0 (Fig.2L, Fig.6Q, Fig.S2C).

J: Please provide quantification for this important data about RNAi validation.

We performed the requested experiment and now added RNAi validation of crol-RNAi to Fig.S2I-L as suggested by the reviewer.

FigS3

A: again, testing against zero is not suggested.

We agree and now display Fig.2L without statistics.

FigS4

B-E I do not see an increase in wg::GFP or fz3::RFP in the images of MF sample compared to the VF sample.

We revised the visualization of these fluorophores by showing greyscale images with improved quality (Fig.S4A-E) and as pointed out by the statistics in Fig.S4C,F the difference is significant and 20% and 49% respectively. Please see comment#17 for Wg::GFP and comment #19 for fz3-RFP visualization.

Is the intensity of wg::GFP really increased or are there just more cells expressing wg? How are the ISC/EB cells identified in the quantification of fz3-RFP?

Please see our answer to comments #4 + #17, in which we detail on this issue extensively. Briefly, we added new data showing Wg::GFP measurements in EC of R5 which are more precise and appropriate to support our results. ISC/EB were unequivocally identified by antibody staining against Arm and the EE marker Pros. Please see comments #24 and #32 for details on markers and the quantifications of fz3-RFP levels. We also added a whole detailed section on how measurements of Wg::GFP, Crol::GFP and fz3-RFP were performed and normalized to the material and methods section (lines 486-515).

Reviewer #4 (Remarks to the Author):

Uncategorized References

1. Mitchell, N.C. *et al.* The Ecdysone receptor constrains wingless expression to pattern cell cycle across the *Drosophila* wing margin in a Cyclin B-dependent manner. *BMC Dev Biol* **13**, 28 (2013).
2. Koelle, M.R. *et al.* The *Drosophila* EcR gene encodes an ecdysone receptor, a new member of the steroid receptor superfamily. *Cell* **67**, 59-77 (1991).
3. White, K.P., Hurban, P., Watanabe, T. & Hogness, D.S. Coordination of *Drosophila* metamorphosis by two ecdysone-induced nuclear receptors. *Science* **276**, 114-7 (1997).
4. Zipper, L., Jassmann, D., Burgmer, S., Görlich, B. & Reiff, T. Ecdysone steroid hormone remote controls intestinal stem cell fate decisions via the PPAR γ -homolog Eip75B in *Drosophila*. *Elife* **9**(2020).
5. Walker, V.K. & Ashburner, M. The control of ecdysterone-regulated puffs in *Drosophila* salivary glands. *Cell* **26**, 269-77 (1981).
6. G, N.L. *et al.* A screen of small molecule and genetic modulators of life span in female *Drosophila* identifies etomoxir, RH5849 and unanticipated temperature effects. *Fly (Austin)* **16**, 397-413 (2022).
7. Okamoto, N. *et al.* A Membrane Transporter Is Required for Steroid Hormone Uptake in *Drosophila*. *Dev Cell* **47**, 294-305.e7 (2018).
8. Mitchell, N., Cranna, N., Richardson, H. & Quinn, L. The Ecdysone-inducible zinc-finger transcription factor Crol regulates Wg transcription and cell cycle progression in *Drosophila*. *Development* **135**, 2707-16 (2008).
9. Reiff, T. *et al.* Endocrine remodelling of the adult intestine sustains reproduction in *Drosophila*. *Elife* **4**, e06930 (2015).
10. Liang, J., Balachandra, S., Ngo, S. & O'Brien, L.E. Feedback regulation of steady-state epithelial turnover and organ size. *Nature* **548**, 588-591 (2017).
11. Tian, A., Benchabane, H., Wang, Z. & Ahmed, Y. Regulation of Stem Cell Proliferation and Cell Fate Specification by Wingless/Wnt Signaling Gradients Enriched at Adult Intestinal Compartment Boundaries. *PLoS Genet* **12**, e1005822 (2016).
12. Fang, H., Martinez-Arias, A. & de Navascués, J. Autocrine and paracrine Wingless signalling in the *Drosophila* midgut by both continuous gradient and asynchronous bursts of wingless expression [version 1; peer review: 3 approved with reservations]. *F1000Research* **5**(2016).
13. Buchon, N. *et al.* Morphological and molecular characterization of adult midgut compartmentalization in *Drosophila*. *Cell Rep* **3**, 1725-38 (2013).
14. Sawyer, J.K., Cohen, E. & Fox, D.T. Interorgan regulation of *Drosophila* intestinal stem cell proliferation by a hybrid organ boundary zone. *Development* **144**, 4091-4102 (2017).
15. Antonello, Z.A., Reiff, T., Ballesta-Illan, E. & Dominguez, M. Robust intestinal homeostasis relies on cellular plasticity in enteroblasts mediated by miR-8-Escargot switch. *EMBO J* **34**, 2025-41 (2015).
16. Antonello, Z.A., Reiff, T. & Dominguez, M. Mesenchymal to epithelial transition during tissue homeostasis and regeneration: Patching up the *Drosophila* midgut epithelium. *Fly (Austin)* **9**, 132-7 (2015).
17. Nayak, B.K. & Hazra, A. How to choose the right statistical test? *Indian J Ophthalmol* **59**, 85-6 (2011).
18. Micchelli, C.A. & Perrimon, N. Evidence that stem cells reside in the adult *Drosophila* midgut epithelium. *Nature* **439**, 475-9 (2006).
19. Ohlstein, B. & Spradling, A. The adult *Drosophila* posterior midgut is maintained by pluripotent stem cells. *Nature* **439**, 470-474 (2006).
20. Ohlstein, B. & Spradling, A. Multipotent *Drosophila* intestinal stem cells specify daughter cell fates by differential notch signaling. *Science* **315**, 988-92 (2007).
21. Tian, A. *et al.* Damage-induced regeneration of the intestinal stem cell pool through enteroblast mitosis in the *Drosophila* midgut. *EMBO J* **41**, e110834 (2022).

22. Neophytou, C., Soteriou, E. & Pitsouli, C. The Sterol Transporter Npc2c Controls Intestinal Stem Cell Mitosis and Host-Microbiome Interactions in *Drosophila*. *Metabolites* **13**(2023).
23. Robinson, P.D., Morgan, E.D., Wilson, I.D. & Lafont, R. The metabolism of ingested and injected [3H]ecdysone by final instar larvae of *Heliothis armigera*. *Physiological Entomology* **12**, 321-330 (1987).
24. Wing, K.D., Slawewski, R.A. & Carlson, G.R. RH 5849, a Nonsteroidal Ecdysone Agonist: Effects on Larval Lepidoptera. *Science* **241**, 470-2 (1988).
25. Ahmed, S.M.H. *et al.* Fitness trade-offs incurred by ovary-to-gut steroid signalling in *Drosophila*. *Nature* **584**, 415-419 (2020).
26. Liu, P., Fu, X. & Zhu, J. Juvenile hormone-regulated alternative splicing of the taiman gene primes the ecdysteroid response in adult mosquitoes. *Proc Natl Acad Sci U S A* **115**, E7738-E7747 (2018).
27. D'Avino, P.P. & Thummel, C.S. crooked legs encodes a family of zinc finger proteins required for leg morphogenesis and ecdysone-regulated gene expression during *Drosophila* metamorphosis. *Development* **125**, 1733-45 (1998).
28. Bownes, M., Dubendorfer, A. & Smith, T. Ecdysteroids in Adult Males and Females of *Drosophila-Melanogaster*. *Journal of Insect Physiology* **30**, 823-830 (1984).
29. Li, Y. *et al.* Transcription Factor Antagonism Controls Enteroendocrine Cell Specification from Intestinal Stem Cells. *Sci Rep* **7**, 988 (2017).
30. Biteau, B. & Jasper, H. Slit/Robo signaling regulates cell fate decisions in the intestinal stem cell lineage of *Drosophila*. *Cell Rep* **7**, 1867-75 (2014).
31. Hu, D.J., Yun, J., Elstrott, J. & Jasper, H. Non-canonical Wnt signaling promotes directed migration of intestinal stem cells to sites of injury. *Nat Commun* **12**, 7150 (2021).
32. Ohlstein, B. & Spradling, A. The adult *Drosophila* posterior midgut is maintained by pluripotent stem cells. *Nature* **439**, 470-4 (2006).
33. Jonkman, J., Brown, C.M., Wright, G.D., Anderson, K.I. & North, A.J. Tutorial: guidance for quantitative confocal microscopy. *Nature Protocols* **15**, 1585-1611 (2020).
34. Sun, H. *et al.* Wnt/ β -catenin signaling within multiple cell types dependent upon kramer regulates *Drosophila* intestinal stem cell proliferation. *iScience* **27**, 110113 (2024).
35. Schnabl, B., Valletta, D., Kirovski, G. & Hellerbrand, C. Zinc finger protein 267 is up-regulated in hepatocellular carcinoma and promotes tumor cell proliferation and migration. *Exp Mol Pathol* **91**, 695-701 (2011).
36. Yang, H. *et al.* Knockdown of zinc finger protein 267 suppresses diffuse large B-cell lymphoma progression, metastasis, and cancer stem cell properties. *Bioengineered* **13**, 1686-1701 (2022).
37. Lopez-Garcia, J. *et al.* ZNF366 is an estrogen receptor corepressor that acts through CtBP and histone deacetylases. *Nucleic Acids Res* **34**, 6126-36 (2006).
38. Stossi, F. *et al.* Transcriptional Profiling of Estrogen-Regulated Gene Expression via Estrogen Receptor (ER) α or ER β in Human Osteosarcoma Cells: Distinct and Common Target Genes for These Receptors. *Endocrinology* **145**, 3473-3486 (2004).
39. Patel, P.H., Dutta, D. & Edgar, B.A. Niche appropriation by *Drosophila* intestinal stem cell tumours. *Nat Cell Biol* **17**, 1182-92 (2015).
40. Jin, Y. *et al.* EGFR/Ras Signaling Controls *Drosophila* Intestinal Stem Cell Proliferation via Capicua-Regulated Genes. *PLOS Genetics* **11**, e1005634 (2015).
41. Port, F., Muschalik, N. & Bullock, S.L. Systematic Evaluation of *Drosophila* CRISPR Tools Reveals Safe and Robust Alternatives to Autonomous Gene Drives in Basic Research. *G3 (Bethesda)* **5**, 1493-502 (2015).
42. Zipper, L., Batchu, S., Kaya, N.H., Antonello, Z.A. & Reiff, T. The MicroRNA miR-277 Controls Physiology and Pathology of the Adult *Drosophila* Midgut by Regulating the Expression of Fatty Acid beta-Oxidation-Related Genes in Intestinal Stem Cells. *Metabolites* **12**(2022).
43. Port, F. & Bullock, S.L. Augmenting CRISPR applications in *Drosophila* with tRNA-flanked sgRNAs. *Nat Methods* **13**, 852-4 (2016).

44. Bartscherer, K., Pelte, N., Ingelfinger, D. & Boutros, M. Secretion of Wnt ligands requires Evi, a conserved transmembrane protein. *Cell* **125**, 523-33 (2006).
45. Bänziger, C. *et al.* Wntless, a conserved membrane protein dedicated to the secretion of Wnt proteins from signaling cells. *Cell* **125**, 509-22 (2006).
46. Christensen, C.F., Laurichesse, Q., Loudhaief, R., Colombani, J. & Andersen, D.S. *Drosophila* activins adapt gut size to food intake and promote regenerative growth. *Nature Communications* **15**, 273 (2024).
47. O'Brien, L.E., Soliman, S.S., Li, X. & Bilder, D. Altered modes of stem cell division drive adaptive intestinal growth. *Cell* **147**, 603-14 (2011).
48. Hudry, B., Khadayate, S. & Miguel-Aliaga, I. The sexual identity of adult intestinal stem cells controls organ size and plasticity. *Nature* **530**, 344-8 (2016).
49. Benguettat, O. *et al.* The DH31/CGRP enteroendocrine peptide triggers intestinal contractions favoring the elimination of opportunistic bacteria. *PLOS Pathogens* **14**, e1007279 (2018).
50. Fujita, Y., Kosakamoto, H. & Obata, F. Microbiota-derived acetylcholine can promote gut motility in *Drosophila melanogaster*. *Philosophical Transactions of the Royal Society B: Biological Sciences* **379**, 20230075 (2024).
51. Jiang, H. & Edgar, B.A. EGFR signaling regulates the proliferation of *Drosophila* adult midgut progenitors. *Development* **136**, 483-93 (2009).

Dear Reviewers,

we thank all reviewers for their constructive suggestions that led to our revised manuscript. We agree with Reviewer 2 that a core advantage of using Drosophila as a model is the generation of robust datasets based on a high number of biological replicates. In this second revision, we performed a substantial number of experiments to increase biological replicates and demonstrate their reproducibility. Thanks to the comments of Reviewer #2, this second revised manuscript augments our findings in all requested experiments and the scientific quality.

Please find below a point-by-point response to each concern raised by the reviewers. We took the liberty to consecutively number the comments and refer to prior comments for similar concerns.

REVIEWER COMMENTS

Reviewer #1 (Remarks to the Author):

The authors have addressed my concerns and the manuscript seems ready for publication

We thank Reviewer #1 for his comments and positive evaluation of our revised manuscript.

Reviewer #2 (Remarks to the Author):

The authors have provided additional data and revised their statistical analyses to strengthen their working model. In their rebuttal letter, they provide a thorough and detailed response to the reviewers' individual comments.

We thank the Reviewer for his constructive contribution of our revised manuscript.

However, my main concern with this study remains and relates to the reproducibility of the data and the biological significance of the findings. The authors base their conclusions and model on a limited number of biological replicates. Although they quantify signal intensities in a large number of cells, these cells are often derived from only three to four adult Drosophila guts.

We agree with the Reviewer and increased the number of biological replicates by at least a factor of two that stem from at least three independent experiments. Importantly, all performed experiments confirmed our previous findings as shown in the new figures (Fig.1,3,4,S1,S4). Please find details below in 3).

2) Given the inherent and well-documented variability among adult flies, coupled with the fact that the effect of RH5849 requires feeding (NOTE: administration of RH5849 has been shown to reduce food intake by approximately 75%, as reported by Landis et al., 2022, Fly,

<https://doi.org/10.1080/19336934.2022.2149209>), I find the number of biological replicates insufficient to support the conclusions drawn in this study.

We agree with the Reviewer that a higher number of independent replicates augments statistical significance even further and increased the number of replicates and independent experiments here. As the reviewer points out with the cited publication, conducting experiments by feeding small molecules is not trivial and indeed results in some variability. We have now more than ten years of experience in conducting feeding experiments (Reiff et al., 2015/Zipper et al., 2020) in which we make sure that substances are taken up readily by two major measures:

- 1) We starve the flies for 4 hours before the experiment, which leads to immediate feeding behavior.*
- 2) When we test new substances, we perform preliminary experiments in which we visualize uptake of given substances by adding the inert blue dye (Erioglaucine Disodium Salt (E133) (BLD Pharmatech Ltd., Shanghai, China)) known from the so called 'Smurf assay' allowing to determine its passage time to the abdominal part of the gut and defecation (Rera et al, 2011). In preliminary experiments with starved female and male flies fed with RH5849/blue dye and control flies fed with dye only, we did not detect any delayed food uptake and determined over a timeline of experiments that after 48h RH5849 treatment elicits the most pronounced effects in females (Zipper et al, 2020) but no effect is observed in males (this study).*

We added a more precise description on feeding experiments to the Materials and Methods section (lines 444-454).

3) This concern is particularly relevant for the quantification of signal intensities, such as Crol::GFP, wg::GFP, and fz3-RFP in the cells of the R5 region, as well as comparisons between virgin/mated female flies, control/ovoD1, and MeOH/RH5849 treatments. One of the major and appreciated advantages of using the *Drosophila* model is the ability to generate robust, reproducible data from a large number of biological replicates in independent, temporally-separated experiments. Given that the differences observed are modest, less than 20%, I would expect that the experiment should have been blinded to avoid bias in the interpretation.

As pointed out in 1), we raised the number of biological replicates/independent experiments and detected identical statistical differences in all comparisons. For simplicity in this rebuttal letter, we provide a side-by-side comparison of the initial data and the new comprehensive data for the second revision (see appendix). All new data is incorporated into the new figures of the second revision (Fig.1,3,4,S1,S4).

The reviewer states that some of the observed changes are in a modest range. In ongoing- as well as in past studies on pregnancy related adaptations, we observe modest but also significant and consistent changes. Importantly, these changes can be reproduced robustly over the years and different experiment sets and experimentators. At HHU (Heinrich Heine University) and in line with the DFG (German Research Council), we commit to the rules of good scientific practice and experiments are analysed blinded.

As our physiological (mating, Fig.1,S4), pharmacological (RH5849, Fig.1,3) and genetic (ovoD1, Fig.3,S1) augmentations of Ecdysone signaling unisono confirm the genetic experiments (Figs.1,4,5,S5), we are convinced that we provide sufficient lines of evidence to support our conclusions according to the state of the art.

4) Additionally, in the authors' response to reviewer comment 24 and in the revised Materials and Methods section, they state that fluorescence signal intensities were normalized to mating-induced area changes. However, they do not provide any evidence to support the claims of "dilution of actual hormone dosage" or "dilution and degradation of Wg ligands." For this normalization approach to be valid, the authors need to present unequivocal data demonstrating these dilution effects.

As stated in comment #24, we performed normalization to mating-induced area changes to visualize the actual hormone dosage that reaches a single cell in a more numerous mating adapted intestine. As a consequence, normalizing data is a necessity that is shown in our mathematical modelling and experiments, which then proves an underlying dilution of hormone dosage per cell (Fig.7A-C,G,G' and S7E,F) as well as the known degradation of Wg ligands (Fig.7G,G') (Dubois et al, 2001) when ISC numbers are constant (S7E,F).

Since the response of EC to 20HE depends on the concentration of the former, dilution is actually a logical consequence that can be extracted using simplicity arguments; actually proving the necessity for normalization. The reasoning runs as follows

1/ First of all, the release of 20HE from ovaries to the gut is independent of gut growth

2/ The following molecular players (20HE-Crol-Wg) form part of gut size adaptation and trigger the increase on EC cell number. 20HE-Crol-Wg increases are independent of normalization (see attached Figure without normalization of the same dataset).

3/ The number of EC cells increases whereas, according to 1/, the amount of hormone does not

4/ In consequence, the claim that concentration is not decreasing --which implies substituting 1/ by another statement-- would necessarily refer to a speculative feedback mechanism from the gut after the mating-dependent growth finished that would dampen 20HE-release by the ovaries. Such a mechanism is unknown, would complexify the explanation and its postulation is not required to explain the observed data.

Our choice is thus lead by Occam's razor-like reasoning, since including other mechanisms than dilution and the already demonstrated degradation of Wg-ligands would be speculative and add another layer of unnecessary complexity to our proposed mechanism and its subsequent mathematical model. To better illustrate this, we added a scheme with our evidence from the figures on the next page.

Scheme depicting our evidence including the according data and indicating the figures.

Comparisons of original fluorophore intensities without normalization

A Crol protein levels upon mating induced 20HE, administration of 20HE agonist and ablation of ovaries

B Wnt/Wg ligand expression and signalling activity upon mating induced 20HE, administration of 20HE agonist and ablation of ovaries

(A) Quantification of Crol::GFP intensities in ISC/EB and EC upon 20HE levels increased by mating, oral administration of 20HE agonist RH5849 and genetic ablation of ovaries. (B-C) Quantification of (B) Wnt/Wg ligand expression in EC and (C) Wnt/Wg signalling activity in ISC/EB upon 20HE levels increased by mating, oral administration of 20HE agonist RH5849 and genetic ablation of ovaries. Scatter dot plots show individual values with indication of means and standard deviations. Asterisks denote significances from (B) Mann Whitney U tests (**** $p < 0.0001$).

5) The functional genetic and epistasis experiments are the stronger part of the study. At the same time, I find it interesting to see the same controls in response 20 and Figure 2K. How many times have these experiments been repeated. Are the data all coming from one experimental trial?

We agree with the reviewer that genetic manipulations such as up- and downregulation of factors strongly reveal underlying mechanisms and thus support our findings in physiology of mating and the pharmacology. Concerning the replicates, numbers of guts are indicated with a lowercase n in the figures. These biological replicates stem from at least three different independent experimental trials in the second revision including according parallel controls.

Reviewer #3 (Remarks to the Author):

The authors have adequately addressed all of my concerns, and I find this manuscript to fulfill the high standards required for publication in Nature Communications.

Congratulations for the authors for their nice study.

We thank Reviewer #3 for his comments and the appreciation of our study.

Reviewer #4 (Remarks to the Author):

We thank Reviewer #4 for his comments and the positive evaluation of our study.

Appendix: Side-by-side comparison of previous and 2nd revision experiments.

Fig.1F

Fig.1I

Fig.1J

Fig.3I

Fig.3J

Fig.4B

Fig.4 C,D

Fig.S1F

Fig.S11

Fig.S4C,F

1st revision

2nd revision

References

- Dubois L, Lecourtois M, Alexandre C, Hirst E, Vincent JP (2001) Regulated endocytic routing modulates wingless signaling in *Drosophila* embryos. *Cell* 105: 613-624
- Rera M, Bahadorani S, Cho J, Koehler CL, Ulgherait M, Hur JH, Ansari WS, Lo T, Jr., Jones DL, Walker DW (2011) Modulation of longevity and tissue homeostasis by the *Drosophila* PGC-1 homolog. *Cell metabolism* 14: 623-634
- Zipper L, Jassmann D, Burgmer S, Gorlich B, Reiff T (2020) Ecdysone steroid hormone remote controls intestinal stem cell fate decisions via the PPARgamma-homolog Eip75B in *Drosophila*. *Elife* 9: e55795

The authors provided satisfactory answers to my questions. The repetition of the experiments and the addition of the data to the manuscript strengthen their conclusions. It is very encouraging to see the reproducibility of the phenotypes in the authors' hands. As such, the study provides a solid and exciting foundation for future follow-up studies.

I would like to invite the authors to review the figure below, which outlines the benefits of using a color-blind-friendly palette. This could enhance accessibility of their study to general readership.